# Community Detection for Hypergraph Networks via Regularized Tensor Power Iteration

## Abstract

To date, social network analysis has largely focused on pairwise interactions. In contrast, the study of higher-order interactions through hypergraph representations offers new insights. In this work, we investigate community detection in such settings. A natural baseline approach is to project the hypergraph onto a standard graph and then apply existing community detection methods; however, we show that this projection can lead to substantial information loss. To address this issue, we propose a new method that operates directly on the hypergraph structure. At the core of our approach is a regularized higher-order orthogonal iteration (reg-HOOI) algorithm, which computes an approximate low-rank decomposition of the adjacency tensor associated with the hypergraph. Compared to existing tensor decomposition methods, such as HOSVD and the vanilla HOOI algorithm, reg-HOOI achieves improved performance, particularly in sparse settings. Building on this decomposition, we further extend the SCORE method (Jin, 2015) from graph-based networks to hypergraphs, resulting in a new procedure that we term Tensor-SCORE.

On the theoretical side, we introduce a degree-corrected block model for hypergraphs (hDCBM) and show that Tensor-SCORE achieves consistent community detection across a wide range of network sparsity levels and degree heterogeneity. As a byproduct, we establish convergence rates for estimating the principal subspace using reg-HOOI under different initialization schemes, including two new initialization methods we propose: a diagonal-removed HOSVD and a randomized graph projection.

We apply our method to both a legislator hypergraph and a disease hypergraph, obtaining encouraging results. These findings suggest that modeling higher-order interactions captures information that is not preserved in standard graph-based representations.

**Keywords:** concentration inequality, reg-HOOI, HOSVD, incoherent operator norm, non-uniform hypergraph, tensor PCA

**Mathematics Subject Classification (2020):** 62H30

# 1 Introduction

Social networks is a standard tool for representing complex social relations. Conventional social networks are *graph* networks which only represent pairwise interactions. However, many social relationships involve more than two subjects, and *hypergraph* networks become natural representations. A hypergraph consists of a set of nodes $V$ and a set of hyperedges $E$. Each hyperedge is a subset of $V$; if this subset contains $m$ nodes, it is called an order-$m$ hyperedge. A graph can

Table 1: Hypergraph network data sets analyzed in this paper.

| Dataset | Node | Hyperedge | #Nodes | #Hyperedges |
|---------|------|-----------|--------|-------------|
| Legislator | legislator | bill (multiple sponsors) | 120 | 802 |
| MEDLINE | disease | paper (multiple diseases mentioned) | 190 | 9,351 |

be viewed as a special hypergraph with only order-2 hyperedges. There are many examples of hypergraph networks, and Table 1 lists the two datasets we will analyze in this paper.

It is believed that, going beyond pairwise interactions, the study of higher-order social interactions is useful to bring new insight of underlying social connections (Benson et al., 2016). For example, higher-order interactions of species are found to be a possible mechanism for maintaining a stable coexistence of diverse competitors (Grilli et al., 2017). In this paper, we are interested in utilizing a hypergraph network for community detection, that is, to cluster nodes into socially similar groups ("communities").

A natural baseline for analyzing hypergraph networks is to project them onto graphs and apply existing methods designed for graph networks. In many cases, this projection is performed implicitly during data preprocessing. For example, many network datasets were originally hypergraphs but have been converted into graphs by replacing each order-$m$ hyperedge with an $m$-clique. Examples include the academic collaboration networks (Grossman, 2002; Newman, 2001; Ji and Jin, 2017), which are converted from coauthorship hypergraphs, and the congress voting networks (Fowler, 2006; Lee et al., 2016), which originate from co-sponsorship hypergraphs. Even when the hypergraph representation is retained, the subsequent statistical analysis often involves an implicit projection to a graph. A notable example is the use of hypergraph Laplacian (Zhou et al., 2007; Ghoshdastidar and Dukkipati, 2017; Chien et al., 2018; Kim et al., 2018), which actually transforms the hypergraph into a weighted graph, where the edge weight between two nodes is proportional to the counts of hyperedges that involve both nodes.

However, the projection-to-graph approach can lead to substantial information loss. This phenomenon is illustrated in the Legislator dataset. In Figure 1, we compare the spectral embeddings (see Section 5.2 for details) obtained from the projected graph and the original hypergraph. The nodes (i.e., legislators) are known to belong to seven parties. In the projected graph, the parties are largely indistinguishable, whereas in the original hypergraph they are clearly separable. This loss of information can also be characterized theoretically. In Section 3.4, we present an example showing that some nonzero population "eigenvalues" of the hypergraph become zero after projection to a graph. This observation motivates our development of community detection methods that operate directly on hypergraphs.

A hypergraph is represented by an adjacency tensor. Let $H = (V, E)$ denote a hypergraph with $n$ nodes, where $V = \{1, 2, \ldots, n\}$. When each hyperedge $e \in E$ has an order $m$, we call $H$ an *$m$-uniform hypergraph*. We focus primarily on uniform hypergraphs throughout this paper, and extend our methodology to non-uniform hypergraphs in Section 3.5. The adjacency tensor of an $m$-uniform hypergraph is an $m$-way array $\mathcal{A} \in \mathbb{R}^{n^m}$, where, for all $1 \leq i_1, \ldots, i_m \leq n$,

$$\mathcal{A}(i_1, \ldots, i_m) = \begin{cases} 1, & \text{if } \{i_1, \ldots, i_m\} \text{ is a hyperedge,} \\ 0, & \text{otherwise.} \end{cases}$$

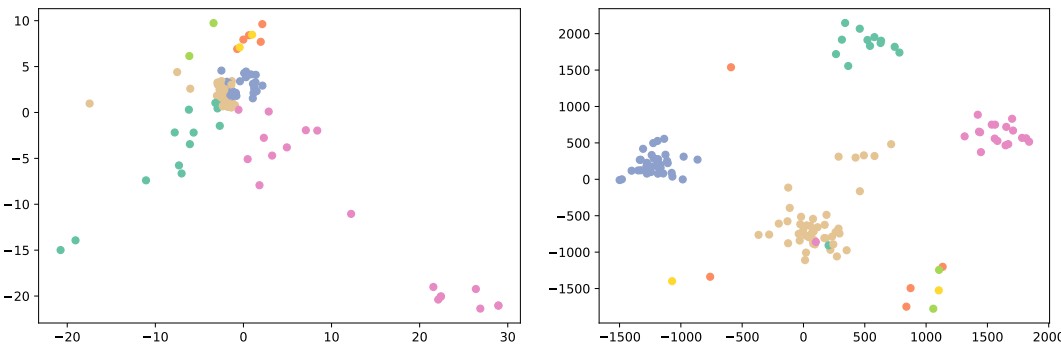

Figure 1: Projecting a hypergraph onto a graph leads to substantial information loss (dataset: Legislator). Left: spectral embedding of the projected graph. Right: spectral embedding of the original hypergraph. Colors indicate party afflictions in the Peruvian Congress. The separation of parties is much clearer in the right panel, leading to improved recovery of the true communities (clustering error: 48.3% on the left versus 11.7% on the right).

By convention, we exclude self-hyperedges, so $\mathcal{A}(i_1, \ldots, i_m) = 0$ when any of $i_1, \ldots, i_m$ are equal.

Our methodology involves applying a Tucker decomposition (Tucker, 1966) on $\mathcal{A}$. The Tucker decomposition is a form of higher order PCA, and it outputs a factor matrix whose columns will be used as "eigenvectors" of the tensor. Popular methods for computing the Tucker decomposition include higher-order singular value decomposition (HOSVD) (De Lathauwer et al., 2000a) and higher-order orthogonal iteration (HOOI) (De Lathauwer et al., 2000b). HOSVD unfolds the tensor to a huge rectangular matrix and conducts singular value decomposition. HOOI starts with an initial solution and runs power iteration. Statistical properties of these methods have been studied primarily under the assumption that the data tensor has Gaussian entries. For example, it was shown that HOSVD is statistically sub-optimal (Richard and Montanari, 2014) and that HOOI can significantly improve HOSVD and attain statistical optimality (Zhang and Xia, 2018). In this paper, we show that similar insight holds when the data tensor is a hypergraph adjacency matrix. Different from the Gaussian setting in Richard and Montanari (2014); Zhang and Xia (2018); Xia and Zhou (2019), the adjacency tensor can be extremely sparse, so we have to carefully modify the HOOI algorithm. We propose a regularized HOOI (reg-HOOI), where at each power iteration, we truncate large entries of the factor matrix to control its $\ell^\infty$-norm. Our method can reliably compute the Tucker decomposition of a hypergraph adjacency tensor, even when it is very sparse.

The Tucker decomposition produces a factor matrix $\hat{\Xi} \in \mathbb{R}^{n \times K}$, where $K$ is the pre-specified Tucker rank, which we set equal to the number of communities. Let $\hat{\xi}_1, \ldots, \hat{\xi}_K \in \mathbb{R}^n$ denote the columns of $\hat{\Xi}$. These vectors can be viewed as "eigenvectors" of $\mathcal{A}$, and we use them for community detection. A straightforward approach is to apply $k$-means clustering to rows of $\hat{\Xi}$, but it performs unsatisfactorily because $\hat{\xi}_1, \ldots, \hat{\xi}_K$ are heavily influenced by degree heterogeneity. A similar phenomenon has been observed for standard graph networks, and Jin (2015) introduced an operation for removing the effects of degree heterogeneity from eigenvectors, referred to as *scale-invariant mapping*. Let $h : \mathbb{R}^K \mapsto \mathbb{R}_+$ be a homogeneous function, i.e., satisfying $h(ax) = ah(x)$ for all $x \in \mathbb{R}^K$ and $a > 0$). The scale-invariant mapping is a row-wise normalization on $\hat{\Xi}$:

$$\hat{r}_i = \frac{1}{h(\hat{x}_i)}\hat{x}_i, \qquad \text{where } \hat{x}_i \in \mathbb{R}^K \text{ is the } i\text{-th row of } \hat{\Xi}, \qquad 1 \le i \le n. \tag{1.1}$$

There are many possible choices of $h$; a particularly common special case is $h(x)$ equal to the first coordinate of $x$, which corresponds to the SCORE normalization (Jin, 2015). Whether such scale-invariant mappings are useful for hypergraphs is not immediate, since the $\hat{\Xi}$ in our setting arises from a Tucker decomposition of a tensor rather than a spectral decomposition of a matrix. Under a degree-corrected block model for hypergraphs, we generalize the insight of Jin (2015) and show that the SCORE normalization remains valid.

Combining the above results, we propose our method, *tensor-SCORE*, a two-step algorithm for community detection in hypergraph networks. In the first step, we apply the newly developed reg-HOOI algorithm to obtain $\hat{\Xi}$. In the second step, we compute $\hat{r}_1, \dots, \hat{r}_n$ using (1.1) and then perform $k$-means clustering on these vectors.

## 1.1   Related works

There is a relatively small literature on spectral methods for hypergraph community detection. One approach projects the hypergraph onto a weighted graph and performs standard spectral clustering, either based on the adjacency matrix (Ghoshdastidar and Dukkipati, 2015a) or the Laplacian (Zhou et al., 2007; Ghoshdastidar and Dukkipati, 2017). Another approach directly performs HOSVD on the adjacency tensor and clusters nodes using the resultant factor matrix (Ghoshdastidar and Dukkipati, 2014, 2015b).

Our method has major differences from the above ones. First, we use tensor power iteration to obtain the spectral decomposition of a hypergraph. This is significantly different from the projection-to-graph approach (Ghoshdastidar and Dukkipati, 2015a; Zhou et al., 2007; Ghoshdastidar and Dukkipati, 2017) and the HOSVD approach (Ghoshdastidar and Dukkipati, 2014, 2015b). The projection-to-graph approach could result in information loss (see Section 3.4) in certain settings and hence, its success requires additional conditions on the spectrum of the projected graph and these conditions are hard to check in practice. The HOSVD approach faces no such issue, but it yields a slower rate of convergence than that of ours (see Section 4). Second, the existing methods rely on a relatively ideal assumption of *no degree heterogeneity*. In contrast, we consider a much more realistic setting where we allow for *(potentially severe) degree heterogeneity*. To our best knowledge, there are no existing methods for hypergraph community detection that allow for degree heterogeneity.

Another line of works studied statistical limits of community detection in hypergraphs (Angelini et al., 2015; Ahn et al., 2017; Lin et al., 2017; Chien et al., 2018; Kim et al., 2018). These works are primarily interested in extremely sparse networks where the average node degree is bounded. To get sharp minimax rate (Lin et al., 2017; Chien et al., 2018) or phase transition threshold (Ahn et al., 2017; Kim et al., 2018), they have to restrict to a relatively narrow model where the network has no degree heterogeneity and satisfies particular symmetry conditions (e.g., equal probability for cross-community hyperedges, equal-size communities, etc.). In comparison, our focus is to develop practical algorithms for real networks, and it is crucial to work on a realistic model that allows degree heterogeneity and does not need symmetry conditions. We also note that our method applies to a wide range of sparsity levels such that the average node degree can grow with $n$ slowly at a logarithmic rate.

A key component of our method is the regularized power iteration algorithm applied to the

network adjacency tensor. This is connected to recent interests in applying power iteration for tensor decomposition and tensor completion (Richard and Montanari, 2014; Hopkins et al., 2015; Xia et al., 2021; Liu et al., 2022; Zhang and Xia, 2018). These works assume the data tensor has Gaussian entries, so the results are not applicable here. We recognize that the standard power iteration is unsatisfactory in our setting, especially when the adjacency tensor is very sparse. We introduce a regularized version of power iteration and show that it yields the desired accuracy on the output factor matrix. In particular, we prove that power iteration can significantly improve HOSVD, generalizing the conclusion of Richard and Montanari (2014); Zhang and Xia (2018) from Gaussian settings to network settings.

Another key component of our method is the SCORE normalization (1.1) on the factor matrix from power iteration. This is connected to existing works on community detection for graph networks (e.g., Bickel and Chen (2009); Rohe et al. (2011)), especially those that accommodate degree heterogeneity (Jin, 2015; Qin and Rohe, 2013; Zhang et al., 2020; Gao et al., 2018). The approach of normalizing eigenvectors to remove degree effects has been well understood in community detection for graph networks, but its extension to hypergraph networks is yet to be attempted. We show that similar ideas continue to work for hypergraph networks.

After our manuscript first appeared on arXiv, a number of interesting works on tensor and hypergraph analysis have emerged. While related to our work, these papers differ from ours in several important aspects. Some focus on alternative but related models. For example, Hu and Wang (2022) extends hDCBM to non-binary hypergraphs, Agterberg and Zhang (2025) goes beyond community detection to study mixed-membership estimation, and Zhen and Wang (2023) considers a more general formulation of degree heterogeneity. Other works develop alternative algorithms for community detection. In particular, Yu and Zhu (2025) proposes likelihood-based methods, while Wang et al. (2023) develops a projected tensor power iteration algorithm. There is also a growing literature on statistical inference problems. For example, Yuan et al. (2022) develops delicate testing procedures for detecting the existence of communities, and Jin et al. (2021) establishes sharp information-theoretic lower bounds. Another active direction concerns temporal dependence. For instance, Zhu and Yao (2026) introduces an autoregressive model for hypergraphs, and Han et al. (2024) studies matrix time series that are modeled through low-rank tensors. Despite these exciting developments, our work continues to play an important role in the literature. In particular, our model, methodology, and theoretical analysis provide a useful foundation for several of these subsequent developments.

## 1.2 Our contributions

- We propose Tensor-SCORE as a new method for hypergraph community detection. This is the first method that uses tensor power iteration technique for community detection and also the first method that accommodates degree heterogeneity.

- We introduce a degree-corrected block model for hypergraphs (hDCBM), which generalizes the stochastic block model for hypergraphs (hSBM) by permitting degree heterogeneity. Under hDCBM, we show that our method achieves consistent community detection for a wide range of sparsity and degree heterogeneity. In the special case of hSBM, the error rate of our method is faster than those of HOSVD-based methods and compares favorably

with the best known rate of spectral methods.

- A key component of our method is a regularized power iteration algorithm for computing the Tucker decomposition of a hypergraph adjacency tensor. Our theory provides the first theoretical guarantee of power iteration on hypergragh tensors, which complements the known results for Gaussian tensors.

- We apply our method to two real hypergraph datasets. The results provide clear evidence that modeling higher-order interactions captures information that is not accessible through pairwise interactions alone.

The remainder of this paper is organized as follows. In Section 2, we review the background of tensor decomposition and introduce necessary terminologies. Readers who are familiar with these materials can skip this section. In Section 3, we describe our method, Tensor-SCORE, and explain its rationale. Section 4 contains the main theoretical results, where we conduct sharp spectral analysis of power iteration and present the rate of convergence of Tensor-SCORE. Section 5 contains simulations and real data results. Discussions can be found in Section 6, and proofs are relegated to the appendix.

## 2   Some background of tensor decomposition

We review some terminology of tensors that will be repeatedly used in this paper. We also review the standard tensor decomposition methods, such as HOSVD and HOOI. Readers who are familiar with these materials can skip this section.

We call $\mathcal{X} = \{\mathcal{X}(i_1, \ldots, i_m)\}_{1 \leq i_1, \ldots, i_m \leq n}$ an $m$-way tensor of dimension $n$. A tensor $\mathcal{X}$ is supersymmetric if $\mathcal{X}(i_1, \ldots, i_m) = \mathcal{X}(j_1, \ldots, j_m)$ whenever $(j_1, \ldots, j_m)$ is a permutation of $(i_1, \ldots, i_m)$. Let $T^m(\mathbb{R}^n)$ denote the set of all $m$-way tensors of dimension $n$, and let $S^m(\mathbb{R}^n)$ be the subset of supersymmetric tensors.

**Definition 2.1** (Diagonal of a tensor). *For any tensor $\mathcal{X} \in T^m(\mathbb{R}^n)$, $\mathrm{diag}(\mathcal{X})$ is the tensor whose $(i_1, \ldots, i_m)$ element is equal to the corresponding element of $\mathcal{X}$ if some indices of $i_1, \ldots, i_m$ are identical, and is equal to $0$ otherwise.*

Our definition of $\mathrm{diag}(\mathcal{X})$ is different from the conventional one (e.g., in Kolda and Bader (2009)). We use this definition for its convenience in dealing with self-edges.

Matricization is the operation of re-arranging a tensor $\mathcal{X} \in T^m(\mathbb{R}^n)$ into an $n \times n^{m-1}$ matrix. In this paper, we are primarily interested in the mode-1 matricization, denoted as $\mathcal{M}_1(\mathcal{X})$, where the $(i_1, \ldots, i_m)$-th element of $\mathcal{X}$ is mapped to the $(i_1, 1 + \sum_{k=2}^m (i_k - 1)n^{k-2})$-th element of the matrix. It is more convenient to define matricization using slices of a tensor. A slice of a tensor is a matrix by fixing $(m - 2)$ of the indices. Let $\mathcal{X}^{(i_3, \ldots, i_m)} \in \mathbb{R}^2$ be a slice of $\mathcal{X}$ by fixing the last $(m - 2)$ indices, and there are a total of $n^{m-2}$ such slices. Then, the matricization of $\mathcal{X}$ is formed by placing these slices together. For example, for $m = 4$, $\mathcal{M}_1(\mathcal{X}) = [\mathcal{X}^{(11)}, \mathcal{X}^{(12)}, \ldots, \mathcal{X}^{(nn)}] \in \mathbb{R}^{n \times n^3}$.

We can multiply a tensor by one matrix or multiple matrices. For a tensor $\mathcal{X} \in T^m(\mathbb{R}^n)$, a matrix $U \in \mathbb{R}^{K \times n}$, and an integer $1 \leq j \leq m$, the $j$-mode product $\mathcal{X} \times_j U$ is an $m$-way tensor with only $n^{m-1}K$ entries, where the $j$-th index only ranges from 1 to $K$ and the

$(i_1, \ldots, i_{j-1}, k, i_{j+1}, \ldots, i_m)$-th element of $\mathcal{X} \times_j U$ equals to $\sum_{i_j=1}^n \mathcal{X}(i_1, \ldots, i_m)U(k, i_j)$. We can multiply $\mathcal{X} \times_j U$ by $U$ again on another mode $k \neq j$. In this paper, we often use two forms of multiplications: $\mathcal{X} \times_2 U \times_3 \cdots \times_m U$, which is an $n \times K \times \cdots \times K$ tensor, and $\mathcal{X} \times_1 U \times_2 \cdots \times_m U$, which is an $m$-way tensor of dimension $K$. The spectral norm of $\mathcal{X}$ is defined as $\|\mathcal{X}\| = \max_{h_1, \ldots, h_m \in \mathbb{R}^n : \|h_k\|=1} \mathcal{X} \times_1 h_1^\top \times_2 \cdots \times_m h_m^\top$.

**Definition 2.2** (Tucker decomposition)**.** *For a tensor $\mathcal{X} \in T^m(\mathbb{R}^n)$, if there exists $\mathcal{C} \in T^m(\mathbb{R}^K)$ and $B_1, \ldots, B_m \in \mathbb{R}^{n \times K}$ such that, $\mathcal{X} = \mathcal{C} \times_1 B_1 \times_2 \cdots \times_m B_m$, or equivalently, for all $1 \leq i_1, \ldots, i_m \leq n$,*

$$\mathcal{X}(i_1, \ldots, i_m) = \sum_{1 \leq k_1, \ldots, k_m \leq K} \mathcal{C}(k_1, \ldots, k_m) \prod_{j=1}^m B_j(i_j, k_j),$$

*then we write $\mathcal{X} = [\mathcal{C}; B_1, \ldots, B_m]$ and call it a rank-$K$ Tucker decomposition of $\mathcal{X}$, where $\mathcal{C}$ is called a core tensor and $B_1, \ldots, B_m$ are called factor matrices.*

The Tucker decomposition is not unique. For example, for an orthogonal matrix $O \in \mathbb{R}^{K \times K}$, let $\mathcal{C}^* = \mathcal{C} \times_1 O \times_2 \cdots \times_m O$ and $B_m^* = B_m O^\top$. It is easy to see that $[\mathcal{C}; B_1, \ldots, B_m] = [\mathcal{C}^*; B_1^*, \ldots, B_m^*]$. Therefore, what we care is the column space of the factor matrices.

Given $K \geq 1$ and a supersymmetric tensor $\mathcal{X} \in S^m(\mathbb{R}^n)$, it is interesting to find its approximate rank-$K$ Tucker decomposition, i.e., to find $\mathcal{C} \in S^m(\mathbb{R}^K)$ and $B \in \mathbb{R}^{n \times K}$ such that $\mathcal{X} \approx [\mathcal{C}; B, \ldots, B]$. The factor matrix of $B$ is of particular interest. We review two common methods for extracting $B$. One is higher-order singular value decomposition (HOSVD) (De Lathauwer et al., 2000a). It conducts SVD on the mode-1 matricization $\mathcal{M}_1(\mathcal{X}) \in \mathbb{R}^{n \times n^{m-1}}$ and takes the first $K$ left singular vectors to form $B$. The other is higher-order orthogonal iteration (HOOI) (De Lathauwer et al., 2000b). It starts from an initial solution $B^{(0)}$ and iteratively updates $B^{(k)}$ to $B^{(k+1)}$ as follows: Obtain $G^{(k)} = \mathcal{M}_1(\mathcal{X} \times_2 (B^{(k)})^\top \times_3 \cdots \times_m (B^{(k)})^\top) \in \mathbb{R}^{n \times K^{m-1}}$. Conduct SVD on $G^{(k)}$ and take the first $K$ left singular vectors to form $B^{(k+1)}$.

# 3 Tensor-SCORE for hypergraph community detection

We first introduce a degree-corrected block model for hypergraphs in Section 3.1. Next, in Section 3.2, we consider an oracle case where the probability of all hyperedges are known, so that a "signal" tensor is available. We study the output of tensor decomposition and explain the rationale of the normalization (1.1) in our setting. In Section 3.3, we introduce our method, Tensor-SCORE, where the key behind it is a regularized HOOI algorithm for reliably extracting a factor matrix from $\mathcal{A}$. Last, in Section 3.4, we compare our method with the approach of projecting a hypergraph to a graph, and in Section 3.5, we discuss the extension to non-uniform hypergraphs.

## 3.1 The hDCBM model and its tensor form

Recall that we have an $m$-uniform hypergraph $H = (V, E)$, where $V = \{1, 2, \ldots, n\}$. We assume that the nodes in the network consist of $K \geq 2$ disjoint communities $V = V_1 \cup V_2 \cup \cdots \cup V_K$. Let $\theta_i > 0$ be the degree heterogeneity parameter associated with node $i$, $1 \leq i \leq n$. Let $\mathcal{P} \in S^m(\mathbb{R}^K)$ be a nonnegative symmetric tensor. We further assume that each entry of the

adjacency tensor $\mathcal{A}$ is a Bernoulli random variable, independent of others, i.e.,

$$\{\mathcal{A}(i_1,\ldots,i_m) : 1 \leq i_1 < i_2 < \ldots < i_m \leq n\} \text{ are independent.} \tag{3.1}$$

We model that

$$\mathbb{P}(\mathcal{A}(i_1,\ldots,i_m) = 1) = \mathcal{P}(k_1,\ldots,k_m)\prod_{j=1}^{m}\theta_{i_j}, \qquad \text{if } i_j \in V_{k_j}, 1 \leq j \leq m. \tag{3.2}$$

We call (3.1)-(3.2) the hypergraph degree-corrected block model (hDCBM).

In hDCBM, the low-dimensional tensor $\mathcal{P}$ characterizes the difference across communities and the parameters $\theta_i > 0$ capture the individual degree heterogeneity. When $m = 2$, hDCBM reduces to DCBM for a typical graph (Karrer and Newman, 2011). When all the $\theta_i$'s are equal, hDCBM reduces to the stochastic block model for hypergraphs (hSBM) (Ghoshdastidar and Dukkipati, 2014).

We now introduce a tensor form of hDCBM. For each $1 \leq i \leq n$, we define a membership vector $\pi_i \in \mathbb{R}^K$ such that $\pi_i(k) = 1\{i \in V_k\}$, $1 \leq k \leq K$. Then, the right handside of (3.2) can be equivalently written as $\sum_{1 \leq k_1,k_2,\ldots,k_m \leq K} \mathcal{P}(k_1,\ldots,k_m)\prod_{j=1}^{m}\theta_{i_j}\pi_{i_j}(k_j)$. This has almost the same form as in Definition 2.2, except for the case where some indices in $(i_1,\ldots,i_m)$ are equal. Write $\Pi = [\pi_1,\ldots,\pi_n]'$ and $\Theta = \mathrm{diag}(\theta_1,\ldots,\theta_n)$. We introduce a tensor $\mathcal{Q} \in S^m(\mathbb{R}^n)$ by

$$\mathcal{Q} = [\mathcal{P}; \Theta\Pi,\ldots,\Theta\Pi]. \tag{3.3}$$

Then, $\mathcal{Q}$ and $\mathbb{E}[\mathcal{A}]$ are the same except for the diagonal entries (Definition 2.1). We write $\mathcal{A} = \mathcal{Q} - \mathrm{diag}(\mathcal{Q}) + (\mathcal{A} - \mathbb{E}[\mathcal{A}])$. We view $\mathcal{Q}$ as the "signal" part and view $(\mathcal{A} - \mathcal{Q})$ as the "noise" part. The "signal" tensor admits a rank-$K$ Tucker decomposition.

## 3.2 The oracle approach

We consider an oracle case where the "signal" tensor $\mathcal{Q}$ is observable. We study the output of conducting tensor decomposition on $\mathcal{Q}$ and investigate how to use it to recover $\Pi$. The insight gained in the oracle case can be easily extend to the real case where $\mathcal{A}$ is observed.

First, we apply HOSVD to $\mathcal{Q}$. Recall that $\mathcal{M}_1(\mathcal{Q})$ is the mode-1 matricization of $Q$. The rank of this matrix is $\leq K$. Let $\lambda_1 \geq \lambda_2 \geq \ldots \geq \lambda_K$ be the singular values of $\mathcal{M}_1(\mathcal{Q})$, and let $\xi_1, \xi_2, \ldots, \xi_K \in \mathbb{R}^n$ be the associated left singular vectors. Write $\Xi = [\xi_1,\ldots,\xi_K]$. Then, $\Xi$ is the factor matrix output by HOSVD.

**Lemma 3.1** (HOSVD on $\mathcal{Q}$). *Consider the hDCBM where $\mathcal{Q} = [\mathcal{P}; \Theta\Pi,\ldots,\Theta\Pi]$. Let $d \in \mathbb{R}^K$ be the vector such that $d_k = \|\theta\|^{-1}(\sum_{i \in V_k} \theta_i^2)^{1/2}$ for $1 \leq k \leq K$ and let $D = \mathrm{diag}(d)$ be the $K \times K$ diagonal matrix whose diagonal entries come from $d$. Define a $K \times K^{m-1}$ matrix $G = \mathcal{M}_1(\mathcal{P} \times_1 D \times_2 \cdots \times_m D)$. We assume $G$ has a rank $K$. Let $\kappa_1 \geq \kappa_2 \geq \ldots \geq \kappa_K$ and $u_1, u_2, \ldots, u_K \in \mathbb{R}^K$ be the singular values and left singular vectors of $G$. Then,*

$$\lambda_k = \kappa_k\|\theta\|^m, \qquad \xi_k = \|\theta\|^{-1}\Theta\Pi D^{-1}u_k, \qquad 1 \leq k \leq K.$$

Next, we apply HOOI to $\mathcal{Q}$. Suppose this algorithm is initialized by $\widetilde{\Xi} = [\widetilde{\xi}_1,\ldots,\widetilde{\xi}_K]$ and let

$\Xi^{(j)} = [\xi_1^{(j)}, \ldots, \xi_K^{(j)}]$ be the output after $j$-th power iteration. By the design of the algorithm, $\Xi^{(j)}$ contains the first $K$ left singular vectors of $\mathcal{M}_1(\mathcal{Q} \times_2 \Xi^{(j-1)\top} \times_3 \cdots \times_m \Xi^{(j-1)\top})$.

**Lemma 3.2** (HOOI on $\mathcal{Q}$). *Consider the hDCBM where $\mathcal{Q} = [\mathcal{P}; \Theta\Pi, \ldots, \Theta\Pi]$. Let $G$ be the same as in Lemma 3.1, and assume $G$ has a rank $K$. Let $\Xi$ be the factor matrix from HOSVD. We assume the initialization $\widetilde{\Xi}$ of HOOI satisfies $\|\widetilde{\Xi}\widetilde{\Xi}' - \Xi\Xi'\| < 1$.*

- *After the first power iteration, $\Xi^{(1)}$ has the same column space as that of the HOSVD factor matrix, i.e., $\Xi^{(1)} = \Xi O$, where $O \in \mathbb{R}^{K \times K}$ is an orthogonal matrix.*

- *Starting from the second power iteration, the solution remains unchanged, i.e., $\Xi^{(j)} = \Xi^{(1)}$, for all $j \geq 2$.*

- *If it is initialized by HOSVD, the output is always the same as HOSVD, no matter how many iterations have been run; i.e., if $\widetilde{\Xi} = \Xi$, then $\Xi^{(j)} = \Xi$, for all $j \geq 1$.*

The above results show that the output of HOSVD and HOOI have similar forms: By Lemma 3.1, we can write $\Xi = \Theta\Pi V$, where $V = \|\theta\|^{-1}D^{-1}[u_1, \ldots, u_K] \in \mathbb{R}^{K \times K}$. By Lemma 3.2, the output of HOOI is only different from that of HOSVD by an orthogonal matrix $O$; it leads to $\Xi = \Theta\Pi \cdot VO$. Therefore, we can write the factor matrix from both tensor decomposition methods via a universal form:

$$\Xi = \Theta\Pi B, \qquad \text{for some } K \times K \text{ matrix } B. \tag{3.4}$$

Last, we use (3.4) to obtain a community detection method. Let $x_1, \ldots, x_n \in \mathbb{R}^K$ be the rows of $\Xi$, and let $b_1, \ldots, b_K \in \mathbb{R}^K$ be the rows of $B$. It is not hard to see that $x_i = \theta_i \cdot b_k$, for all $i \in V_k, 1 \leq k \leq K$. In the case of no degree heterogeneity, $\theta_i$'s are equal. Hence, $\{x_i\}_{i=1}^n$ take only $K$ distinct values, corresponding to $K$ communities. Therefore, we can apply $k$-means clustering to rows of $\Xi$ and exactly recover the community membership. Unfortunately, in the presence of degree heterogeneity, the rows of $\Xi$ no longer have such a nice property. We now adopt the idea of SCORE in (1.1). Given any homogeneous function $h : \mathbb{R}^K \to \mathbb{R}$, define

$$r_i = \frac{1}{h(x_i)}x_i, \qquad 1 \leq i \leq n. \tag{3.5}$$

By property of a homogeneous function, $h(\theta_i b_k) = \theta_i \cdot h(b_k)$. It follows that

$$r_i = \frac{1}{h(\theta_i b_k)} \cdot \theta_i b_k = \frac{1}{\not\theta_i \cdot h(b_k)} \cdot \not\theta_i \cdot b_k = \frac{1}{h(b_k)} \cdot b_k, \qquad \text{for } i \in V_k.$$

We see that, even though $\theta_i$'s are unknown, the above operation can automatically remove the effects of $\theta_i$'s without estimating them. Now, $\{r_i\}_{i=1}^n$ only take $K$ distinct values, and so we can apply $k$-means clustering to $r_i$'s to exactly recover the community memberships. This idea was introduced in Jin (2015). In the appendix of that paper, the author suggested two choices of $h$: The first is $h(x) = x(1)$, i.e., each $x_i$ is divided by its first coordinate. It can be shown that, under mild conditions, the first coordinate of $x_i$ is always positive, so that $r_i$ is well-defined. This operation is commonly called the SCORE normalization. The second is $h(x) = \|x\|_q$, i.e., each $x_i$ is normalized by its $\ell^q$-norm; the special case of $q = 2$ coincides with a heuristic

normalization step in spectral clustering (Ng et al., 2002). In our problem, these is no essential difference among choices of $h$. We use the SCORE normalization. For this choice of $h$, the first coordinate of $r_i$ is always equal to 1; for convenience, we re-define $r_i$ by removing its first coordinate. Write $R = [r_1, r_2, \ldots, r_n] \in \mathbb{R}^{n \times (K-1)}$. It follows that

$$R(i, k) = \xi_{k+1}(i)/\xi_1(i), \qquad 1 \le i \le n, \ 1 \le k \le K - 1. \tag{3.6}$$

These insights motivate an approach to recovering $\Pi$ from $\mathcal{Q}$, which we call *Oracle Tensor-SCORE*:

(1) Apply either HOSVD or HOOI to $\mathcal{Q}$. Let $\Xi$ be the resulting factor matrix.

(2) Obtain the matrix $R$ as in (3.6). Apply $k$-means clustering to rows of $R$.

It is easy to see that Oracle Tensor-SCORE exactly recovers the communities.

## 3.3 Tensor-SCORE

We now extend Oracle Tensor-SCORE to the real case where $\mathcal{A}$, instead of $\mathcal{Q}$, is observed. Similar to the oracle case, the first step is to conduct tensor decomposition on $\mathcal{A}$ to obtain a factor matrix $\hat{\Xi} \in \mathbb{R}^{n \times K}$. In the oracle case, this is done by HOSVD or HOOI; in fact, the two algorithms give exactly the same $\Xi$, provided that HOOI is initialized by HOSVD. Now, in the real case, our hope is that the column space of $\hat{\Xi}$ is close to the column space of $\Xi$. We measure the loss by $d(\hat{\Xi}, \Xi) = \|\hat{\Xi}\hat{\Xi}' - \Xi\Xi'\|$. In Figure 2 (left panel), we plot the evolution of $d(\hat{\Xi}, \Xi)$ in a simulation example, where we start from HOSVD and iteratively conduct power iteration (to be described below). It suggests that power iteration significantly improves the accuracy. This phenomenon can be justified in theory: In Section 4.3, we show that power iteration yields a satisfactory rate of convergence on $d(\hat{\Xi}, \Xi)$, which cannot be attained by HOSVD. Note that this has been very different from the oracle case, where power iteration starting from HOSVD does not change the solution at all (Lemma 3.2).

The next question is how to conduct power iteration properly. In Section 2, we have reviewed the standard HOOI. Given $\hat{\Xi}^{old}$, it obtains $\hat{\Xi}^{new}$ by extracting the first $K$ left singular vectors of $\mathcal{M}_1(\mathcal{A} \times_2 (\hat{\Xi}^{old})^\top \times_3 \cdots \times_m (\hat{\Xi}^{old})^\top) \in \mathbb{R}^{n \times K^{m-1}}$. In the theoretical analysis, we find out that this vanilla HOOI does not give the desired rate, due to that the maximum row-wise $\ell^2$-norm of $\hat{\Xi}$ is not well controlled during the iteration. We propose a regularized HOOI (reg-HOOI), where we "regulate" the maximum row-wise $\ell^2$-norm of $\hat{\Xi}$ at each iteration. In detail, for any $\delta \in (0, 1]$, define $\mathbb{O}_\delta^{n \times K} := \{U \in \mathbb{R}^{n \times K} : U^\top U = I_K, \ \max_{j \in [n]} \|e_j^\top U\| \le \delta\}$. For any matrix $\Xi \in \mathbb{R}^{n \times K}$, its "projection" to $\mathbb{O}_\delta^{n \times K}$ is defined by ((Keshavan et al., 2010, Remark 6.2))

$$\mathcal{P}_{\mathbb{O}_\delta^{n \times K}}(\Xi) = \text{left singular vectors of } \widetilde{\Xi}, \quad \text{where } \widetilde{\Xi}(i, :) = \Xi(i, :) \frac{\min\{\delta, \|\Xi(i, :)\|\}}{\|\Xi(i, :)\|}. \tag{3.7}$$

Given $\hat{\Xi}^{old}$, we first compute $\hat{\Xi}_*^{new}$ using the vanilla HOOI, then we project $\hat{\Xi}_*^{new}$ to $\mathbb{O}_\delta^{n \times K}$ to obtain $\hat{\Xi}^{new}$. This gives the reg-HOOI algorithm.

We then introduce an empirical counterpart of the matrix $R$ in (3.6). Let $\hat{\Xi}$ be the factor

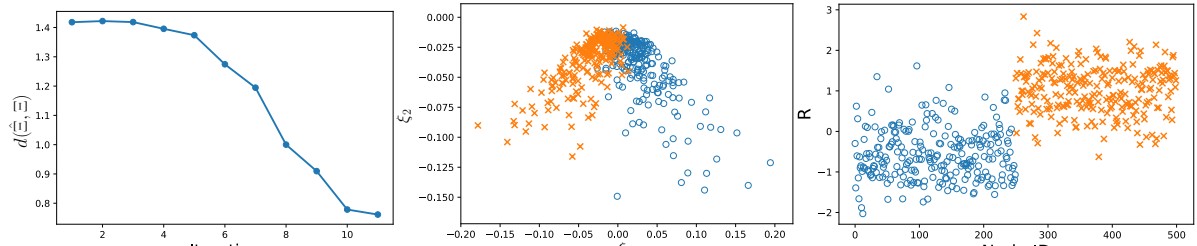

Figure 2: A simulation example. $(m, K, n) = (3, 2, 500)$, $\mathcal{P}(k, \ell, s) = 0.3 + 0.3 \cdot 1\{k = \ell = s\}$, and $\theta_i$'s are *iid* drawn from $1/[7 \times \text{Unif}(0, 1) + 1]/\sqrt{5}$. Left: Evolution of $d(\hat{\Xi}, \Xi)$ in reg-HOOI, where the initialization is HOSVD. Middle: Plot of rows of $\hat{\Xi} \in \mathbb{R}^{n \times 2}$. Right: Plot of rows of $\hat{R} \in \mathbb{R}$ versus node index. The colors correspond to true communities.

matrix from reg-HOOI and let $T > 0$ be a threshold. Define $\hat{R} \in \mathbb{R}^{n \times (K-1)}$ by

$$\hat{R}(i, :) = \hat{R}^*(i, :) \frac{\min\{T, \|\hat{R}^*(i, :)\|\}}{\|\hat{R}^*(i, :)\|}, \qquad \text{where} \quad \hat{R}^*(i, k) = \frac{\hat{\xi}_{k+1}(i)}{\hat{\xi}_1(i)}, 1 \leq k \leq K - 1. \qquad (3.8)$$

We are ready to formally introduce Tensor-SCORE:

*Tensor-SCORE* (Input: adjacency tensor $\mathcal{A}$, number of communities $K$, an initial factor matrix $\hat{\Xi}^{init}$, and tuning parameters $\delta \in (0, 1)$ and $T > 0$. Output: community labels):

1. Reg-HOOI: Initialize $\hat{\Xi}^{(0)} = \hat{\Xi}^{init}$. For $t = 1, 2, \cdots$:

   - (Regularization). Obtain $\hat{\Xi}_*^{(t-1)} = \mathcal{P}_{\mathbb{O}_\delta^{n \times K}}(\hat{\Xi}^{(t-1)})$ using (3.7).

   - (HOOI). Conduct SVD on $\mathcal{M}_1\big(\mathcal{A} \times_2 (\hat{\Xi}_*^{(t-1)})^\top \times_3 \cdots \times_m (\hat{\Xi}_*^{(t-1)})^\top\big)$. Let $\hat{\Xi}^{(t)}$ be the matrix consisting of the first $K$ left singular vectors.

   Let $\hat{\Xi} = [\hat{\xi}_1, \ldots, \hat{\xi}_K]$ denote the output.

2. SCORE: Obtain the matrix $\hat{R} \in \mathbb{R}^{n \times (K-1)}$ as in (3.8)

3. Clustering: Apply *k*-means clustering on rows of $\hat{R}$ to get community labels.

The method has two tuning parameters $\delta$ and $T$. Let $L_i = \sum_{1 \leq i_2, \ldots, i_m \leq n} \mathcal{A}(i, i_2, \ldots, i_m)$ be the degree of node $i$, for $1 \leq i \leq n$. We set $(\delta, T)$ as follows:

$$\delta = 2\sqrt{K} \cdot \big(\max_{1 \leq i \leq n} L_i\big) / \big(\sum_{i=1}^n L_i^2\big)^{1/2}, \qquad T = \sqrt{\log(n)}. \qquad (3.9)$$

The method also requires an initial estimate $\hat{\Xi}^{init}$ of the factor matrix. By our theory in Section 4, this initial estimate only needs to satisfy a mild condition, that is $d(\hat{\Xi}^{init}, \Xi) \leq \epsilon_0$, for a constant $\epsilon_0 \in (0, 1/4)$. Below, we propose two initializations:

- Initialization 1: Diagonal-removed HOSVD. Obtain $G = \mathcal{M}_1(\mathcal{A})[\mathcal{M}_1(A)]' \in \mathbb{R}^{n \times n}$, where $\mathcal{M}_1(\mathcal{A})$ is the mode-1 matricization of $\mathcal{A}$. Let $\hat{\Xi}^{init}$ be the matrix consisting of the first $K$ left singular vectors of $[G - \text{diag}(G)]$.

- Initialization 2: Randomized graph projection. Given $\epsilon \in (0,1)$, generate a random vector $\eta \in \mathbb{R}^n$ by $\eta_i \stackrel{iid}{\sim} \text{Unif}(1-\epsilon, 1+\epsilon)$. Let $\hat{\tilde{\Xi}}^{init}$ be the matrix consisting of the first $K$ eigenvectors of $(\mathcal{A} \times_3 \eta^\top \times_4 \cdots \times_m \eta^\top) \in \mathbb{R}^{n \times n}$.

Initialization 1 is a modification of HOSVD. In fact, if we conduct SVD on $G$ instead of $[G - \text{diag}(G)]$, it reduces to the standard HOSVD. The purpose of removing the diagonal of $G$ is to improve the performance on sparse networks. Initialization 2 can be viewed as a generalization of the graph projection to be discussed in Section 3.4. In fact, when $\epsilon = 0$ (so that $\eta_n = 1_n$), it reduces to the standard graph projection. The purpose of randomly perturbing $1_n$ in a local neighborhood is to avoid potential information loss. We analyze both initializations in Section 4.2.

Figure 2 illustrates the use of Tensor-SCORE. The left panel shows that reg-HOOI reliably estimates the principal subspace. In the middle panel, we plot the rows of $\hat{\Xi}$. Due to severe degree heterogeneity, a direct application of $k$-means clustering yields unsatisfactory results. In the right panel, we plot the rows of $\hat{R}$ (since $K = 2$, $\hat{R} \in \mathbb{R}^n$ is a vector; for better visualization, we add an x-axis which is node index). The two communities are well separated in this plot, suggesting that the SCORE normalization successfully removes the effects of nuisance degree parameters.

**Remark** (*Complexity*). The main computational cost of Tensor-SCORE comes from the reg-HOOI algorithm. The per-iteration complexity of reg-HOOI is $O(mKn^m + n^2 K^{m-1})$, where the two terms are contributed by computing the tensor multiplication and the SVD. In Section 4.3, we show that reg-HOOI has linear convergence, so that the number of iterations needed is only a logarithmic function of the required accuracy. The complexity of the two initializations is $O(n^{m+1})$ and $O(n^{m+1} + n^3)$, respectively. As a result, when $(m, K)$ are bounded, Tensor-SCORE is a polynomial-time algorithm.

**Remark** (*The case of unknown $K$*). Our method assumes the number of communities is given. When $K$ is unknown, we can estimate it from data. Fixing an integer $r$ that is an upper bound of $K$, we run reg-HOOI with $K = r$ to obtain $\tilde{\Xi} \in \mathbb{R}^{n \times r}$. Let $\tilde{\sigma}_1 > \tilde{\sigma}_2 > \cdots > \tilde{\sigma}_r$ be the first $r$ singular values of $\mathcal{M}_1(\mathcal{A} \times_3 \tilde{\Xi}^\top \times_4 \cdots \times_m \tilde{\Xi}^\top)$. Let

$$\hat{K} = \max\{2 \le k \le r : \tilde{\sigma}_k / \tilde{\sigma}_{k-1} \le \log(\log(n))\}. \tag{3.10}$$

In the rare case that $\tilde{\sigma}_k / \tilde{\sigma}_{k-1} > \log(\log(n))$ for all $2 \le k \le r$, we simply let $\hat{K} = r$.

## 3.4 Comparison with projection-to-graph

We compare our tensor decomposition approach with the approach of projecting a hypergraph to a graph. Given the adjacency tensor $\mathcal{A}$ of an $m$-uniform hypergraph $H = (V, E)$, define a weighted graph $G = (V, \tilde{E})$, where its adjacency matrix $\tilde{A}$ is such that

$$\tilde{A}(i, j) = \sum_{\substack{1 \le i_3, i_4, \ldots, i_m \le n \\ (i, j, i_3, \ldots, i_m) \text{ are distinct}}} \mathcal{A}(i, j, i_3, \ldots, i_m), \qquad 1 \le i \ne j \le n.$$

In this projected graph, the weighted edge between two nodes equals to the total count of hyperedges involving both nodes.

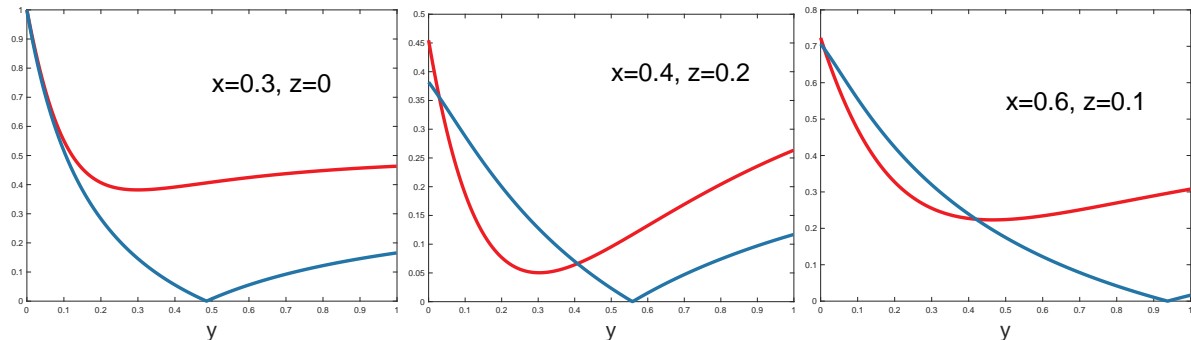

Figure 3: The information loss caused by projecting a hypergraph to a graph (setting: the 3-uniform hDCBM with 2 communities, parametrized by $(x, y, z)$). In all three plots, we fix $(x, z)$, let $y$ range in $[0, 1]$, and plot $IF_h$ (red line) and $IF_g$ (blue line). When the blue line hits zero, it indicates unwanted information loss

In hDCBM, $\mathcal{Q}$ is the "signal" part of $\mathcal{A}$ (see Section 3.1). We can similarly define the "signal" part of $\widetilde{A}$ by $\Omega = \mathcal{Q} \times_3 1_n^\top \times_4 \cdots \times_m 1_n^\top \in \mathbb{R}^{n \times n}$. Let $s_{\max}(\cdot)$ and $s_{\min}(\cdot)$ denote the maximum and minimum singular value of a matrix, respectively. Introduce two quantities

$$IF_h = \frac{s_{\min}(\mathcal{M}_1(\mathcal{Q}))}{s_{\max}(\mathcal{M}_1(\mathcal{Q}))}, \qquad IF_g = \frac{s_{\min}(\Omega)}{s_{\max}(\Omega)}.$$

They are both properly scaled. When $IF = 0$, the minimum singular value of the "signal" matrix is zero, indicating unwanted "information loss." In this case, some leading empirical eigenvectors are extremely noisy, and spectral methods will yield unsatisfactory performance. Therefore, the larger $IF$, the better.

Our observation is that, projection-to-graph has the risk of dragging $IF_g$ to zero. We illustrate it with an example.

**Example** (*3-uniform hDCBM with 2 communities*): Consider an hDCBM with $m = 3$ and $K = 2$. Fixing $x, y, z \in [0, 1]$, the core tensor $\mathcal{P}$ is such that $\mathcal{P}(1, 1, 1) = \mathcal{P}(2, 2, 2) = x$, $\mathcal{P}(1, 2, 2) = y$ and $\mathcal{P}(1, 1, 2) = z$, where other entries follow by symmetry. Let $V_1$ and $V_2$ be the two communities. The degree parameters satisfy that $\sum_{i \in V_1} \theta_i^2 = \sum_{i \in V_2} \theta_i^2$.

In Figure 3, we plot $IF_h$ and $IF_g$ for a variety of $(x, y, z)$. It is seen that $IF_h$ never hits zero, but $IF_g$ can be zero or get close to zero. By elementary linear algebra, $IF_g$ becomes zero if and only if $(x + y)(x + z) - (y + z)^2 = 0$. There are infinitely many such solutions. Hence, the projection-to-graph approach has the risk of losing information. The success of methods based on projection-to-graph typically requires additional conditions (Zhou et al., 2007; Ghoshdastidar and Dukkipati, 2017; Chien et al., 2018; Kim et al., 2018), such as no degree heterogeneity and symmetry conditions on $\mathcal{P}$ (e.g., $y = z$ in this example). However, such conditions are hard to check in practice. It is thus valuable to develop community detection methods that operate on the hypergraph directly.

**Remark** (*The induced graph by HOSVD*). HOSVD is equivalent to conducting PCA on the matrix $A^* = \mathcal{M}_1(\mathcal{A})[\mathcal{M}_1(\mathcal{A})']$. Viewing $A^*$ as an adjacency matrix, it gives rise to an induced

(weighted) graph $G^*$. It is interesting to compare $G^*$ with the graph $G$ above. By definition,

$$A^*(i, j) = \sum_{\substack{1 \leq i_2, \ldots, i_m \leq n, \\ (i_2, \ldots, i_m) \text{ are distinct} \\ \{i,j\} \cap \{i_2, \ldots, i_m\} = \emptyset}} \mathcal{A}(i, i_2, \ldots, i_m) \mathcal{A}(j, i_2, \ldots, i_m).$$

We define a "hyper-triangle" as $i$-$S$-$j$, where $S$ is a subset containing $(m-1)$ indices such that $\{i\} \cup S$ and $\{j\} \cup S$ are both hyper-edges (for $m = 2$, it reduces to the usual definition of a triangle in a graph). Then, in $G^*$, the weighted edge between two nodes equals to the count of "hyper-triangles" that link two nodes. This graph is different from $G$.

### 3.5 Extension to non-uniform hypergraphs

So far, we have been focusing on uniform hypergraphs. We now discuss two extensions of Tensor-SCORE to non-uniform hypergraphs. Let $V = \{1, 2, \ldots, n\}$ be the set of nodes and suppose $V = V_1 \cup \ldots \cup V_K$ is a partition to true communities. Let $H = (V, E)$ be a non-uniform hypergraph, where the size of hyperedges ranges from 2 to $m_0$.

In the first extension, we represent $H$ by a collection of uniform hypergraphs $H^{(m)} = (V, E^{(m)})$, where $E^{(m)}$ is the set of order-$m$ hyperedges, $2 \leq m \leq m_0$. We apply the first two steps of Tensor-SCORE to obtain $\hat{R}^{(m)} \in \mathbb{R}^{n \times (K-1)}$ for each uniform hypergraph. Then, we apply $k$-means clustering to rows of the matrix $\hat{R} = [\hat{R}^{(2)}, \hat{R}^{(3)}, \ldots, \hat{R}^{(m_0)}]$.

The rationale behind this approach is to impose an hDCBM for each $H^{(m)}$, where these models share the same membership matrix $\Pi$ but have their individual $\theta^{(m)} \in \mathbb{R}_+^n$ and $\mathcal{P}^{(m)} \in S^m(\mathbb{R}^K)$. Under this model, we can apply $k$-means to each $\hat{R}^{(m)}$, but stacking them together combines information in hyperedges of different sizes. This idea has the same spirit as the extension of SCORE to directed graphs (Ji and Jin, 2017).

In the second extension, we introduce a set of dummy nodes indexed by $-3$ to $-m_0$. We create an $m_0$-uniform hypergraph with nodes $V_* = \{-m_0, \ldots, -3, 1, 2, \ldots, n\}$. For any hyperedge $\{i_1, \ldots, i_m\}$ with $m < m_0$, we fill in the dummy node to make it a size-$m_0$ hyperedge $\{i_1, \ldots, i_m, -(m+1), \ldots, -m_0\}$. Then, we apply the first two steps of Tensor-SCORE to obtain $\hat{R}^{(n+m_0-2) \times K}$. In the last step, we throw away the rows of $\hat{R}$ associated with the dummy nodes and apply $k$-means clustering to the remaining rows.

The rationale behind this approach is to start from an hDCBM for order-$m_0$ hyperedges and induce a model for lower-order hyperedges. Consider an hDCBM for $m_0$-uniform hypergraphs, with the membership matrix $\Pi$ and parameters $\theta \in \mathbb{R}_+^n$ and $\mathcal{P} \in S^{m_0}(\mathbb{R}^K)$. We now add the dummy nodes $-(m+1), \ldots, -m_0$ with degree parameters $\omega_{-(m+1)}, \ldots, \omega_{-m_0}$ and let their membership vectors be $\pi_{-(m+1)} = \ldots = \pi_{-m_0} = (\frac{1}{K}, \ldots, \frac{1}{K})$. Then, for a lower-order hyperedge $\{i_1, \ldots, i_m\}$ such that $i_j \in V_{k_j}$, by adding the dummy node to make it an order-$m_0$ hyper-edge, we obtain an induced model

$$\mathbb{P}(\mathcal{A}(i_1, \ldots, i_m, -(m+1), \ldots, -m_0) = 1) = (\omega_{-(m+1)} \cdots \omega_{-m_0})^{-1} \cdot \mathcal{P}_*^{(m)}(k_1, \ldots, k_m) \cdot \theta_{i_1} \cdots \theta_{i_m},$$

where $\mathcal{P}_*^{(m)} = \frac{1}{K^{m_0-m}}[\mathcal{P} \times_{(m+1)} 1_K \times_{(m+2)} \cdots \times_{m_0} 1_K] \in S^m(\mathbb{R}^K)$ is a "projection" of $\mathcal{P} \in S^{m_0}(\mathbb{R}^K)$. In this induced model, the $\omega$'s serve as an inflation factor for lower-order edges: If

$\omega_{-(m+1)} \cdots \omega_{-m_0} < 1$, then lower-order edges have higher probabilities.

## 4  Statistical properties

Let $H = (V, E)$ be an $m$-uniform hypergraph that satisfies hDCBM model in Section 3.1. The hDCBM is not identifiable: For any positive diagonal matrix $\tilde{D} \in \mathbb{R}^{K \times K}$, if we change $(\theta, \mathcal{P})$ to $(\tilde{D}^{-1}\theta, \mathcal{P} \times_1 \tilde{D} \times_2 \cdots \times_m \tilde{D})$, the distribution of $\mathcal{A}$ remains unchanged. To ensure identifiability, we assume $\mathcal{P}(k, k, k) = 1$, for $1 \leq k \leq K$. As a result, the network sparsity is captured by the degree parameters $\theta_i$.

Let $V = V_1 \cup \ldots \cup V_K$ be the partition of nodes into communities. We assume

$$\max_k |V_k| \leq C \min_k |V_k|, \qquad \max_{1 \leq k \leq K} \sum_{i \in V_k} \theta_i^2 \leq C \min_{1 \leq k \leq K} \sum_{i \in V_k} \theta_i^2. \tag{4.1}$$

Introduce a diagonal matrix $D = \mathrm{diag}(d_1, d_2, \ldots, d_K)$, where $d_k = \|\theta\|^{-1}(\sum_{i \in V_k} \theta_i^2)^{1/2}$, for $1 \leq k \leq K$. Define $G = \mathcal{M}_1(\mathcal{P} \times_1 D \times_2 \cdots \times_m D) \in \mathbb{R}^{K \times K^{m-1}}$. Let $\kappa_1, \ldots, \kappa_K$ be the singular values of $G$, arranged in the descending order. Let $u_1, \ldots, u_K \in \mathbb{R}^K$ be the associated left singular vectors. As $n \to \infty$, for a positive sequence $\gamma_n$, we assume

$$\|\mathcal{P}\|_{\max} \leq C, \qquad \min\{\kappa_1 - \kappa_2, \ \kappa_K\} \geq \gamma_n. \tag{4.2}$$

We also assume

$$u_1 \text{ is a positive vector satisfying } \max_{1 \leq k \leq K} u_1(k) \leq C \min_{1 \leq k \leq K} u_1(k). \tag{4.3}$$

Let $\theta_{\max}$ and $\theta_{\min}$ be the maximum and minimum of $\theta_i$'s. Define

$$err_n = \frac{\sqrt{K^{m-2}\theta_{\max}^{m-1}\|\theta\|_1 \log(n)} + (\sqrt{K}\theta_{\max}/\|\theta\|)^{m-1}\log(n)}{\sqrt{n}\theta_{\min}\|\theta\|^{m-1}} + \frac{K\theta_{\max}^2}{\sqrt{n}\theta_{\min}\|\theta\|}.$$

We assume

$$K\gamma_n^{-2}err_n^2 \log(n) \to 0. \tag{4.4}$$

These conditions are mild. Condition (4.1) requires that that the degree distributions are balanced across communities. Condition (4.2) is mild, since the matrix $G$ is properly scaled. Condition (4.3) is satisfied when all entries of $\mathcal{P}$ are lower bounded by a constant and $\max_k d_k \asymp \min_k d_k$. Regarding Condition (4.4), when $K = O(1)$, $\theta_{\max} \asymp \theta_{\min}$ and $\gamma_n \asymp 1$, it becomes $n^{m-1}\bar{\theta}^m \gg \log^2(n)$, where $\bar{\theta}$ is the average of $\theta_i$. Since $n^{m-1}\bar{\theta}^m$ is the order of average node degree, our theory indeed covers a wide range of sparsity.

Tensor-SCORE plugs in an initial estimate $\hat{\Xi}^{init}$. We impose a condition on $\hat{\Xi}^{init}$. Recall that $\mathcal{Q}$ denotes the "signal" tensor in hDCBM, and $\Xi \in \mathbb{R}^{n \times K}$ is the matrix of the first $K$ left singular vectors of $\mathcal{M}_1(\mathcal{Q})$.

**Condition 1** (Initialization). $\|\hat{\Xi}^{init}(\hat{\Xi}^{init})^\top - \Xi\Xi^\top\| \leq \epsilon_0$, *for a constant* $\epsilon_0 \in (0, 1/4)$.

This is a mild condition, which basically says that the column space of $\hat{\Xi}^{init}$ cannot be orthogonal

to the column space of $\Xi$. In Section 4.2, we study when the two initializations in Section 3.3 satisfy this requirement.

## 4.1 Rate of convergence of Tensor-SCORE

Given the community partition by Tensor-SCORE, let $\hat{\Pi} = [\hat{\pi}_1, \ldots, \hat{\pi}_n]'$, where $\hat{\pi}_i = e_k$, if node $i$ is clustered to community $k$. We measure the performance by the Hamming error of clustering:

$$\mathcal{L}(\hat{\Pi}, \Pi) = \min_{\tau:\text{a permutation on } 1,2,\ldots,K} \Big( \sum_{i=1}^{n} 1\{i \in \hat{V}_k, i \notin V_{\tau(k)}\} \Big). \tag{4.5}$$

Let $\theta_{\max}, \theta_{\min}$, and $\bar{\theta}$ denote the maximum, minimum, and average of $\theta_i$'s. The following theorem is proved in the appendix:

**Theorem 1** (Rate of convergence of Tensor-SCORE). *Fix $m \geq 2$ and consider a sequence of hDCBM indexed by $n$, where $(K, \Pi, \theta, \mathcal{P})$ are allowed to depend on $n$. As $n \to \infty$, suppose (4.1)-(4.4) are satisfied. We apply Tensor-SCORE for community detection, where the initial estimate of the factor matrix satisfies Condition 1. We set $T = \sqrt{K \log(n)}$ and $\delta = C_0 \sqrt{K} \theta_{\max}/\|\theta\|$ for a properly large constant $C_0 > 0$. In the reg-HOOI step, we run $t_{\max}$ iterations, with $t_{\max} \geq m \log(n\theta_{\max}/\|\theta\|)$. With probability $1 - n^{-2}$, the estimate $\hat{\Pi}$ from Tensor-SCORE satisfies that*

$$\mathcal{L}(\hat{\Pi}, \Pi) \leq C n \gamma_n^{-2} err_n^2.$$

*Furthermore, if $\theta_{\max} \asymp \theta_{\min}$ and $K = o(\gamma_n \sqrt{n})$, then with probability $1 - n^{-2}$,*

$$\mathcal{L}(\hat{\Pi}, \Pi) = n \cdot O\Big( \frac{K^{m-2} \log(n)}{\gamma_n^2 n^{m-1} \bar{\theta}^m} \Big).$$

We make a few remarks.

1. *Connection to average node degree.* It is common to express the error rate of community detection in terms of average node degree. For each node $i$, define its degree $\mathcal{D}_i$ as the total number of hyperedges that involves this node. Let $d^* = n^{-1} \sum_{i=1}^{n} \mathbb{E}[\mathcal{D}_i]$ be the expected average degree. When $\theta_{\max} \asymp \theta_{\min}$, we have $d^* \asymp n^{m-1} \bar{\theta}^m$. Theorem 1 and condition (4.4) translate to:

$$\mathcal{L}(\hat{\Pi}, \Pi) = n \cdot O\Big( \frac{K^{m-2} \log(n)}{\gamma_n^2 d^*} \Big), \qquad \text{provided that} \quad d^* \gg K^{m-1} \gamma_n^{-2} \log^2(n).$$

   When $K = O(1)$ and $\gamma_n \asymp 1$, it only requires that $d^* \gg \log^2(n)$. Hence, our method covers a wide range of sparsity. Moreover, the clustering error is $\mathcal{L}(\hat{\Pi}, \Pi) \ll n \cdot \frac{1}{\log(n)} = o(n)$, and it yields consistent community detection.

2. *Implication on graph networks.* When $m = 2$, hDCBM reduces to the DCBM (Karrer and Newman (2011)) for regular graph networks. When $\theta_{\max} \asymp \theta_{\min}$, Theorem 1 implies

$$\mathcal{L}(\hat{\Pi}, \Pi) = n \cdot O((\gamma_n^2 n \bar{\theta}^2)^{-1} \log(n)).$$

   This has recovered the rate of SCORE for graph community detection (Jin, 2015).

3. *The impact of the diagonal tensor.* The expression of $err_n$ has two terms, where the first term comes from an upper bound of $\|\mathcal{A} - \mathbb{E}\mathcal{A}\|_\delta$ and the second term is from the upper bound of $\|\mathrm{diag}(\mathcal{Q})\|$. If there is no degree heterogeneity, the second term disappears, since substracting $\mathrm{diag}(\mathcal{Q})$ from $\mathcal{Q}$ does not change the eigenvectors (see, for example, Lemma 4.1 of Ghoshdastidar and Dukkipati (2017)). However, when there exists degree heterogeneity, the second term cannot be avoided. Fortunately, this term is often negligible, provided that $\theta_{\max}/\bar{\theta}$ is not too large. For example, when $\theta_{\max} \asymp \theta_{\min}$ and $K = o(\gamma_n \sqrt{n})$, the contribution of this term to $\mathcal{L}(\hat{\Pi}, \Pi)$ is $o(1)$, which is basically zero (since clustering error only takes integer values).

The symmetric hSBM model has been considered in a number of works (Ghoshdastidar and Dukkipati, 2014, 2015a, 2017; Chien et al., 2018; Kim et al., 2018). It is a special case of hDCBM that has no degree heterogeneity and satisfies particular symmetry constraints, and Theorem 1 is directly applicable.

**Corollary 1** (The example of symmetric hSBM)**.** Fix $m \geq 2$ and consider an hDCBM where $\theta_i \equiv \alpha_n^{1/m}$ for $\alpha_n \to 0$, $|V_k| \in [K^{-1}n - 1, \, K^{-1}n]$ for $1 \leq k \leq K$, and the core tensor satisfies $\mathcal{P}(s_1, s_2, \ldots, s_K) = 1$ when $s_k$'s are equal and $\mathcal{P}(s_1, s_2, \ldots, s_K) = b$ otherwise, for $b \in (0, 1)$. Suppose $K = o(\sqrt{n})$ and $(1 - b)^2 n^{m-1} \alpha_n \gg K^{m-1} \log^2(n)$. With probability $1 - n^{-2}$,

$$\mathcal{L}(\hat{\Pi}, \Pi) = n \cdot O\Big( \frac{K^{2m-2} \log(n)}{(1 - b)^2 n^{m-1} \alpha_n} \Big).$$

We compare it with existing results for symmetric hSBM. Ghoshdastidar and Dukkipati (2014) considered an HOSVD approach. With $K$ and $\alpha_n$ being fixed, they derived that $\mathcal{L}(\hat{\Pi}, \Pi) = n \cdot \widetilde{O}((1 - b)^{-4} n^{-[m-2+\frac{1}{2m}]})$. The rate has a sub-optimal dependence on $n$ and $(1 - b)$; moreover, since $\alpha_n$ is fixed, it only applies to dense networks. Ghoshdastidar and Dukkipati (2017) studied an approach by projecting the hypergraph to a weighted graph. The rate they obtained (Ghoshdastidar and Dukkipati, 2017, Corollary 5.1) is the same as ours. In comparison, our method is applicable in broader settings such as those with degree heterogeneity and/or information loss.

## 4.2 Performance of two initializations

In Section 3.3, we propose two methods as initializations of Tensor-SCORE. We show that they satisfy Condition 1 under mild conditions.

First, we consider an initialization by the diagonal removed HOSVD.

**Theorem 2** (Initialization by diagonal removed HOSVD)**.** *Fix $m \geq 2$ and consider an hDCBM where* (4.1)*-*(4.3) *hold. We apply the diagonal removed HOSVD to the adjacency tensor $\mathcal{A}$ to obtain $\hat{\Xi}$. Suppose*

$$\min\Big\{ \frac{\sqrt{\theta_{\max}^m \|\theta\|_1^m \log(n)} + \log(n)}{\gamma_n^2 \|\theta\|^{2m}}, \, \frac{\theta_{\max}^2}{\gamma_n^2 \|\theta\|^2} \Big\} \to 0. \tag{4.6}$$

*With probability $1 - n^{-2}$, Condition 1 is satisfied for any constant $\epsilon_0 > 0$. For the case of $\theta_{\max} \asymp \theta_{\min}$, the requirement* (4.6) *reduces to $\gamma_n^4 n^m \bar{\theta}^{2m} \gg \log(n)$.*

We have a few observations:

1. *The required network sparsity.* Same as before, let $d^*$ be the expected average node degree. Note that $d^* \asymp n^{m-1}\bar{\theta}^m$, when $\theta_{\max} \asymp \theta_{\min}$. As a result, the requirement in Theorem 2 becomes $d^* \gg \gamma_n^{-2}\sqrt{n^{m-2}\log(n)}$. Therefore, this initialization works for networks that are moderately dense.

2. *Comparison with standard HOSVD.* If we use the standard HOSVD as initialization, Condition 1 is satisfied if $\gamma_n^2 n\bar{\theta}^m \gg \log(n)$ (Lemma A.1). This requirement translates to $d^* \gg \gamma_n^{-2}n^{m-2}\log(n)$, which is much more stringent than the requirement of diagonal removed HOSVD.

Next, we consider an initialization by randomized graph projection. Introduce $\mathcal{P}_* = \mathcal{P} \times_1 D \times_2 \cdots \times_m D$, where $D$ is the same diagonal matrix we have used to state Condition (4.2). Define $\widetilde{\lambda}_{\min}(\mathcal{P}_*, v) = \lambda_{\min}(\mathcal{P}_* \times_3 v^\top \times_4 \cdots \times_m v^\top)/\|v\|^{m-2}$, for $v \in \mathbb{R}^K$ such that $\|v\| \neq 0$. The following proposition is proved in the appendix, where $s_{\min}(\cdot)$ denotes the minimum singular value of a matrix:

**Proposition 4.1.** *For any $v \in \mathbb{R}^K$, $\widetilde{\lambda}_{\min}(\mathcal{P}_*, v) \leq s_{\min}(\mathcal{M}_1(\mathcal{P}_*))$.*

Let $\mathcal{S}_K(\epsilon) = \{D^{-1}\Pi^\top\Theta\eta : 1-\epsilon \leq \eta_i \leq 1+\epsilon, 1 \leq i \leq n\}$, where $\epsilon \in (0,1)$ is the tuning parameter used in the randomized graph projection. As $n \to \infty$, for a positive sequence $\widetilde{\gamma}_n \to 0$, we assume

$$\max_{v \in \mathcal{S}_K(\epsilon)} \{\widetilde{\lambda}_{\min}(\mathcal{P}_*, v)\} \geq \widetilde{\gamma}_n. \tag{4.7}$$

We recall that $\kappa_K$ is the minimum singular value of $\mathcal{M}_1(\mathcal{P}_*)$. Condition (4.2) ensures that $\kappa_K \geq \gamma_n$. It follows from Proposition 4.1 that $\widetilde{\gamma}_n$ can never be larger than $\gamma_n$.

**Theorem 3** (Initialization by randomized graph projection)**.** *Fix $m \geq 2$ and consider an hDCBM where (4.1)-(4.3) hold. Additionally, assume (4.7) holds with $S_K(\varepsilon)$ so that $\varepsilon \leq (\widetilde{\gamma}_n/4m\|\mathcal{M}_1(\mathcal{P}_*)\|)(1-\varepsilon)^{m-2}/(1+\varepsilon)^{m-3}$. We use the randomized graph projection to obtain $\widehat{\Xi}$. Suppose*

$$\left(\frac{1+\varepsilon}{1-\varepsilon}\right)^m \cdot \min\left\{ \frac{\sqrt{\theta_{\max}\|\theta\|_1^{m-1}\log(n)} + \log(n)}{\widetilde{\gamma}_n\|\theta\|^2\|\theta\|_1^{m-2}}, \quad \frac{\theta_{\max}^2}{\widetilde{\gamma}_n\|\theta\|^2} \right\} \to 0. \tag{4.8}$$

*With probability $1 - n^{-2}$, Condition 1 is satisfied for any constant $\epsilon_0 > 0$. For the case of $\theta_{\max} \asymp \theta_{\min}$, the requirement (4.6) reduces to $\widetilde{\gamma}_n^2 n^{m-1}\bar{\theta}^m \gg \log(n)$.*

We have a few observations:

1. *The required network sparsity.* Letting $d^*$ be the expected average node degree, when $\theta_{\max} \asymp \theta_{\min}$, the requirement in Theorem 3 becomes $d^* \gg \widetilde{\gamma}_n^{-2}\log(n)$. If $\widetilde{\gamma}_n \geq CK^{m-1}\gamma_n$, then this requirement is weaker than the requirement of network sparsity for our main algorithm (see the first remark behind Theorem 1). In other words, this initialization works for the whole range of sparsity of interest.

2. *Comparison with standard graph projection.* The standard approach to projecting a hypergraph to a weighted graph (see Section 3.4) corresponds to $\epsilon = 0$ and $\eta$ being fixed as

$\mathbf{1}_n$. In the condition (4.7), the left hand side increases with $\epsilon$. This means, by exploring a lot of directions $\eta$ when projecting the hypergraph, we can increase $\widetilde{\gamma}_n$, so as to increase the range of sparsity that this initialization works. In particular, for examples in Section 3.4 where $IF_g = 0$ for $\eta = \mathbf{1}_n$, the randomization on $\eta$ allows us to escape from the zero-eigenvalue situation and avoid unwanted information loss.

To summarize, for sufficiently dense networks (e.g., the average degree is $\gg \gamma_n^{-2}\sqrt{n^{m-2}}$), we recommend using the diagonal removed HOSVD as initialization, for it avoids the issue of potential information loss. For sparser networks, we recommend using the randomized graph projection as initialization.

## 4.3 Spectral decomposition by reg-HOOI

A key component of tensor-SCORE is estimating the column space of $\Xi$ by a tensor power iteration algorithm. We now investigate the performance of reg-HOOI, which results are of independent interest. To make the theory applicable to broader settings, we drop Conditions (4.1)-(4.4); instead, we let the results depend on the unbalanceness of communities and orders of singular values explicitly. Define $\beta(\mathcal{Q}) = \max_{i_1, \cdots, i_{m-1}} \sum_{i_m=1}^n \mathcal{Q}(i_1, \cdots, i_m)$. This quantity governs the error bounds in theorems below. It is seen that $\beta(\mathcal{Q}) \leq \|\mathcal{P}\|_{\max} \cdot \theta_{\max}^{m-1}\|\theta\|_1$, but this upper bound is often loose; hence, we prefer to express results in terms of $\beta(\mathcal{Q})$. Same as before, we write $d_k = \|\theta\|^{-1}(\sum_{i \in V_k} \theta_i^2)^{1/2}$, $1 \leq k \leq K$, and let $\kappa_1, \ldots, \kappa_K$ denote the singular values of the matrix $G = \mathcal{M}_1(\mathcal{P} \times_1 D \times_2 \cdots \times_m D)$. The following theorem establishes the contraction property in the power iteration:

**Theorem 4** (Linear convergence of reg-HOOI). *Consider an hDCBM, where we apply the reg-HOOI algorithm to $\mathcal{A}$, with an initialization that satisfies Condition (1), for $\varepsilon_0 \in (0, 1/4)$. Let $\hat{\Xi}^{(t)}$ be the estimation at iteration $t$, for $t \geq 1$. Suppose*

$$(1-3\varepsilon_0)^{\frac{m-1}{2}}\kappa_K\|\theta\|^m \geq C_1(m!)(4K)^{\frac{m}{2}}\max\left\{\sqrt{mn\delta^2\beta(\mathcal{Q})}, \, (m\log n)^{\frac{m+3}{2}}\delta^{m-2}\sqrt{\|\mathcal{P}\|_{\max}\theta_{\max}^2\|\theta\|_1^{m-2}n\log n}\right\},$$

*for some absolute constants $C_1, C_2 > 0$. Write*

$$\zeta_n(\mathcal{Q}) = 2^m \frac{\sqrt{(m!)K^{m-2}\beta(\mathcal{Q})\log n} + (m!)\delta^{m-1}\log n + (m!)\kappa_1\theta_{\max}^2\|\theta\|^{m-2}/d_{\min}^2}{\kappa_K\|\theta\|^m}.$$

*Then, with probability at least $1 - n^{-2}$ for all $1 \leq t \leq T - 1$,*

$$\left\|\hat{\Xi}^{(t+1)}(\hat{\Xi}^{(t+1)})^\top - \Xi\Xi^\top\right\| \leq \frac{1}{2}\left\|\hat{\Xi}^{(t)}(\hat{\Xi}^{(t)})^\top - \Xi\Xi^\top\right\| + C_3\zeta_n(\mathcal{Q}),$$

*for some absolute constant $C_3 > 0$. Therefore, after at most $t_0 = O\big(1 \vee m\log(n\|\theta\|/\theta_{\max})\big)$ iterations, with probability at least $1 - n^{-2}$,*

$$\left\|\hat{\Xi}^{(t_0)}(\hat{\Xi}^{(t_0)})^\top - \Xi\Xi^\top\right\| \leq 2C_3\zeta_n(\mathcal{Q}).$$

Theorem 4 essentially applies to any random Bernoulli tensor whose expectation has a low rank. Unlike the study of random Gaussian tensors (e.g., Zhang and Xia (2018)), the study of

Bernoulli tensors requires technical efforts to deal with the extreme sparsity. A typical step in the analysis of power iteration algorithms is to derive a sharp concentration inequality for the tensor operator norm of $(\mathcal{A} - \mathbb{E}\mathcal{A})$. Unfortunately, for Bernoulli tensors, even the best bound for $\|\mathcal{A} - \mathbb{E}\mathcal{A}\|$ is too large to yield the desired result in Theorem 4. Alternatively, we borrow the notion of incoherent tensor operator norm, which was initially introduced by Yuan and Zhang (2017). Our design the reg-HOOI algorithm guarantees that it has satisfying performance as long as the incoherent operator norm of $(\mathcal{A} - \mathbb{E}\mathcal{A})$ is reasonably small. Then, our main technical effort is on deriving a sharp concentration inequality for the incoherent operator norm of $(\mathcal{A} - \mathbb{E}\mathcal{A})$.

In detail, for any $\delta \in [n^{-1/2}, 1]$, let $\mathcal{U}_{j_1 j_2}(\delta) = \big\{ \mathbf{u}_1 \otimes \cdots \otimes \mathbf{u}_k : \|\mathbf{u}_j\| \leq 1, \forall j; \ \|\mathbf{u}_j\|_{\mathsf{max}} \leq \delta, j \neq j_1, j_2 \big\}$. This is the set of all rank-one tensors satisfying incoherent conditions in "directions" other than $j_1$ and $j_2$. The $\delta$-*incoherent operator norm* of a $k$-th order tensor $\mathcal{T}$ is defined by $\|\mathcal{T}\|_\delta = \sup_{\mathcal{X} \in \mathcal{U}(\delta)} \langle \mathcal{T}, \mathcal{X} \rangle$, where $\mathcal{U}(\delta) = \bigcup_{1 \leq j_1 < j_2 \leq k} \mathcal{U}_{j_1 j_2}(\delta)$. Here, $\mathcal{U}(\delta)$ is the collection of all rank-one tensors satisfying certain incoherence conditions in all but two directions. When $\delta = 1$, it reduces to the regular tensor operator norm.

**Theorem 5** (Concentration inequality of incoherent operator norm)**.** *Consider an hDCBM, where $\|\mathcal{P}\|_{\mathsf{max}} \theta_{\mathsf{max}} \|\theta\|_1^{m-1} \geq \log n$. There are absolute constants $C_1, C_2 > 0$ such that for all $t \geq (m!)2^m \cdot \max\big\{ C_1 \sqrt{m^5 n \delta^2 \beta(\mathcal{Q})} \log^2 n, \ C_2 m (m \log n)^{\frac{(m+7)}{2}} \delta^{m-2} \sqrt{\|\mathcal{P}\|_{\mathsf{max}} \theta_{\mathsf{max}}^2 \|\theta\|_1^{m-2} n \log n} \big\}$,*

$$\mathbb{P}(\|\mathcal{A} - \mathbb{E}\mathcal{A}\|_\delta \geq 3t) \leq 2n^{-2}$$
$$+ (10m^3 \log^2 n) \exp\Big( -\frac{2^{-2m-2}t^2}{8(m!)\delta^2 \beta(\mathcal{Q})m^4 \log^4 n} \Big) + (10m^3 \log^2 n) \exp\Big( -\frac{3}{16} \cdot \frac{2^{-(m+1)}t}{2(m!)\delta^{m-2}m^2 \log^2 n} \Big).$$

**Remark** *(Why it is crucial to use the incoherent norm)*: When $\delta$ is appropriately small, the incoherent norm of $(\mathcal{A} - \mathbb{E}\mathcal{A})$ can be much smaller than its canonical operator norm. For example, under the conditions of Theorem 5, if $K, m$ are bounded, $\theta_{\mathsf{max}} \asymp \theta_{\mathsf{min}}$ and $\delta \asymp n^{-1/2}$, we can show that, with high probability,

$$\|\mathbb{E}\mathcal{A}\| \lesssim \kappa_1 \|\theta\|^m, \qquad \|\mathcal{A} - \mathbb{E}\mathcal{A}\| \gtrsim 1, \qquad \|\mathcal{A} - \mathbb{E}\mathcal{A}\|_\delta \lesssim \sqrt{\|\mathcal{P}\|_{\mathsf{max}} \cdot n\theta_{\mathsf{max}}^m}.$$

In the sparse regime, it is possible that $\|\mathbb{E}\mathcal{A}\| = o(1)$. This means, if we build our analysis on the concentration of $\|\mathcal{A} - \mathbb{E}\mathcal{A}\|$, we will never be able to get any consistency results. It is thus necessary to use the incoherent norm.

# 5 Experiment Results

We evaluate the performance of tensor-SCORE on a wide range of synthetic and real-world networks. For comparison, four other popular methods for community detection on hypergraphs are also considered here. As mentioned in Section 1, the most common approach is to project a hypergraph to a binary graph and then use any existing community detection method for graphs. This is our first candidate for comparison and it is referenced as "binary projection". A variant of this approach is to weight the edges in the projected graph by the counts of hyperedges, which we refer to as "weighted projection". In both situations we use the well-known method SCORE for community detection. An improvement upon the ad hoc projection

methods is developed by Ghoshdastidar and Dukkipati (2017) which constructs a normalized Laplacian matrix from a hypergraph and applies spectral clustering on the Laplacian matrix. This method is named NHCut. That paper does not deal with degree heterogeneity, but the authors added an eigenvector normalization step to improve its performance. Therefore, they were implicitly using the idea of SCORE. The last candidate is HOSVD which applies higher-order SVD to adjacency tensors and clusters the rows of the resulting singular matrices; it is the one most similar to our method in spirit. In all the experiments, we use the default values suggested above for the parameters in tensor-SCORE unless otherwise stated; there are no tuning parameters for the other methods except for the number of communities. The metric for performance is the clustering error against ground-truth communities as defined in (4.5), which is the percent of nodes assigned to wrong communities. Two initializations are used for tensor-SCORE: tensor-SCORE-1 uses NHCut and tensor-SCORE-2 uses HOSVD.

## 5.1    Synthetic Networks

We first evaluate tensor-SCORE on a variety of synthetic networks generated by the hypergraph degree-corrected block model (hDCBM) (Section 3). The hDCBM consists of $n$ nodes divided into 2 equally sized communities. The node degree parameters $\theta_i$ are drawn from a power-law distribution $\theta_i^{-\alpha}$ with smaller $\alpha$ corresponding to more heterogeneous degree distributions. Edge size is fixed at $m = 3$ meaning that the simulated networks are 3-uniform hypergraphs. (We will evaluate the methods on non-uniform real networks.) The likelihood for nodes from same or different communities to form edges are controlled by the $2 \times 2 \times 2$ core tensor $\mathcal{P}$.

In the first set of experiments, we vary degree heterogeneity parameter $\alpha$ while keeping other parameters fixed at $n = 300$, $\mathcal{P}_{111} = \mathcal{P}_{222} = 0.3$, $\mathcal{P}_{122} = 0.2$, $\mathcal{P}_{112} = 0$. Table 2(A) presents the average clustering error for each method over 20 simulations (same below). Most real networks are found with very heterogenous degree distributions ($\alpha \in [2, 3]$), rendering community detection a challenging task. But tensor-SCORE outperforms all the other methods regardless of $\alpha$, and achieves low error rate even at $\alpha = 2$. In the second set of experiments, we gradually increase the likelihood to form edges between communities ($\mathcal{P}_{122}$), making the community structure weaker (the other parameters are fixed at $n = 300$, $\mathcal{P}_{111} = \mathcal{P}_{222} = 0.3$, $\mathcal{P}_{112} = 0$, $\alpha = 2$). Table 2(B) shows the average clustering error for each method as $\mathcal{P}_{122}$ increases. In general, error rate increases with $\mathcal{P}_{122}$ for all the methods; but tensor-SCORE still outperforms the others by a big margin. As $\mathcal{P}_{122}$ approaches 0.5, the graph-based methods are subjected to information loss (recalling Fig. 3), which is clear from their high error rates, while tensor-SCORE still achieves a relatively low error rate. The last set of experiments adjust the size of the hypergraphs (i.e., number of nodes $n$). The other parameters are fixed at $\mathcal{P}_{111} = \mathcal{P}_{222} = 0.3$, $\mathcal{P}_{122} = 0.2$, $\mathcal{P}_{112} = 0$, $\alpha = 2$. and the results are shown in Table 2(C). As the graph size grows the "structure signal" also becomes stronger and the community detection results get better in general. NHCut achieves comparable and even lower error rate than tensor-SCORE when $n = 400$ and 500. It is reasonable since this is a case with strong signals and no information loss.

Next, we further demonstrate the power of tensor-SCORE on real-world hypergraph networks. In real-world data analysis, tensor-SCORE uses HOSVD as initialization to avoid po-

Table 2: Performance of the community detection methods on synthetic hypergraphs simulated from the hDCBM under different parameter settings: (A) varying degree heterogeneity parameter $\alpha$; (B) varying likelihood to form edges between communities $\mathcal{P}_{122}$; (C) varying number of nodes $n$. The numbers in columns 2 - 6 are the percent of misclassified nodes against ground truth (clustering error). Each value of clustering error is the average over 20 simulations. Tensor-SCORE-1 uses NHCut as initialization while Tensor-SCORE-2 uses HOSVD as initialization. The lowest clustering error under each parameter setting is in bold face.

|     | $\alpha$ | Binary Projection | Weighted Projection | NHCut | HOSVD | Tensor SCORE-1 | Tensor SCORE-2 |
|-----|----------|-------------------|---------------------|-------|-------|----------------|----------------|
| (A) | 2        | 44.8%             | 46.5%               | 5.1%  | 45.5% | **0.3%**       | 6.3%           |
|     | 3        | 7.9%              | 33.4%               | 4.4%  | 13.6% | **3.1%**       | 7.5%           |
|     | 4        | 4.6%              | 40.0%               | 2.2%  | 19.1% | **1.3%**       | 9.5%           |
|     | 5        | 1.1%              | 38.5%               | 0.3%  | 1.7%  | **0%**         | **0%**         |
|     | $\mathcal{P}_{122}$ | Binary Projection | Weighted Projection | NHCut | HOSVD | Tensor SCORE-1 | Tensor SCORE-2 |
|     | 0.2      | 7.9%              | 33.4%               | 4.4%  | 13.6% | **3.1%**       | 7.5%           |
| (B) | 0.3      | 29.3%             | 40.6%               | 10.8% | 6.5%  | **0.7%**       | 0.8%           |
|     | 0.4      | 39.5%             | 45.7%               | 35.7% | 1.6%  | 22.8%          | **0%**         |
|     | 0.5      | 34.1%             | 47.1%               | 45.5% | 1.5%  | 17.4%          | **0%**         |
|     | $n$      | Binary Projection | Weighted Projection | NHCut | HOSVD | Tensor SCORE-1 | Tensor SCORE-2 |
|     | 200      | 11.3%             | 34.4%               | 8.2%  | 22.5% | **5.9%**       | 16.9%          |
| (C) | 300      | 7.9%              | 33.4%               | 4.4%  | 13.6% | **3.1%**       | 7.5%           |
|     | 400      | 5.2%              | 30.0%               | **0.4%** | 5.9% | 2.7%         | 4.4%           |
|     | 500      | 6.4%              | 40.2%               | **2.6%** | 21.1% | 3.8%        | 15%            |

tential information loss.

## 5.2 Legislator Network

The first real-world hypergraph under consideration is a network of legislators in the Congress of the Republic of Peru Lee et al. (2016). The legislators in the Peruvian Congress constitute the nodes in this hypergraph network, and a group of nodes are connected by a hyperedge if they sponsored the same bill. In total there are 130 nodes (i.e., legislators) and 3687 hyperedges which correspond to all the bills proposed from 2006 to 2011 in the Peruvian Congress. This is a typical example where group interactions dominate, and hence hypergraphs provide a more faithful representation of the network than do dyadic ties. However, due to the lack of tools for hypergraph analysis, researchers often project this kind of networks to graphs for further study, which might result in information loss. Here we empirically demonstrate that tensor-SCORE better captures the party affiliation of the legislators than other graph methods and even those designed for hypergraphs. Before turning to the results, we note that the Peruvian Congress experienced a significant amount of restructuring during 2006 to 2011 and there were frequent shifts of the legislators' political identities. Therefore, we only include bills proposed in the first half of the 2006-2007 political year to construct the hypergraph when the Congress is relative stable and there were 7 parties at that time. This leaves us with 120 nodes and 802 hyperedges.

As there are 7 parties, we set the number of communities ($K$) to be 7. Accordingly, the output from tensor-SCORE is a factor matrix of $K - 1 = 6$ columns. We then cluster the rows of this matrix into 7 communities using K-means clustering. To visualize the clustering result,

we treat each row $i$ as an embedding of node $i$ in $R^6$, and then project the embeddings into $R^2$ through MDS (multidimensional scaling) for visualization. The result is shown in Figure 4A with nodes colored by their true party affiliations. Four communities clearly stand out in the plot which correspond to the four large parties. The 3 small parties are almost clustered together as they are small (two of them have only two members) and often vote with others strategically. As an illustration of how tensor-SCORE improves upon initial values and reg-HOOI result, we also visualize the positions of the nodes in the initial factor matrix (Figure 4B) and the positions of the nodes in the output of reg-HOOI (before normalizing the rows) (Figure 4C).

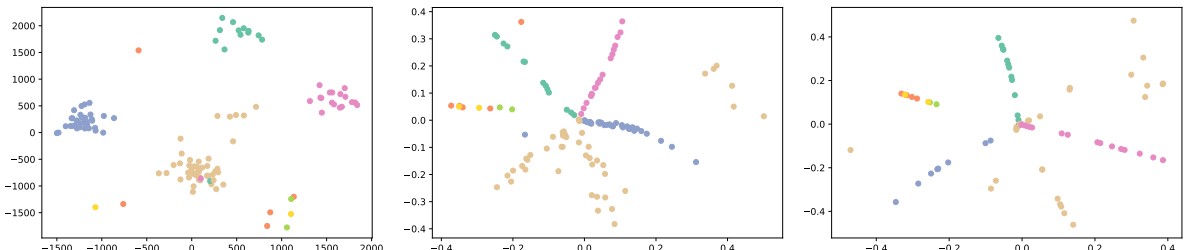

Figure 4: Clustering results from tensor-SCORE on the legislator network in the Peruvian Congress. Each dot denotes a node and its position is determined by the corresponding row in the factor matrix output by tensor-SCORE (A), the initial factor matrix (B), and the factor matrix output by reg-HOOI before row normalization (C). The nodes are colored by their party affiliations.

Compared to the ground-truth party affiliation, tensor-SCORE only misclassifies 14 nodes among 120 and achieves a low error rate of 11.7%. For comparison, clustering error from other community detection methods is summarized in Table 3. We can see that on this dataset tensor-SCORE outperforms all the other methods by a big margin. Its error rate is about 5 times smaller than HOSVD, 4 times smaller than binary and weighted projections, and 2 times smaller than NHCut.

## 5.3 Disease network from MEDLINE

We next examine a hypergraph network of diseases extracted from the MEDLINE database. The database contains more than 20 million papers published in biomedical sciences. Each paper is annotated with MeSH terms which indicate what chemicals, diseases, proteins, and other concepts are used in the paper. From the papers published in 1960 we construct a hypergraph of diseases, where nodes are MeSH terms for diseases of two kinds: Neoplasms (C04) and Nerve System Diseases (C10), and hyperedges are papers. The disease type will be used as the ground truth to assess the community detection methods. There are 9351 hyperedges and 190 nodes with 140 nodes in the Neoplasms community and 50 nodes in the other. We note that although there are only 2 types of diseases, we set $K = 6$ in the implementation of reg-HOOI algorithm to avoid information loss as the real data might not be generated from exactly a rank-2 model. In the end, the nodes are still divided into 2 clusters in the $k$-means clustering.

The clustering results are visualized in Figure 5, following the format of visualization for the legislator data. Compared to the ground-truth disease type, tensor-SCORE only misclassifies 8 cases among 190 and achieves a low error rate of 4.2%. It achieves the same error rate as NHCut but again outperforms the other methods. The comparisons are summarized in Table 3.

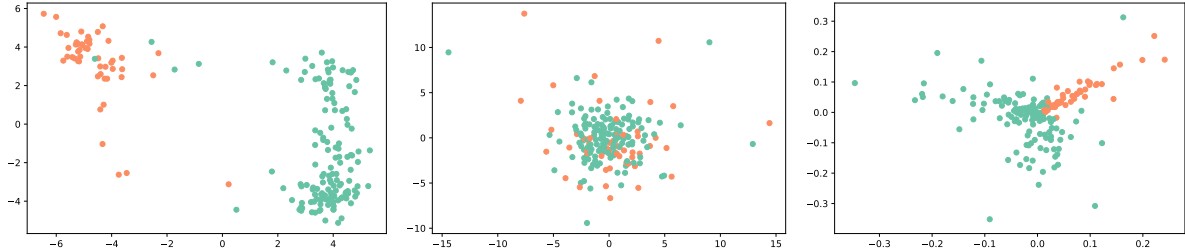

Figure 5: Clustering results from tensor-SCORE on the disease network from the MEDLINE dataset. Each dot denotes a node and its position is determined by the corresponding row in the factor matrix output by tensor-SCORE (A), the initial factor matrix (B), and the factor matrix output by reg-HOOI before row normalization (C). The nodes are colored by their disease type with orange corresponding to Nerve System Diseases (C10) and green to Neoplasms (C04).

Table 3: Performance of the community detection methods on two real-world hypergraph networks: legislators and diseases. The numbers in the table are the counts of misclassified nodes with misclassification rates in the parentheses. Tensor-SCORE uses HOSVD as initialization.

|  | tensor SCORE | Binary projection | Weighted projection | NHCut | HOSVD |
|---|---|---|---|---|---|
| Legislator | **14** (11.7%) | 58 (48.3%) | 52 (43.3%) | 23 (19.2%) | 67 (55.8%) |
| Disease | **8** (4.2%) | 13 (6.8%) | 46 (24.2%) | **8** (4.2%) | 12 (6.3%) |

# 6 Conclusion

This paper studies community detection for a hypergraph network. Existing works (e.g., Ahn et al. (2017); Angelini et al. (2015); Chien et al. (2018); Ghoshdastidar and Dukkipati (2014, 2015b); Kim et al. (2018); Lin et al. (2017)) only focus on the settings without degree heterogeneity. We consider a more general and realistic setting that allows for (potentialy severe) degree heterogeneity. We propose a spectral method, tensor-SCORE, which generalizes the community detection method SCORE (Jin, 2015) from graph networks to hypergraph networks. We introduce a degree-corrected block model for hypergraphs (hDCBM) and provide the rate of convergence of the Hamming clustering error.

Our work has several innovations. The first is a precise characterization of the population eigen-structure of hDCBM (i.e., Lemmas 3.1-3.2). It makes the connection between eigenvectors and targeted community labels transparent, and paves the way for generalizing SCORE to hypergraphs. The second is a new power iteration algorithm, reg-HOOI, for computing the spectral decomposition of network adjacency tensor. While power iteration methods are known to have superior performance in spectral decomposition of Gaussian tensors (Richard and Montanari, 2014; Zhang and Xia, 2018), their use in hypergraph tensors has never been investigated. In the hypergraph literature, existing spectral decomposition methods include the HOSVD (Ghoshdastidar and Dukkipati, 2014) and projection-to-graph (Zhou et al., 2007; Ghoshdastidar and Dukkipati, 2017). We discover that power iteration has the advantage of simultaneously mitigating information loss and achieving sharp error rate. Third, we propose two methods for initializing power iteration. They can be respectively thought as improved versions of the standard HOSVD and the standard graph projection. These initializations per se are interesting new spectral decomposition methods. Our last innovation is on the theoretical side. The asymptotic

behavior of hypergraph tensors largely differs from the well-studied Gaussian tensors, and we need to develop new technical tools. Our main technical result is a sharp concentration inequality on the incoherence norm of a sparse random tensor (Theorem 5), which is of independent interest.

Our idea can be extended in many ways. The error rate we obtain decays polynomially with the network size and degree parameters. By combining our method with an additional refinement step (Chien et al., 2018), we can obtain an exponentially decaying rate. In fact, according to the recent result of Abbe et al. (2020) which shows that spectral methods are able to achieve exponential rates for graph network community detection, we conjecture that our method, without refinement, already attains an exponentially fast rate. However, to prove this conjecture is extremely challenging, since it requires sharp row-wise large-deviation bounds for the factor matrix from tensor spectral decomposition. We leave it to future work.

The mixed membership models (Airoldi et al., 2008; Jin et al., 2024) are generalizations of block models by allowing a node to belong to multiple communities via fractional memberships. The tensor-SCORE can be extended to mixed membership estimation, by combining it with the simplex vertex hunting techniques (Jin et al., 2024; Ke and Wang, 2024) used in mixed membership learning. Another problem of great interest is the global testing (Jin et al., 2021), where we test the null hypothesis that the hypergraph network has only one community versus the alternative hypothesis that it contains multiple communities. The paper largely focuses on uniform hypergraphs. We provide a dummy-node approach for applying our method on non-uniform hypergraphs. While this approach works reasonably well in practice, it leaves much space for improvement. How to optimally aggregate information of hyperedges of different orders is an interesting research question.

The encouraging results on real data manifests the promise of tensor-SCORE as a hypergraph analysis method in practice. As the research community increasingly realize the importance of higher-order structures in systems of biology (Grilli et al., 2017), transportation and neural networks (Benson et al., 2016), knowledge discovery (Shi et al., 2015), and many other scientific fields, tensor-SCORE provides a new tool to tackle this complexity for novel scientific insights.

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

# A    Proofs of results in main text

Recall from Lemma 3.1 that $\mathcal{Q} = \mathcal{C} \times_1 \Xi \times_2 \cdots \times_m \Xi$ where the core tensor $\mathcal{C} = \|\theta\|^m \cdot \mathcal{P} \times_1 (UD) \times_2 \cdots \times_m (UD)$ where the orthogonal matrix $U$ has columns $u_k$s defined in Lemma 3.1. We also denote $\mathcal{P}_{\mathsf{max}} = \|\mathcal{P}\|_{\mathsf{max}}$. Let $\tilde{\otimes}$ denote the Kronecker product. In addition, $\otimes$ denotes the tensor product so that $u_1 \otimes u_2 \otimes \cdots \otimes u_m \in \mathbb{R}^{d_1 \times \cdots \times d_m}$ if $u_j \in \mathbb{R}^{d_j}$. On the other hand, $u_1 \tilde{\otimes} u_2 \tilde{\otimes} \cdots \tilde{\otimes} u_m \in \mathbb{R}^{d_1 d_2 \cdots d_m}$. Let $\sigma_1(M) \geq \sigma_2(M) \geq \cdots$ denote the singular values of $M$.

## A.1    Bounds of Diagonal Tensor

Recall that the adjacency tensor $\mathcal{A}$ has expectation

$$\mathbb{E}\mathcal{A} = \mathcal{Q} - \mathrm{diag}(\mathcal{Q})$$

where the low-rank tensor $\mathcal{Q}$ admit HOSVD as claimed in Lemma 3.1. The diagonal tensor $\mathrm{diag}(\mathcal{Q})$ is defined by

$$[\mathrm{diag}(\mathcal{Q})](i_1, \cdots, i_m) = \begin{cases} 0, & \text{if } i_1 \neq i_2 \neq \cdots \neq i_m \\ \mathcal{Q}(i_1, \cdots, i_m), & \text{otherwise} . \end{cases}$$

Note that $\mathrm{diag}(\mathcal{Q})$'s eigenspaces can be different from $\mathcal{Q}$'s eigenspaces.

In this section, we investigate the size of the diagonal tensor and prove the upper bound for $\|\mathrm{diag}(\mathcal{Q})\|$ where $\|\cdot\|$ denote the tensor operator norm.

**Lemma 1.** The following bound holds

$$\|\mathrm{diag}(\mathcal{Q})\| \leq \binom{m}{2} \kappa_1 \theta_{\mathsf{max}}^2 \|\theta\|^{m-2} d_{\mathsf{min}}^{-2}$$

where $\kappa_1$ is defined as in Lemma 3.1.

Recall from Lemma 3.1 that the smallest non-zero singular value of $\mathcal{M}(\mathcal{Q})$ is $\kappa_K \|\theta\|^m$. If $m^2 \kappa_1 d_{\mathsf{min}}^{-2} = O(\kappa_K)$, then we simply have the relative affect from the diagonal part bounded by $\|\mathrm{diag}(\mathcal{Q})\|/\sigma_K(\mathcal{M}(\mathcal{Q})) = O(\theta_{\mathsf{max}}^2/\|\theta\|^2)$.

*Proof of Lemma 1.* For any pair $(k_1, k_2) \in \{1, \cdots, m\}$ and $k_1 \neq k_2$, we denote a $m$-th order tensor $\mathcal{D}_{\mathcal{Q}}^{(k_1, k_2)}$ whose entries are

$$\mathcal{D}_{\mathcal{Q}}^{(k_1, k_2)}(i_1, i_2, \cdots, i_m) = \begin{cases} 0, & \text{if } i_{k_1} \neq i_{k_2} \\ \mathcal{Q}(i_1, \cdots, i_m) & \text{if } i_{k_1} = i_{k_2}. \end{cases}$$

The tensor $\mathcal{D}_{\mathcal{Q}}^{k_1, k_2}$ represents the diagonal of $\mathcal{Q}$ restricted to $i_{k_1} = i_{k_2}$. We define the following $m$-th order tensor $\mathcal{D}_{\mathcal{Q}}$ by

$$\mathcal{D}_{\mathcal{Q}} = \sum_{(k_1, k_2) \in [m]} \mathcal{D}_{\mathcal{Q}}^{(k_1, k_2)}.$$

We claim that $\|\mathrm{diag}(\mathcal{Q})\| \leq \|\mathcal{D}_{\mathcal{Q}}\|$. Recall that both $\mathrm{diag}(\mathcal{Q})$ and $\mathcal{D}_{\mathcal{Q}}$ have non-negative entries. It suffices to show that $\mathrm{diag}(\mathcal{Q}) \leq \mathcal{D}_{\mathcal{Q}}$ elementwisely since the operator norm of a

non-negative tensor can always be realized by the inner product with non-negative unit-norm vectors. Let's consider any $(i_1, \cdots, i_m)$-th entry. If $i_1 \neq i_2 \neq \cdots \neq i_m$, then, by the definitions of $\operatorname{diag}(\mathcal{Q})$ and $\mathcal{D}_\mathcal{Q}$, we have

$$\operatorname{diag}(\mathcal{Q})(i_1, \cdots, i_m) = 0 \quad \text{and} \quad \mathcal{D}_\mathcal{Q}(i_1, \cdots, i_m) = 0.$$

On the other hand, if some indices in $i_1, \cdots, i_m$ are equal, then there exist $k_1, k_2 \in [m]$ so that $i_{k_1} = i_{k_2}$. Then,

$$\operatorname{diag}(\mathcal{Q})(i_1, \cdots, i_m) = \mathcal{D}_\mathcal{Q}^{(k_1, k_2)}(i_1, \cdots, i_m)$$

implying that

$$\operatorname{diag}(\mathcal{Q})(i_1, \cdots, i_m) \leq \mathcal{D}_\mathcal{Q}(i_1, \cdots, i_m).$$

Therefore, we conclude that $\operatorname{diag}(\mathcal{Q})$ is dominated by $\mathcal{D}_\mathcal{Q}$ elementwisely implying that $\|\operatorname{diag}(\mathcal{Q})\| \leq \|\mathcal{D}_\mathcal{Q}\|$. It suffices to prove the upper bound of $\|\mathcal{D}_\mathcal{Q}\|$. Clearly,

$$\|\mathcal{D}_\mathcal{Q}\| \leq \sum_{(k_1, k_2) \in [m]} \|\mathcal{D}_\mathcal{Q}^{(k_1, k_2)}\| = \binom{m}{2} \|\mathcal{D}_\mathcal{Q}^{(1,2)}\|$$

where the last equality is due to the symmetricity of $\mathcal{Q}$. Write

$$\mathcal{D}_\mathcal{Q}^{(1,2)} = \sum_{i=1}^n (e_i \otimes e_i) \otimes \mathcal{Q}(i, i, :, :, \cdots, :)$$

where $e_i \in \mathbb{R}^n$ denotes the $i$-th canonical basis vector. For any unit-norm vectors $u_1, \cdots, u_m$, we have

$$
\begin{aligned}
\langle \mathcal{D}_\mathcal{Q}^{(1,2)}, u_1 \otimes \cdots \otimes u_m \rangle &= \sum_{i=1}^n u_1(i) u_2(i) \langle \mathcal{Q}(i, i, :, \cdots, :), u_3 \otimes \cdots \otimes u_m \rangle \\
&\leq \max_i |\langle \mathcal{Q}(i, i, :, \cdots, :), u_3 \otimes \cdots \otimes u_m \rangle| \cdot \sum_{i=1}^n |u_1(i) u_2(i)| \\
&\leq \max_i |\langle \mathcal{Q}(i, i, :, \cdots, :), u_3 \otimes \cdots \otimes u_m \rangle| \cdot \|u_1\|_{\ell_2} \|u_2\|_{\ell_2} \\
&\leq \max_i |\langle \mathcal{Q}(i, i, :, \cdots, :), u_3 \otimes \cdots \otimes u_m \rangle| \\
&\leq \max_{1 \leq i \leq m} \|\mathcal{Q}(i, i, :, \cdots, :)\|.
\end{aligned}
$$

Recall the definition of $\mathcal{Q}$ and its Tucker decomposition $\mathcal{Q} = [\mathcal{C}; \Xi, \cdots, \Xi]$, we write

$$\mathcal{Q}(i, i, :, \cdots, :) = \mathcal{C} \times_1 (e_i^\top \Xi) \times_2 (e_i^\top \Xi) \times_3 (\Xi) \times_4 \cdots \times_m (\Xi).$$

By definition of $\mathcal{C}$,

$$\|\mathcal{Q}(i, i, :, \cdots, :)\| \leq \|\mathcal{C}\| \|e_i^\top \Xi\|^2 \leq \kappa_1 \|\theta\|^m \cdot \frac{\theta_i^2}{\|\theta\|^2} \cdot \frac{1}{d_{\min}^2} \leq \kappa_1 \theta_i^2 \|\theta\|^{m-2} d_{\min}^{-2}.$$

Therefore,

$$\|\mathcal{D}_Q^{(1,2)}\| \leq \kappa_1 \theta_{\max}^2 \|\theta\|^{m-2} d_{\min}^{-2}$$

and we conclude with

$$\|\mathrm{diag}(\mathcal{Q})\| \leq \binom{m}{2}\kappa_1\theta_{\mathsf{max}}^2\|\theta\|^{m-2}d_{\mathsf{min}}^{-2}.$$

$\square$

## A.2    Warm Initializations by standard HOSVD

To show the advantage of diagonal-removed HOSVD, we present the performance of spectral initialization by the vanilla HOSVD.

**Lemma A.1** (HOSVD initialization). *Denote $\widehat{\Xi}^{(0)}$ the top-K left singular vectors of $\mathcal{M}_1(\mathcal{A})$. Then, with probability at least $1 - n^{-2}$,*

$$\|\widehat{\Xi}^{(0)}\widehat{\Xi}^{(0)\top} - \Xi\Xi\| \leq C_1(m!)\frac{\sqrt{\mathcal{P}_{\mathsf{max}}\theta_{\mathsf{max}}\|\theta\|_1^{m-1}\log n} + \kappa_1\sqrt{K}\theta_{\mathsf{max}}^2\|\theta\|^{m-2}d_{\mathsf{min}}^{-2} + \log n}{\kappa_K\|\theta\|^m} \tag{A.1}$$

*for some large enough constant $C_1 > 0$.*

*Proof of Lemma A.1.* Recall that

$$\mathcal{M}_1(\mathcal{A}) = \mathcal{M}_1(\mathcal{Q}) + \mathcal{M}_1(\mathcal{A} - \mathbb{E}\mathcal{A}) + \mathcal{M}_1(\mathrm{diag}(\mathcal{Q})).$$

We first prove the upper bound of $\|\mathcal{M}_1(\mathrm{diag}(\mathcal{Q}))\|$. Similarly, as in the proof of Lemma 1,

$$\begin{aligned}
\|\mathcal{M}_1(\mathrm{diag}(\mathcal{Q}))\| \leq \|\mathcal{M}_1(\mathcal{D}_{\mathcal{Q}})\| &\leq \sum_{(k_1,k_2)\in[m]}\|\mathcal{M}_1(\mathcal{D}_{\mathcal{Q}}^{(k_1,k_2)})\| \\
&\leq m^2 \cdot \max_{(k_1,k_2)\in[m]}\|\mathcal{M}_1(\mathcal{D}_{\mathcal{Q}}^{(k_1,k_2)})\|.
\end{aligned}$$

W.L.O.G., we prove the bound for $\|\mathcal{M}_1(\mathcal{D}_{\mathcal{Q}}^{(1,2)})\|$. Recall that

$$\mathcal{D}_{\mathcal{Q}}^{(1,2)} = \sum_{i=1}^{n}(e_i \otimes e_i) \otimes \mathcal{Q}(i,i,:,\cdots,:).$$

We write

$$\mathcal{M}_1(\mathcal{D}_{\mathcal{Q}}^{(1,2)}) = \sum_{i=1}^{n} e_i \otimes \left(\mathrm{vec}(\mathcal{Q}(i,i,:,\cdots,:))\tilde{\otimes}e_i\right)$$

where $\tilde{\otimes}$ denotes the Kronecker product. For $u \in \mathbb{R}^n$ and $v \in \mathbb{R}^{n^{m-1}}$, we have

$$\langle \mathcal{M}_1(\mathcal{D}_{\mathcal{Q}}^{(1,2)}), u \otimes v \rangle = \sum_{i=1}^{n} \langle u, e_i \rangle \langle \text{vec}(\mathcal{Q}(i,i,:,\cdots,:)) \tilde{\otimes} e_i, v \rangle$$

$$\leq \sqrt{\sum_{i=1}^{n} \langle u, e_i \rangle^2 \sum_{i=1}^{n} \langle \text{vec}(\mathcal{Q}(i,i,:,\cdots,:)) \tilde{\otimes} e_i, v \rangle^2}$$

$$\leq \|u\| \cdot \sqrt{\sum_{i=1}^{n} \langle \text{vec}(\mathcal{Q}(i,i,:,\cdots,:)) \tilde{\otimes} e_i, v \rangle^2}$$

$$\leq \|u\| \|v\| \cdot \max_{1 \leq i \leq n} \|\mathcal{Q}(i,i,:,\cdots,:)\|_{\text{F}} \leq \theta_{\max}^2 \|u\| \|v\| \cdot \kappa_1 \sqrt{K} \|\theta\|^{m-2} d_{\min}^{-2}$$

where we used the fact $\text{rank}(\mathcal{M}_m(\mathcal{Q}(i,i,:,\cdots,:))) \leq K$ and $\|\mathcal{M}_m(\mathcal{Q}(i,i,:,\cdots,:))\| \leq \kappa_1 \theta_{\max}^2 \|\theta\|^{m-2} d_{\min}^{-2}$. Therefore, we conclude that

$$\|\mathcal{M}_1(\mathcal{D}_{\mathcal{Q}}^{(1,2)})\| \leq \kappa_1 \theta_{\max}^2 \sqrt{K} \|\theta\|^{m-2} d_{\min}^{-2}$$

and so that

$$\|\mathcal{M}_1(\text{diag}(\mathcal{Q}))\| \leq \kappa_1 m^2 \sqrt{K} \cdot \theta_{\max}^2 \|\theta\|^{m-2} d_{\min}^{-2}.$$

We then prove the upper bound of $\|\mathcal{M}_1(\mathcal{A}) - \mathbb{E}\mathcal{M}_1(\mathcal{A})\|$. Recall that

$$\mathcal{M}_1(\mathcal{A}) = \sum_{i_1 \neq \cdots \neq i_m}^{n} \mathcal{A}(i_1, \cdots, i_m) \cdot \sum_{(j_1,\cdots,j_m) \in \mathfrak{P}(i_1,\cdots,i_m)} e_{j_1} (e_{j_2} \tilde{\otimes} \cdots \tilde{\otimes} e_{j_m})^\top$$

where $\mathfrak{P}(i_1, \cdots, i_m)$ denotes the set of all permutations of $(i_1, \cdots, i_m)$. For any $(i_1, \cdots, i_m)$,

$$\left\| \mathcal{A}(i_1, \cdots, i_m) \cdot \sum_{(j_1,\cdots,j_m) \in \mathfrak{P}(i_1,\cdots,i_m)} e_{j_1} (e_{j_2} \tilde{\otimes} \cdots \tilde{\otimes} e_{j_m})^\top \right\| \leq m!.$$

Meanwhile,

$$\left\| \sum_{i_1 \neq \cdots \neq i_m}^{n} \mathbb{E}\mathcal{A}(i_1, \cdots, i_m)^2 \left( \sum_{(j_1,\cdots,j_m) \in \mathfrak{P}(i_1,\cdots,i_m)} e_{j_1} (e_{j_2} \tilde{\otimes} \cdots \tilde{\otimes} e_{j_m})^\top \right) \right.$$

$$\left. \cdot \left( \sum_{(j_1,\cdots,j_m) \in \mathfrak{P}(i_1,\cdots,i_m)} e_{j_1} (e_{j_2} \tilde{\otimes} \cdots \tilde{\otimes} e_{j_m})^\top \right)^\top \right\|$$

$$\leq (m!) \cdot \max_{i_1 \in [n]} \sum_{i_2,\cdots,i_m \geq 1}^{n} \mathbb{E}\mathcal{A}(i_1, \cdots, i_m).$$

By the definition of $\mathcal{Q}$, we obtain

$$\max_{i_1 \in [n]} \sum_{i_2,\cdots,i_m \geq 1}^{n} \mathbb{E}\mathcal{A}(i_1, \cdots, i_m) \leq \max_{i_1 \in [n]} \sum_{i_2,\cdots,i_m \geq 1}^{n} \mathcal{Q}(i_1, \cdots, i_m)$$

$$\leq \mathcal{P}_{\max} \cdot \theta_{\max} \cdot \|\theta\|_1^{m-1}$$

where we used the fact $\mathcal{Q}(i_1, \cdots, i_m) \leq \mathcal{P}_{\mathsf{max}} \cdot \prod_{j=1}^m \theta_{i_j}$ and so that

$$\sum_{i_2, \cdots, i_m \geq 1}^n \mathcal{Q}(i_1, \cdots, i_m) \leq \sum_{i_2, \cdots, i_m \geq 1}^n \mathcal{P}_{\mathsf{max}} \cdot \prod_{j=1}^m \theta_{i_j} \leq \mathcal{P}_{\mathsf{max}} \theta_{i_1} \cdot \|\theta\|_1^{m-1}.$$

By matrix Bernstein inequality (Tropp (2012)), with probability at least $1 - \frac{1}{n^2}$, we get

$$\|\mathcal{M}_1(\mathcal{A}) - \mathbb{E}\mathcal{M}_1(\mathcal{A})\| \leq C_1(m!)\sqrt{\mathcal{P}_{\mathsf{max}} \cdot \theta_{\mathsf{max}} \|\theta\|_1^{m-1} \log n} + C_2(m!) \log n$$

for some absolute constants $C_1, C_2 > 0$. By Davis-Kahan Theorem,

$$\|\hat{\Xi}^{(0)}(\hat{\Xi}^{(0)})^\top - \Xi\Xi^\top\| \leq \sqrt{2} \cdot \frac{\|\mathcal{M}_1(\mathrm{diag}(\mathcal{Q}))\| + \|\mathcal{M}_1(\mathcal{A}) - \mathbb{E}\mathcal{M}_1(\mathcal{A})\|}{\sigma_K(\mathcal{M}_1(\mathcal{Q}))}.$$

Together with $\sigma_K(\mathcal{M}_1(\mathcal{Q})) = \kappa_K \|\theta\|^m$, with probability at least $1 - n^{-1}$, we get

$$\|\hat{\Xi}^{(0)}(\hat{\Xi}^{(0)})^\top - \Xi\Xi^\top\| \leq \frac{C_1 \kappa_1 m^2 \sqrt{K} \cdot \theta_{\mathsf{max}}^2 \|\theta\|^{m-2} d_{\mathsf{min}}^{-2} + C_2 m!\sqrt{\mathcal{P}_{\mathsf{max}} \theta_{\mathsf{max}} \|\theta\|_1^{m-1} \log n} + C_3 m! \log n}{\kappa_K \|\theta\|^m}$$

where $C_1, C_2, C_3 > 0$ are absolute constants. $\qquad \square$

## A.3   Proof of Lemma 3.1

By hDCBM and the definition of Tucker decomposition, we have

$$\mathcal{Q} = [\mathcal{P}; \Theta\Pi, \ldots, \Theta\Pi] = \mathcal{P} \times_1 (\Theta\Pi) \times_2 \cdots \times_m (\Theta\Pi).$$

Introduce the tensor $\mathcal{P}_* = \mathcal{P} \times_1 D \times_2 \cdots \times_m D$, where $D = \mathrm{diag}(d_1, \ldots, d_K)$. By elementary calculations, the above equation can be re-written as

$$\mathcal{Q} = \mathcal{P}_* \times_1 (\Theta\Pi D^{-1}) \times_2 \cdots \times_m (\Theta\Pi D^{-1})$$

The mode-1 matricization of $\mathcal{Q}$ thus has the form (see (Kolda and Bader, 2009, Page 462))

$$\mathcal{M}_1(\mathcal{Q}) = (\Theta\Pi D^{-1}) \cdot \mathcal{M}_1(\mathcal{P}_*) \cdot [\underbrace{(\Theta\Pi D^{-1})\tilde{\otimes}\cdots\tilde{\otimes}(\Theta\Pi D^{-1})}_{m-1}]^\top \qquad (\text{A.2})$$

where $\tilde{\otimes}$ denotes the Kronecker product. Note that $\mathcal{M}_1(\mathcal{P}_*)$ is the matrix $G$. Let $\kappa_k > 0$ be the $k$-th largest singular value of $G$ and let $u_k \in \mathbb{R}^K$ and $v_k \in \mathbb{R}^{K^{m-1}}$ be the corresponding left and right singular vectors. Write $\Delta = \mathrm{diag}(\kappa_1, \ldots, \kappa_K)$, $U = [u_1, \ldots, u_K]$, and $V = [v_1, \ldots, v_K]$. Then,

$$\mathcal{M}_1(\mathcal{P}_*) = U\Delta V^\top.$$

We plug it into (A.2) and find that

$$\mathcal{M}_1(\mathcal{Q}) = \underbrace{(\Theta\Pi D^{-1})U}_{\equiv S_1} \cdot \Delta \cdot \underbrace{V^\top[(\Theta\Pi D^{-1})\tilde{\otimes}\cdots\tilde{\otimes}(\Theta\Pi D^{-1})]^\top}_{\equiv S_2^\top}. \qquad (\text{A.3})$$

We study the matrices $S_1$ and $S_2$. Note that $d_k$ can be written as $\|\theta\|^{-1}\sqrt{\sum_{i=1}^{n}\theta_i^2\pi_i(k)}$. It follows that

$$\|\theta\|^2 D^2 = \Pi^\top \Theta^2 \Pi.$$

Combing it with $U^\top U = I_K$, we obtain

$$S_1^\top S_1 = U^\top(D^{-1}\Pi^\top\Theta^2\Pi D^{-1})U = \|\theta\|^2 U^\top U = \|\theta\|^2 I_K.$$

Moreover, since $V^\top V = I_K$, using the universal equalities of $(A\tilde{\otimes}B)(C\tilde{\otimes}D) = (AC)\tilde{\otimes}(BD)$ and $(A\tilde{\otimes}B)^\top = B^\top\tilde{\otimes}A^\top$, we can derive

$$S_2^\top S_2 = V^\top[\underbrace{(D^{-1}\Pi^\top\Theta^2\Pi D^{-1})\tilde{\otimes}\cdots\tilde{\otimes}(D^{-1}\Pi^\top\Theta^2\Pi D^{-1})}_{m-1}]V = \|\theta\|^{2m-2}V^\top V = \|\theta\|^{2m-2}I_K.$$

Let $\Xi = \|\theta\|^{-1}S_1$ and $H = \|\theta\|^{-(m-1)}S_2$. We plug the above equations into (A.3). It gives

$$\mathcal{M}_1(\mathcal{Q}) = \Xi \cdot (\|\theta\|^m\Delta) \cdot H^\top, \qquad \text{where} \quad \Xi^\top\Xi = H^\top H = I_K. \tag{A.4}$$

It is clear that (A.4) gives the SVD of $\mathcal{M}_1(\mathcal{Q})$. Then, the diagonal matrix $\|\theta\|^m\Delta$ contains the singular values, and $\Xi = \|\theta\|^{-1}\Theta\Pi D^{-1}U$ contains the left singular vectors. It implies

$$\lambda_k = \kappa_k\|\theta\|^m, \qquad \xi_k = \|\theta\|^{-1}\Theta\Pi D^{-1}u_k, \qquad 1 \le k \le K.$$

This proves the claim.

## A.4   Proof of Lemma 3.2

Consider the first bullet point. Given an initialization $\widetilde{\Xi}$, the output of the first iteration, $\Xi^{(1)}$, contains left singular vectors of

$$\mathcal{M}_1(\Omega), \qquad \text{where} \quad \Omega \equiv \mathcal{Q} \times_2 \widetilde{\Xi}^\top \times_3 \cdots \times_m \widetilde{\Xi}^\top.$$

Note that $\mathcal{Q} = \mathcal{P} \times_1 (\Theta\Pi) \times_2 \cdots \times_m (\Theta\Pi)$. Using the formula of $\mathcal{X} \times_j A \times_j B = \mathcal{X} \times_j (BA)$ and $\mathcal{X} \times_i A \times_j B = \mathcal{X} \times_j B \times_i A$ (see (Kolda and Bader, 2009, Page 461)), we have

$$\begin{aligned}
\Omega &= \mathcal{P} \times_1 (\Theta\Pi) \times_2 \cdots \times_m (\Theta\Pi) \times_2 \widetilde{\Xi}^\top \times_3 \cdots \times_m \widetilde{\Xi}^\top \\
&= \mathcal{P} \times_1 (\Theta\Pi) \times_2 (\widetilde{\Xi}^\top\Theta\Pi) \times_2 \cdots \times_m (\widetilde{\Xi}^\top\Theta\Pi) \\
&= \mathcal{P}_* \times_1 (\Theta\Pi D^{-1}) \times_2 (\widetilde{\Xi}^\top\Theta\Pi D^{-1}) \times_2 \cdots \times_m (\widetilde{\Xi}^\top\Theta\Pi D^{-1}),
\end{aligned}$$

where $D = \text{diag}(d_1,\ldots,d_K)$ and $\mathcal{P}_* = \mathcal{P} \times_1 D \times_2 \cdots \times_m D$. By the matricization formula ((Kolda and Bader, 2009, Page 462)),

$$\mathcal{M}_1(\Omega) = \Theta\Pi D^{-1} \cdot \mathcal{M}_1(\mathcal{P}_*) \cdot [\underbrace{(\widetilde{\Xi}^\top\Theta\Pi D^{-1})\tilde{\otimes}\cdots\tilde{\otimes}(\widetilde{\Xi}^\top\Theta\Pi D^{-1})}_{m-1}]^\top$$

where $\tilde{\otimes}$ denotes the Kronecker product. Here, $\mathcal{M}_1(\mathcal{P}_*)$ is the matrix $G$. Same as in the proof of Lemma 3.1, let $\mathcal{M}_1(\mathcal{P}_*) = U\Delta V^\top$ be the SVD. It follows that

$$\mathcal{M}_1(\Omega) = \underbrace{\Theta\Pi D^{-1} U}_{S_1} \cdot \Delta \cdot \underbrace{V^\top[(\widetilde{\Xi}^\top \Theta\Pi D^{-1})\tilde{\otimes}\cdots\tilde{\otimes}(\widetilde{\Xi}^\top\Theta\Pi D^{-1})]^\top}_{\equiv \widetilde{S}_2^\top}. \tag{A.5}$$

In the proof of Lemma 3.1, we have shown $S_1 = \|\theta\|\Xi$, where $\Xi$ is the output of applying HOSVD to $\mathcal{Q}$. Since $U$ is an orthogonal matrix, we write $\widetilde{\Xi}^\top\Theta\Pi D^{-1} = \widetilde{\Xi}^\top\Theta\Pi D^{-1}UU^\top = \|\theta\|\widetilde{\Xi}^\top\Xi U^\top$. Plugging it into the definition of $\widetilde{S}_2$ and using the formula $(A\tilde{\otimes}B)(C\tilde{\otimes}D) = AC\tilde{\otimes}BD$ in a reverse way, we have

$$\begin{aligned}
\widetilde{S}_2^\top &= \|\theta\|^{m-1}V^\top[(\widetilde{\Xi}^\top\Xi U^\top)\tilde{\otimes}\cdots\tilde{\otimes}(\widetilde{\Xi}^\top\Xi U^\top)] \\
&= \|\theta\|^{m-1}V^\top[(\widetilde{\Xi}^\top\Xi)\tilde{\otimes}\cdots\tilde{\otimes}(\widetilde{\Xi}^\top\Xi)]\cdot[U^\top\tilde{\otimes}\cdots\tilde{\otimes}U^\top] \\
&= \|\theta\|^{m-1}\Big([U\tilde{\otimes}\cdots\tilde{\otimes}U]\cdot[(\Xi^\top\widetilde{\Xi})\tilde{\otimes}\cdots\tilde{\otimes}(\Xi^\top\widetilde{\Xi})]\cdot V\Big)^\top.
\end{aligned}$$

We plug in the expressions of $S_1$ and $\widetilde{S}_2$ into (A.5). It gives

$$\mathcal{M}_1(\Omega) = \|\theta\|^m \cdot \Xi \cdot \Delta \cdot \Big(\underbrace{[U\tilde{\otimes}\cdots\tilde{\otimes}U]\cdot[(\Xi^\top\widetilde{\Xi})\tilde{\otimes}\cdots\tilde{\otimes}(\Xi^\top\widetilde{\Xi})]\cdot V}_{\equiv \widetilde{H}}\Big)^\top. \tag{A.6}$$

First, the matrix $\Xi^\top\widetilde{\Xi} = I_K - (\widetilde{\Xi}^\top\widetilde{\Xi} - \Xi^\top\Xi)$. Our assumption $\|\Xi\Xi^\top - \widetilde{\Xi}\widetilde{\Xi}^\top\| < 1$ guarantees that this matrix has a full rank $K$. By the universal equality that $\text{rank}(A_1\tilde{\otimes}\cdots\tilde{\otimes}A_k) = \text{rank}(A_1)\times\cdots\times\text{rank}(A_k)$, the matrix $[(\Xi^\top\widetilde{\Xi})\tilde{\otimes}\cdots\tilde{\otimes}(\Xi^\top\widetilde{\Xi})]$ has a full rank $K^{m-1}$. Second, it is known that the Kronecker product of orthogonal matrices is still an orthogonal matrix. Hence, the matrix $[U\tilde{\otimes}\cdots\tilde{\otimes}U]$ is an orthogonal matrix in dimension $K^{m-1}$. Third, the matrix $V$ has orthonormal columns. Combining the above, we conclude that the rank of $\widetilde{H}$ equals to $K^{m-1}$. It follows from (A.6) that the column space of the left singular vectors of $\mathcal{M}_1(\Omega)$ is the same as the column space of $\Xi$, i.e.,

$$\Xi^{(1)} = \Xi O, \qquad \text{for an orthogonal matrix } O.$$

This proves the first claim.

Consider the second and third bullet points. We note that, if $\widetilde{\Xi} = \Xi O$ for an orthogonal matrix $O$, then $\Xi^\top\widetilde{\Xi} = \Xi^\top\Xi O = O$ is an orthogonal matrix. It follows that the matrix $[U\tilde{\otimes}\cdots\tilde{\otimes}U]\cdot[(\Xi^\top\widetilde{\Xi})\tilde{\otimes}\cdots\tilde{\otimes}(\Xi^\top\widetilde{\Xi})]$ is an orthogonal matrix in dimension $K^{m-1}$. Therefore, the matrix $\widetilde{H}$ in (A.6) contains orthonormal columns, and (A.6) is indeed the SVD of $\mathcal{M}_1(\Omega)$. It implies that

$$\Xi^{(1)} = \Xi, \qquad \text{when } \widetilde{\Xi} = \Xi O \text{ for an orthogonal matrix } O.$$

As a result, if we start from $\widetilde{\Xi}$ that has the same column space of $\Xi$ and run one iteration of HOOI, then we will exactly obtain $\Xi$. The last two bullet points follow immediately.

## A.5   Proof of Theorem 1

In Tensor-SCORE, the estimated community labels are obtained from running a $k$-means clustering on rows of $\hat{R}$ (defined in (3.8)). Below, we first study the matrix $\hat{R}$; next, we study its population counterpart $R$; last, we analyze the k-means clustering algorithm and bound the clustering error. For convenience, we introduce the notation

$$\widetilde{err}_n = \frac{\sqrt{K^{m-2}\theta_{\max}^{m-1}\|\theta\|_1 \log(n)} + (\sqrt{K}\theta_{\max}/\|\theta\|)^{m-1}\log(n)}{\|\theta\|^m} + \frac{K\theta_{\max}^2}{\|\theta\|^2}.$$

It is seen that $err_n^2 \asymp [\|\theta\|^2/(n\theta_{\min}^2)] \cdot \widetilde{err}_n^2$.

In preparation, we state a useful result from Theorem 4. By condition (4.2), $\kappa_K \geq \gamma_n$ and $\|\mathcal{P}\|_{\max} = O(1)$; by condition (4.1), $d_{\max} \asymp d_{\min} \asymp 1/\sqrt{K}$. It follows that $\beta(\mathcal{Q}) \leq \|\mathcal{P}\|_{\max}\theta_{\max}^{m-1}\|\theta\|_1 \leq C\theta_{\max}^{m-1}\|\theta\|_1$ and $\kappa_1 \leq \sqrt{K^m}\|\mathcal{P}\|_{\max} \cdot \|D\|^m = O(1)$. We plug them into Theorem 4 and find that, with probability $1 - n^{-2}$,

$$\|\hat{\Xi}\hat{\Xi}^\top - \Xi\Xi^\top\| \leq \frac{C\sqrt{K^{m-2}\theta_{\max}^{m-1}\|\theta\|_1 \log(n)} + C(\sqrt{K}\theta_{\max}/\|\theta\|)^{m-1}\log n + CK\theta_{\max}^2\|\theta\|^{m-2}}{\gamma_n\|\theta\|^m}$$

$$\leq C\gamma_n^{-1}\widetilde{err}_n$$

where $\hat{\Xi}$ denotes the output of regularized-HOOI algorithm. Write $\Xi_0 = [\xi_2, \ldots, \xi_K]$ and $\hat{\Xi}_0 = [\hat{\xi}_2, \ldots, \hat{\xi}_K]$. Note that condition (4.2) says that $\kappa_1 - \kappa_2 \geq \gamma_n$. By using a similar proof to that of Theorem 4, we can get a stronger result, that is

$$\|\hat{\xi}_1\hat{\xi}_1^\top - \xi_1\xi_1^\top\| \leq C\gamma_n^{-1}\widetilde{err}_n, \qquad \|\hat{\Xi}_0\hat{\Xi}_0^\top - \Xi_0\Xi_0^\top\| \leq C\gamma_n^{-1}\widetilde{err}_n.$$

By elementary linear algebra, the above imply that there exists $\omega \in \{\pm 1\}$ and an orthogonal matrix $O \in \mathbb{R}^{(K-1)\times(K-1)}$, such that

$$\|\hat{\xi}_1 - \omega\xi_1\| \leq C\gamma_n^{-1}\widetilde{err}_n, \qquad \|\hat{\Xi}_0 - \Xi_0 O^\top\| \leq C\gamma_n^{-1}\widetilde{err}_n.$$

The rank of the matrix $(\hat{\Xi}_0 - \Xi_0 O^\top)$ is at most $2(K-1)$. It follows that $\|\hat{\Xi}_0 - \Xi_0 O^\top\|_F \leq \sqrt{2(K-1)}\|\hat{\Xi}_0 - \Xi_0 O^\top\| \leq C\sqrt{K}\gamma_n^{-1}\widetilde{err}_n$. We immediate have

$$\|\hat{\xi}_1 - \omega\xi_1\|^2 \leq C\gamma_n^{-2}\widetilde{err}_n^2, \qquad \|\hat{\Xi}_0 - \Xi_0 O^\top\|_F^2 \leq CK\gamma_n^{-2}\widetilde{err}_n^2. \qquad (A.7)$$

We now prove the claim. First, we study the matrix $\hat{R}$. Its population counterpart $R$ is defined in (3.5). Introduce $\hat{R}^* \in \mathbb{R}^{n\times(K-1)}$ by

$$\hat{R}^*(i,k) = \hat{\xi}_{k+1}(i)/\hat{\xi}_1(i), \qquad 1 \leq i \leq n, 1 \leq k \leq K-1.$$

Denote by $r_i^\top$, $\hat{r}_i^\top$ and $(\hat{r}_i^*)^\top$ the $i$-th row of $R$, $\hat{R}$ and $\hat{R}^*$, respectively. Then, $\hat{r}_i$ is obtained by truncating large entries of $\hat{r}_i^*$ when $\|\hat{r}_i^*\| > \sqrt{K\log(n)}$. Let $\Xi_{0,i}^\top$ and $\hat{\Xi}_{0,i}^\top$ be the $i$-th row of $\Xi_0$

and $\hat{\Xi}_0$, respectively. By definition,

$$\hat{r}_i^* = \frac{1}{\hat{\xi}_1(i)}\hat{\Xi}_{0,i}, \qquad r_i = \frac{1}{\xi_1(i)}\Xi_{0,i}.$$

Let $H = \omega^{-1}O$. It follows that

$$
\begin{aligned}
\hat{r}_i^* - Hr_i &= \frac{1}{\hat{\xi}_1(i)}\hat{\Xi}_{0,i} - \frac{1}{\omega\xi_1(i)}O\Xi_{0,i} \\
&= \frac{1}{\omega\xi_1(i)}(\hat{\Xi}_{0,i} - O\Xi_{0,i}) + \Big[\frac{1}{\hat{\xi}_1(i)} - \frac{1}{\omega\xi_1(i)}\Big]\hat{\Xi}_{0,i} \\
&= \frac{1}{\omega\xi_1(i)}(\hat{\Xi}_{0,i} - O\Xi_{0,i}) - \frac{\hat{\xi}_1(i) - \omega\xi_1(i)}{\omega\xi_1(i)}\hat{r}_i^* \qquad\text{(A.8)}
\end{aligned}
$$

By Lemma 3.1, $\xi_1(i) = \|\theta\|^{-1}\theta_i \cdot \pi_i^\top D^{-1}u_1$. First, by condition (4.3), $u_1$ is a positive vector whose entries are all at the order of $1/\sqrt{K}$. Second, by condition (4.1), $D^{-1}$ is a diagonal matrix whose diagonal entries are all at the order of $\sqrt{K}$. Third, $\pi_i$ is a nonnegative vector that sums to 1. Combining the above, we obtain that

$$C^{-1}\|\theta\|^{-1}\theta_i \leq \xi_1(i) \leq C\|\theta\|^{-1}\theta_i, \qquad 1 \leq i \leq n. \qquad\text{(A.9)}$$

Similarly, by Lemma 3.1, $\Xi_{0,i} = \|\theta\|^{-1}\theta_i \cdot \pi_i^\top D^{-1}U_{2:K}$, where $U_{2:K}$ contains the second to $K$-th columns of $U$. It follows that $\|\Xi_{0,i}\| \leq C\|\theta\|^{-1}\theta_i \cdot \|\pi_i\|\|D^{-1}\|\|U_{2:K}\|$, where $\|\pi_i\| \leq 1$, $\|D^{-1}\| \leq C\sqrt{K}$, and $\|U_{2:K}\| \leq 1$. We thus have

$$\|\Xi_{0,i}\| \leq C\sqrt{K}\|\theta\|^{-1}\theta_i, \qquad 1 \leq i \leq n. \qquad\text{(A.10)}$$

We plug (A.9)-(A.10) into (A.8). It yields that

$$
\begin{aligned}
\|\hat{r}_i^* - Hr_i\| &\leq \frac{1}{\xi_1(i)}\|\hat{\Xi}_{0,i} - O\Xi_{0,i}\| + \frac{\|\hat{r}_i^*\|}{\xi_1(i)}|\hat{\xi}_1(i) - \omega\xi_1(i)| \\
&\leq \frac{C\|\theta\|}{\theta_i}\Big(\|\hat{\Xi}_{0,i} - O\Xi_{0,i}\| + \|\hat{r}_i^*\| \cdot |\hat{\xi}_1(i) - \omega\xi_1(i)|\Big). \qquad\text{(A.11)}
\end{aligned}
$$

Introduce a set

$$J = \big\{1 \leq i \leq n : |\hat{\xi}_1(i) - \omega\xi_1(i)| \leq \xi_1(i)/3, \ \|\hat{\Xi}_{0,i} - O\Xi_{0,i}\| \leq \sqrt{K}\xi_1(i)\big\}.$$

For any $i \in J$, $\|\hat{\Xi}_{i,0}\| \leq \|\Xi_{0,i}\| + \|\hat{\Xi}_{0,i} - O\Xi_{0,i}\| \leq \|\Xi_{0,i}\| + \sqrt{K}\xi_1(i)$. Combining this with (A.10)-(A.11), we find out that $\|\hat{\Xi}_{i,0}\| \leq C\sqrt{K}\xi_1(i)$. At the same time, $|\hat{\xi}_1(i)| \geq 2\xi_1(i)/3$. It follows that

$$\|\hat{r}_i^*\| \leq C\sqrt{K}, \qquad \text{for all } i \in J.$$

In particular, such $\hat{r}_i^*$ will not be truncated, i.e., $\hat{r}_i = \hat{r}_i^*$ for $i \in J$. Plugging it into (A.11) and using $\theta_i \geq \theta_{\min}$, we find that

$$\|\hat{r}_i - Hr_i\| = \|\hat{r}_i^* - Hr_i\| \leq \frac{C\|\theta\|}{\theta_{\min}}\Big(\|\hat{\Xi}_{0,i} - O\Xi_{0,i}\| + \sqrt{K}|\hat{\xi}_1(i) - \omega\xi_1(i)|\Big), \quad \text{for } i \in J.$$

We take the sum of squares over $i$ on both sides and use the universal inequality $(a + b)^2 \leq 2a^2 + 2b^2$. It follows that

$$\sum_{i \in J} \|\hat{r}_i - Hr_i\|^2 \leq \frac{C\|\theta\|^2}{\theta_{\min}^2}(\|\hat{\Xi}_0 - \Xi_0 O^\top\|_F^2 + K\|\hat{\xi}_1 - \omega\xi_1\|^2)$$

$$\leq C(\|\theta\|^2/\theta_{\min}^2) \cdot K\gamma_n^{-2}\widetilde{err}_n^2$$

$$\leq CnK\gamma_n^{-2}err_n^2, \tag{A.12}$$

where the second line is from (A.7) and the last line is from the connection between $err_n$ and $\widetilde{err}_n$. Furthermore, we consider $i \notin J$. Let

$$I_1 = \{1 \leq i \leq n : |\hat{\xi}_1(i) - \omega\xi_1(i)| > \xi_1(i)/3\}, \qquad I_2 = \{1 \leq i \leq n : \|\hat{\Xi}_{0,i} - O\Xi_{0,i}\| > \sqrt{K}\xi_1(i)\}.$$

It is seen that $J^c \subset I_1 \cup I_2$. By (A.9)-(A.10), for $i \in I_1$, $|\hat{\xi}_1(i) - \omega\xi_1(i)| > C\theta_{\min}/\|\theta\|$, and for $i \in I_2$, $\|\hat{\Xi}_{0,i} - O\Xi_{0,i}\| > C\sqrt{K}\theta_{\min}/\|\theta\|$. It follows that

$$\frac{C\theta_{\min}^2}{\|\theta\|^2}|I_1| \leq \sum_{i \in I_1} |\hat{\xi}_1(i) - \omega\xi_1(i)|^2 \leq \|\hat{\xi}_1 - \omega\xi_1\|^2, \quad \frac{K\theta_{\min}^2}{\|\theta\|^2}|I_2| \leq \sum_{i \in I_2} \|\hat{\Xi}_{0,i} - O\Xi_{0,i}\|^2 \leq \|\hat{\Xi}_0 - \Xi O^\top\|_F^2.$$

Combining them with (A.7), we immediately have

$$|I_1| \leq \frac{C\|\theta\|^2}{\theta_{\min}^2}\|\hat{\xi}_1 - \omega\xi_1\|^2 \leq Cn\gamma_n^{-2}err_n^2, \qquad |I_2| \leq \frac{C\|\theta\|^2}{K\theta_{\min}^2}\|\hat{\Xi}_0 - \hat{\Xi}O^\top\| \leq Cn\gamma_n^{-2}err_n^2.$$

It follows that

$$|J^c| \leq |I_1| + |I_2| \leq Cn\gamma_n^{-2}err_n^2. \tag{A.13}$$

So far, we have obtained two useful claims, (A.12) and (A.13), about the matrix $\hat{R}$.

Next, we study the matrix $R$. Let the matrix $U$ be the same as in Lemma 3.1. Define $V_* = [\text{diag}(u_1)]^{-1}U_{2:K}$, where $u_1$ is the first column of $U$ and $U_{2:K}$ contains the second to $K$th column of $U$. By Lemma 3.1,

$$\xi_1(i) = \|\theta\|^{-1}\theta_i \cdot d_k^{-1}u_1(k), \qquad \Xi_{0,i} = \|\theta\|^{-1}\theta_i \cdot d_k^{-1}e_k^\top U_{2:K}, \qquad \text{for all } i \in V_k.$$

It follows that

$$r_i = \frac{1}{\xi_1(i)}\Xi_{0,i} = \frac{1}{u_1(k)} \cdot e_k^\top U_{2:K} = e_k^\top V_*, \qquad \text{for all } i \in V_k.$$

Write $v_k^* = V_*^\top e_k \in \mathbb{R}^{K-1}$, for $1 \leq k \leq K$. The above imply that

$$\begin{aligned}&\{r_i\}_{i=1}^n \text{ only take } K \text{ distinct values } v_1^*, v_2^*, \ldots, v_K^*,\\&\text{and } r_i = v_k^* \text{ for all nodes in community } k, \text{ for } 1 \leq k \leq K.\end{aligned} \tag{A.14}$$

Furthermore, observing that $[\mathbf{1}_K, V_*] = [\text{diag}(u_1)]^{-1}U$, we have

$$\begin{bmatrix} 0 \\ v_k^* - v_\ell^* \end{bmatrix} = \begin{bmatrix} \mathbf{1}_K^\top(e_k - e_\ell) \\ V_*^\top(e_k - e_\ell) \end{bmatrix} = \begin{bmatrix} \mathbf{1}_K^\top \\ V_*^\top \end{bmatrix}(e_k - e_\ell) = U^\top[\text{diag}(u_1)]^{-1}(e_k - e_\ell).$$

Here, $U$ is an orthogonal matrix. Additionally, by condition (4.3), all the entries of $u_1$ are at the order of $1/\sqrt{K}$. It follows that $\|v_k^* - v_\ell^*\| = \|[\text{diag}(u_1)]^{-1}(e_k - e_\ell)\| \geq C\sqrt{K}\|e_k - e_\ell\| \geq C\sqrt{2K}$, when $k \neq \ell$. It follows that

$$\min_{1 \leq k \neq \ell \leq K} \|v_k^* - v_\ell^*\| \geq c_0\sqrt{K}, \qquad \text{for a constant } c_0 > 0. \tag{A.15}$$

Last, we analyze the output of applying k-means clustering on $\{\hat{r}_i\}_{i=1}^n$. Let $H$ be the orthogonal matrix in (A.7). In preparation, we consider placing the $K$ cluster centers at $Hv_1^*, \ldots, Hv_K^*$ and assign all nodes in community $k$ to the cluster associated with $v_k^*$. Let $RSS^*$ denote the k-means objective value for this solution. Using (A.14), we have

$$RSS^* = \sum_{k=1}^K \sum_{i \in V_k} \|\hat{r}_i - Hv_k^*\|^2 = \sum_{i=1}^n \|\hat{r}_i - Hr_i\|^2 = \sum_{i \in J} \|\hat{r}_i - Hr_i\|^2 + \sum_{i \notin J} \|\hat{r}_i - Hr_i\|^2.$$

By definition of $\hat{r}_i$, $\|\hat{r}_i\| \leq \sqrt{K\log(n)}$. At the same time, by (A.9)-(A.10), $\|Hr_i\| = \|r_i\| \leq C\sqrt{K}$. It follows that $\|\hat{r}_i - Hr_i\| \leq C\sqrt{K\log(n)}$ for all $i$. As a result,

$$RSS^* \leq \sum_{i \in J} \|\hat{r}_i - Hr_i\|^2 + |J^c| \cdot CK\log(n) \leq CnK\gamma_n^{-2}err_n^2\log(n), \tag{A.16}$$

where we have used (A.12) and (A.13) in the last inequality.

We then consider the solution that minimizes the k-means objective, where the attained objective value is denoted by $\widehat{RSS}$. Let $c_0$ be the same as in (A.15). Define

$$J^* = \{i \in J : \|\hat{r}_i - Hr_i\| \leq c_0\sqrt{K}/9\},$$

The definition of $J^*$ implies that $|J \setminus J^*| \cdot c_0^2 K/9^2 \leq \sum_{i \in J \setminus J^*} \|\hat{r}_i - Hr_i\|^2 \leq \sum_{i \in J} \|\hat{r}_i - Hr_i\|^2$, where by (A.12), $\sum_{i \in J} \|\hat{r}_i - Hr_i\|^2 \leq CnK\gamma_n^{-2}err_n^2$. It follows that

$$|J \setminus J^*| \leq Cn\gamma_n^{-2}err_n^2. \tag{A.17}$$

Additionally, by definition of $J^*$ again,

$$\|\hat{r}_i - Hv_k^*\| = \|\hat{r}_i - Hr_i\| \leq c_0\sqrt{K}/9, \qquad \text{for all } i \in V_k \cap J^*. \tag{A.18}$$

Now, suppose for some $k$, the k-means algorithm places no cluster center within a distance of $c_0\sqrt{K}/3$ to $Hv_k^*$. First, by (A.18), for any $i \in V_k \cap J^*$, the distance from $\hat{r}_i$ to the closest cluster center is $\geq c_0\sqrt{K}/3 - c_0\sqrt{K}/9 = 2c_0\sqrt{K}/9$. Second, by (A.13) and (A.17), $|V_k \setminus J^*| \leq Cn\gamma_n^{-2}err_n^2$. Since condition (4.1) implies $|V_k| \asymp n/K$, we must have $|V_k \cap J^*| \geq Cn/K$. Combining the above,

$$\widehat{RSS} \geq C(n/K) \cdot (2c_0\sqrt{K}/9)^2 \geq Cn.$$

Comparing it with (A.16) and noting that $K\gamma_n^{-2}err_n^2\log(n) = o(1)$ by (4.4), we obtain a contradiction. This means,

$$\text{for each } k, \text{ there is a cluster center within a distance } c_0\sqrt{K}/3 \text{ to } Hv_k^*. \tag{A.19}$$

By (A.15), the distance between two distinct $v_k^*$ and $v_\ell^*$ is at least $c_0\sqrt{K}$. Hence, one cluster center cannot be simultaneously within a distance $c_0\sqrt{K}/3$ to two distinct $v_k^*$. Then, for each $k$, the cluster center that satisfies (A.19) is unique. Denote this unique cluster center by $\hat{v}_k^*$. For $i \in V_k \cap J^*$, by (A.18),

$$\|\hat{r}_i - \hat{v}_k^*\| \le \|\hat{r}_i - Hv_k^*\| + \|Hv_k^* - \hat{v}_k^*\| \le c_0\sqrt{K}/9 + c_0\sqrt{K}/3 \le 4c_0\sqrt{K}/9.$$

At the same time, for $\ell \ne k$, by (A.15), the distance between $\hat{v}_\ell^*$ and $Hr_i = Hv_k^*$ is at least $c_0\sqrt{K} - c_0\sqrt{K}/3 = 2c_0\sqrt{K}/3$. It follows that

$$\|\hat{r}_i - \hat{v}_\ell^*\| \ge \|Hr_i - \hat{v}_\ell^*\| - \|\hat{r}_i - Hr_i\| \ge 2c_0\sqrt{K}/3 - c_0\sqrt{K}/9 \ge 5c_0\sqrt{K}/9.$$

By the nature of k-means, this node $i$ has to be assigned to $\hat{v}_k^*$, but not other $\hat{v}_\ell^*$. We have proved that

$$\text{for each } k, \text{ all nodes in } V_k \cap J^* \text{ are assigned to the cluster center } \hat{v}_k^*. \qquad \text{(A.20)}$$

Therefore, the clustering errors can only occur for nodes not in $J^*$. Using (A.13) and (A.17), we conclude that the number of clustering errors is bounded by

$$|(J_*)^c| \le |J^c| + |J \setminus J^*| \le Cn\gamma_n^{-2}err_n^2.$$

This proves the claim.

## A.6   Proof of Corollary 1

We study the singular values and singular vectors of $G$. In this setting, the diagonal matrix $D$ equals to $(1/\sqrt{K})I_K$, hence, $G = K^{-m/2}\mathcal{M}_1(\mathcal{P})$. The core tensor can be written as

$$\mathcal{P} = \sum_{k=1}^{K}(1-b) \times_1 e_k \times_2 \cdots \times_m e_k + b \times_1 \mathbf{1}_K \times_2 \cdots \times_m \mathbf{1}_K.$$

It follows from the matricization formula ((Kolda and Bader, 2009, Page 462)) that

$$\mathcal{M}_1(\mathcal{P}) = (1-b)\sum_{k=1}^{K} e_k (\underbrace{e_k\tilde{\otimes}\cdots\tilde{\otimes}e_k}_{m-1})^\top + b\,\mathbf{1}_K(\underbrace{\mathbf{1}_K\tilde{\otimes}\cdots\tilde{\otimes}\mathbf{1}_K}_{m-1})^\top$$

where $\tilde{\otimes}$ denotes the Kronecker product. Note that $\mathbf{1}_K\tilde{\otimes}\cdots\tilde{\otimes}\mathbf{1}_K = \mathbf{1}_{K^{m-1}}$ and $e_k\tilde{\otimes}\cdots\tilde{\otimes}e_k$ has only one nonzero entry. It follows that $(e_k\tilde{\otimes}\cdots\tilde{\otimes}e_k)^\top(e_\ell\tilde{\otimes}\cdots\tilde{\otimes}e_\ell) = 1\{k = \ell\}$ and $(e_k\tilde{\otimes}\cdots\tilde{\otimes}e_k)^\top(\mathbf{1}_K\tilde{\otimes}\cdots\tilde{\otimes}\mathbf{1}_K) = 1$. As a result,

$$
\begin{aligned}
\mathcal{M}_1(\mathcal{P})[\mathcal{M}_1(\mathcal{P})]^\top &= (1-b)^2\sum_{k=1}^{K} e_k e_k^\top + b(1-b)\sum_{k=1}^{K}(e_k\mathbf{1}_K^\top + \mathbf{1}_K e_k^\top) + b^2 K^{m-1}\mathbf{1}_K\mathbf{1}_K^\top \\
&= (1-b)^2\sum_{k=1}^{K} e_k e_k^\top + b(1-b)(\mathbf{1}_K\mathbf{1}_K^\top + \mathbf{1}_K\mathbf{1}_K^\top) + b^2 K^{m-1}\mathbf{1}_K\mathbf{1}_K^\top
\end{aligned}
$$

$$= (1-b)^2 \sum_{k=1}^{K} e_k e_k^\top + [b^2 K^{m-1} + 2b(1-b)] \mathbf{1}_K \mathbf{1}_K^\top. \tag{A.21}$$

For any $y, z > 0$, the eigenvalues of the matrix $y \sum_{k=1}^{K} e_k e_k^\top + z \mathbf{1}_K \mathbf{1}_K^\top$ are $\{y + Kz, y, \ldots, y\}$. Therefore, the eigenvalues of $\mathcal{M}_1(\mathcal{P})[\mathcal{M}_1(\mathcal{P})]^\top$ are

$$(1-b)^2 + 2Kb(1-b) + b^2 K^m, \ (1-b)^2, \ \ldots, (1-b)^2.$$

Note that $\kappa_k$'s are the singular values of $K^{-m/2}\mathcal{M}_1(P)$. It follows that

$$\kappa_1 = \sqrt{b^2 + \tfrac{2}{K^{m-1}}b(1-b) + \tfrac{1}{K^m}(1-b)^2}, \qquad \kappa_2 = \cdots = \kappa_K = \tfrac{1}{\sqrt{K^m}}(1-b). \tag{A.22}$$

Furthermore, by (A.21), the leading eigenvector of $\mathcal{M}_1(\mathcal{P})[\mathcal{M}_1(\mathcal{P})]^\top$ is $(1/\sqrt{K})\mathbf{1}_K$. Hence, the leading left singular vector of $G$ is $(1/\sqrt{K})\mathbf{1}_K$. We immediately see that the conditions (4.1)-(4.3) are all satisfied, with $\gamma_n = K^{-m/2}(1-b)$. The claim follows immediately from Theorem 1.

## A.7 Proof of Theorem 2

By the definition of $\mathcal{A}$, we write

$$\left[\mathcal{M}_1(\mathcal{A})\mathcal{M}_1^\top(\mathcal{A}) - \mathrm{diag}(\mathcal{M}_1(\mathcal{A})\mathcal{M}_1^\top(\mathcal{A}))\right]_{j_1 j_2} = [\mathcal{M}_1(\mathcal{Q})\mathcal{M}_1(\mathcal{Q})]_{j_1 j_2}$$
$$+ [\mathcal{M}_1^\top(\mathcal{A})\mathcal{M}_1(\mathcal{A}) - \mathcal{M}_1(\mathcal{Q})\mathcal{M}_1^\top(\mathcal{Q})]_{j_1 j_2}$$

for $1 \le j_1 \ne j_2 \le n$. Denote $\Delta = \mathcal{M}_1(\mathcal{A})\mathcal{M}_1^\top(\mathcal{A}) - \mathcal{M}_1(\mathcal{Q})\mathcal{M}_1^\top(\mathcal{Q})$. Then,

$$\mathcal{M}_1(\mathcal{A})\mathcal{M}_1^\top(\mathcal{A}) - \mathrm{diag}(\mathcal{M}_1(\mathcal{A})\mathcal{M}_1^\top(\mathcal{A}))$$
$$= \mathcal{M}_1(\mathcal{Q})\mathcal{M}_1^\top(\mathcal{Q}) - \mathrm{diag}(\mathcal{M}_1(\mathcal{Q})\mathcal{M}_1^\top(\mathcal{Q})) + (\Delta - \mathrm{diag}(\Delta)).$$

Clearly,
$$\sigma_K(\mathcal{M}_1(\mathcal{Q})\mathcal{M}_1^\top(\mathcal{Q})) \ge \kappa_K^2 \|\theta\|^{2m}$$

and
$$\|\mathrm{diag}(\mathcal{M}_1(\mathcal{Q})\mathcal{M}_1^\top(\mathcal{Q}))\| \le \mathcal{P}_{\max}^2 \theta_{\max}^2 \|\theta\|^{2(m-1)}.$$

It suffices to prove the bound for $\|\Delta - \mathrm{diag}(\Delta)\|$. For $1 \le j_2 \le j_3 \le \cdots \le j_m \le n$, denote

$$\xi_{j_2\cdots j_m} = \left(\mathcal{A}(1, j_2, \cdots, j_m) - \mathcal{Q}(1, j_2, \cdots, j_m), \cdots, \mathcal{A}(n, j_2, \cdots, j_m) - \mathcal{Q}(n, j_2, \cdots, j_m)\right)^\top$$

which has independent entries with zero-means. Denote also

$$q_{j_2\cdots j_m} = \left(\mathcal{Q}(1, j_2, \cdots, j_m), \mathcal{Q}(2, j_2, \cdots, j_m), \cdots, \mathcal{Q}(n, j_2, \cdots, j_m)\right)^\top.$$

Due to symmetricity, we have

$$\|\Delta - \mathrm{diag}(\Delta)\| \leq (m!) \Big\| \sum_{1 \leq j_2 \leq \cdots \leq j_m \leq n} \xi_{j_2 \cdots j_m} \xi_{j_2 \cdots j_m}^\top - \mathrm{diag}(\xi_{j_2 \cdots j_m} \xi_{j_2 \cdots j_m}^\top) \Big\|$$
$$+ 2(m!) \Big\| \sum_{1 \leq j_1 \leq \cdots \leq j_m \leq n} \xi_{j_2 \cdots j_m} q_{j_2 \cdots j_m}^\top - \mathrm{diag}(\xi_{j_2 \cdots j_m} q_{j_2 \cdots j_m}^\top) \Big\|$$

where the first term usually dominates. W.L.O.G., we only prove the upper bound for the first term. Denote the matrix $Z_{j_2 \cdots j_m} = \xi_{j_2 \cdots j_m} \xi_{j_2 \cdots j_m}^\top - \mathrm{diag}(\xi_{j_2 \cdots j_m} \xi_{j_2 \cdots j_m}^\top)$. Clearly, the matrices $Z_{j_2 \cdots j_m}$ are independent with all $1 \leq j_2 \leq \cdots \leq j_m \leq n$. It is easy to check that $\mathbb{E} Z_{j_2 \cdots j_m} Z_{j_2 \cdots j_m}$ is a diagonal matrix. Meanwhile, for $i \in [n]$,

$$\big(\mathbb{E} Z_{j_2 \cdots j_m} Z_{j_2 \cdots j_m}\big)_{ii} \leq \mathbb{E} \|\xi_{j_2 \cdots j_m}\|^2 \xi_{j_2 \cdots j_m}(i)^2 - \mathbb{E} \xi_{j_2 \cdots j_m}(i)^4 = \mathbb{E} \xi_{j_2 \cdots j_m}(i)^2 \sum_{j_1 \neq i}^n \xi_{j_2 \cdots j_m}(j_1)^2$$

$$\leq \mathcal{Q}(i, j_2, j_3, \cdots, j_m) \beta(\mathcal{Q}).$$

As a result,

$$\sum_{1 \leq j_2 \leq \cdots \leq j_m \leq n} \big(\mathbb{E} Z_{j_2 \cdots j_m} Z_{j_2 \cdots j_m}\big)_{ii} \leq \beta(\mathcal{Q}) \cdot \mathcal{P}_{\max} \theta_{\max} \|\theta\|_1^{m-1}.$$

By Bernstein inequality, for all $t > 0$,

$$\mathbb{P}\Big( \|\xi_{j_2 \cdots j_m}\|^2 \geq C_1 \beta(\mathcal{Q}) \sqrt{t} + C_2 t \Big) \leq e^{-t}$$

for some absolute constants $C_1, C_2 > 0$. It implies that the $\psi_1$-norm of $\|Z_{j_2 \cdots j_m}\|$ is bounded by

$$\big\| \|Z_{j_1 \cdots j_m}\| \big\|_{\psi_1} \leq C_1 \beta(\mathcal{Q}) + C_2$$

for some absolute constants $C_1, C_2 > 0$. By matrix Bernstein inequality (Koltchinskii et al. (2011)), with probability at least $1 - n^{-2}$,

$$\|\Delta - \mathrm{diag}(\Delta)\| \leq C_1(m!) \cdot \big( \sqrt{\beta(\mathcal{Q}) \mathcal{P}_{\max} \theta_{\max} \|\theta\|_1^{m-1} \log n} + \log n \big)$$

As a result, by Davis-Kahan theorem, with probability at least $1 - n^{-2}$,

$$\|\widehat{\Xi}^{(0)} \widehat{\Xi}^{(0)\top} - \Xi\Xi\| \leq C_1(m!) \frac{\sqrt{\beta(\mathcal{Q}) \mathcal{P}_{\max} \theta_{\max} \|\theta\|_1^{m-1} \log n} + \log n}{\kappa_K^2 \|\theta\|^{2m}} + \frac{C_2 \mathcal{P}_{\max}^2 \theta_{\max}^2}{\kappa_K^2 \|\theta\|^2}$$

for some absolute constants $C_1, C_2 > 0$.

## A.8 Proof of Proposition 4.1

Fix $v \in \mathbb{R}^K$. Write $B(v) = \mathcal{P}_* \times_3 v^\top \times_4 \cdots \times_m v^\top$, which is a $K \times K$ matrix. Suppose

$$c = \widetilde{\lambda}_{\min}(\mathcal{P}_*, v) \equiv \lambda_{\min}(B(v)) / \| \underbrace{v \otimes v \otimes \cdots \otimes v}_{m-2} \|$$

It suffices to show that

$$\|x^\top \mathcal{M}_1(\mathcal{P}_*)\| \geq c \cdot \|x\|, \qquad \text{for all } x \in \mathbb{R}^K \text{ with } \|x\| \neq 0. \tag{A.23}$$

Below, we show (A.23). Introduce a collection of $K \times K$ matrices

$$\mathcal{P}_*^{(s_3,\ldots,s_m)} = \mathcal{P}_* \times_3 e_{s_3} \times_4 \cdots \times_m e_{s_m}, \qquad \text{for all } 1 \leq e_3,\ldots,e_m \leq K.$$

We can write

$$\mathcal{M}_1(\mathcal{P}^*) = \left[ \mathcal{P}_*^{(1,1,\ldots,1)}, \ \mathcal{P}_*^{(1,2,\ldots,1)}, \ \ldots, \ \mathcal{P}_*^{(K,K,\ldots,K)} \right].$$

It follows that

$$\begin{aligned}
\|x^\top \mathcal{M}_1(\mathcal{P}^*)\|^2 &= \left\| \left[ x^\top \mathcal{P}_*^{(1,1,\ldots,1)}, \ x^\top \mathcal{P}_*^{(1,2,\ldots,1)}, \ \ldots, \ x^\top \mathcal{P}_*^{(K,K,\ldots,K)} \right] \right\|^2 \\
&= \sum_{1 \leq s_3,\ldots,s_m \leq K} \| x^\top \mathcal{P}_*^{(s_3,\ldots,s_m)} \|^2.
\end{aligned} \tag{A.24}$$

At the same time, we can write

$$\begin{aligned}
B(v) &= \mathcal{P}_* \times_3 \left( \sum_{k=1}^K v_k e_k^\top \right) \times_4 \cdots \times_m \left( \sum_{k=1}^K v_k e_k^\top \right) \\
&= \sum_{1 \leq s_3,\ldots,s_m \leq K} \mathcal{P}_* \times_3 (v_{s_3} e_{s_3}^\top) \times_4 \cdots \times_m (v_{s_m} e_{s_m}^\top) \\
&= \sum_{1 \leq s_3,\ldots,s_m \leq K} (v_{s_3} \cdots v_{s_m}) \cdot \mathcal{P}_*^{(s_3,\ldots,s_m)}.
\end{aligned}$$

It follows that

$$\begin{aligned}
\|B(v)x\|^2 &= \left\| \sum_{1 \leq s_3,\ldots,s_m \leq K} (v_{s_3} \cdots v_{s_m}) \cdot (\mathcal{P}_*^{(s_3,\ldots,s_m)} x) \right\|^2 \\
&\leq \left( \sum_{1 \leq s_3,\ldots,s_m \leq K} |v_{s_3} \cdots v_{s_m}| \cdot \|\mathcal{P}_*^{(s_3,\ldots,s_m)} x\| \right)^2 \\
&\leq \left( \sum_{1 \leq s_3,\ldots,s_m \leq K} |v_{s_3} \cdots v_{s_m}|^2 \right) \cdot \left( \sum_{1 \leq s_3,\ldots,s_m \leq K} \|\mathcal{P}_*^{(s_3,\ldots,s_m)} x\|^2 \right) \\
&\leq \| \underbrace{v \tilde\otimes v \tilde\otimes \cdots \tilde\otimes v}_{m-2} \|^2 \cdot \left( \sum_{1 \leq s_3,\ldots,s_m \leq K} \|\mathcal{P}_*^{(s_3,\ldots,s_m)} x\|^2 \right)
\end{aligned} \tag{A.25}$$

where $\tilde\otimes$ denotes the Kronecker product. Combining (A.24) and (A.25), we have

$$\|B(v)x\|^2 \leq \|x^\top \mathcal{M}_1(\mathcal{P}_*)\|^2 \cdot \| \underbrace{v \tilde\otimes v \tilde\otimes \cdots \tilde\otimes v}_{m-2} \|^2.$$

Additionally, by definition of eigenvalues,

$$\|B(v)x\|^2 \geq \lambda_{\min}^2(B(v)) \cdot \|x\|^2 = c^2 \|x\|^2 \cdot \| \underbrace{v \tilde\otimes v \tilde\otimes \cdots \tilde\otimes v}_{m-2} \|^2.$$

Then, (A.23) follows immediately.

## A.9  Proof of Theorem 3

Let $w = (w_1, \cdots, w_n) = D^{-1}\Pi^\top \Theta \eta \in S_K(\varepsilon)$ where $\eta$'s entries follows uniform distribution in $[1-\varepsilon, 1+\varepsilon]$. We write

$$\mathcal{A} \times_3 \eta^\top \times_4 \cdots \times_m \eta^\top$$
$$= \mathcal{Q} \times_3 \eta^\top \times_4 \cdots \times_m \eta^\top + (\mathcal{A} - \mathbb{E}\mathcal{A}) \times_3 \eta^\top \times_4 \cdots \times_m \eta^\top$$
$$- \operatorname{diag}(\mathcal{Q}) \times_3 \eta^\top \times_4 \cdots \times_m \eta^\top.$$

We denote $\tilde{\omega} = D^{-1}\Pi^\top \Theta \tilde{\eta} \in S_K(\varepsilon)$ so that $\tilde{\lambda}_{\min}(\mathcal{P}_*, \tilde{\omega}) \geq \tilde{\gamma}_n$. Then, by the definition of $\mathcal{P}_*$, we obtain

$$\sigma_K\big(\mathcal{Q} \times_3 \eta^\top \times_4 \cdots \times_m \eta^\top\big) = \sigma_K\big(\mathcal{P}_* \times_1 (D^{-1}\Pi^\top \Theta)^\top \times_2 (D^{-1}\Pi^\top \Theta)^\top \times_3 \omega^\top \times_4 \cdots \times_m \omega^\top\big)$$
$$\geq \sigma_K\big(\mathcal{P}_* \times_1 (D^{-1}\Pi^\top \Theta)^\top \times_2 (D^{-1}\Pi^\top \Theta)^\top \times_3 \tilde{\omega}^\top \times_4 \cdots \times_m \tilde{\omega}^\top\big)$$
$$- \sigma_1\big(\mathcal{P}_* \times_1 (D^{-1}\Pi^\top \Theta)^\top \times_2 (D^{-1}\Pi^\top \Theta)^\top (\overset{m}{\underset{3}{\mathsf{X}}} \omega^\top - \overset{m}{\underset{3}{\mathsf{X}}} \tilde{\omega}^\top)\big).$$

By the definition of $\widetilde{\gamma}_n$, we get that

$$\sigma_K\big(\mathcal{Q} \times_3 \eta^\top \times_4 \cdots \times_m \eta^\top\big) \geq \|\theta\|^2 \widetilde{\gamma}_n \|\tilde{\omega}\|^{m-2} - m \cdot \|\mathcal{M}_1(\mathcal{P}_*)\| \|\theta\|^2 \|\omega - \tilde{\omega}\| \cdot (\|\omega\| \vee \|\tilde{\omega}\|)^{m-3}.$$

Recall that $\omega - \tilde{\omega} = D^{-1}\Pi^\top \Theta(\eta - \tilde{\eta})$ implying that $\|\omega - \tilde{\omega}\| \leq \|\theta\| \|\eta - \tilde{\eta}\| \leq \|\theta\| 2\varepsilon\sqrt{n}$. Then, we obtain

$$\sigma_K\big(\mathcal{Q} \times_3 \eta^\top \times_4 \cdots \times_m \eta^\top\big) \geq \|\theta\|^2 \widetilde{\gamma}_n \|\tilde{\omega}\|^{m-2} - 2m\sqrt{n}\|\theta\|^3 \varepsilon \|\mathcal{M}_1(\mathcal{P}_*)\| \cdot (\|\omega\| \vee \|\tilde{\omega}\|)^{m-3}$$
$$\geq \|\theta\|^2 \widetilde{\gamma}_n \|\tilde{\omega}\|^{m-2} \cdot \Big(1 - \frac{2m\sqrt{n}\|\theta\|\varepsilon \|\mathcal{M}_1(\mathcal{P}_\star)\| \cdot (\|\omega\| \vee \|\tilde{\omega}\|)^{m-3}}{\widetilde{\gamma}_n \|\tilde{\omega}\|^{m-2}}\Big)$$
$$\geq \|\theta\|^2 \widetilde{\gamma}_n \|\tilde{\omega}\|^{m-2} \cdot \Big(1 - \frac{2m\varepsilon \|\mathcal{M}_1(\mathcal{P}_\star)\|}{\widetilde{\gamma}_n(1-\varepsilon)} \cdot \Big(\frac{1+\varepsilon}{1-\varepsilon}\Big)^{m-3}\Big)$$

where the last inequality is due to the definition of $\omega = D^{-1}\Pi^\top \Theta \eta$ and $\tilde{\omega} = D^{-1}\Pi^\top \Theta \tilde{\eta}$. Therefore, if

$$\frac{1}{2} \geq \frac{2m\varepsilon \|\mathcal{M}_1(\mathcal{P}_\star)\|}{\widetilde{\gamma}_n(1-\varepsilon)} \cdot \Big(\frac{1+\varepsilon}{1-\varepsilon}\Big)^{m-3}$$

, then $\sigma_K\big(\mathcal{Q} \times_3 \eta^\top \times_4 \cdots \times_m \eta^\top\big) \geq \|\theta\|^2 \widetilde{\gamma}_n \|\tilde{\omega}\|^{m-2}/2$. If $d_{\min} \asymp d_{\max} \asymp 1/\sqrt{K}$, since $\theta$ and $\tilde{\eta}$ are non-negative, then

$$\|\tilde{\omega}\|^2 = \|D^{-1}\Pi^\top \Theta \tilde{\eta}\|^2 \gtrsim K \sum_{k=1}^{K} \langle \theta_{V_k}, \tilde{\eta}_{V_k} \rangle^2 \geq \langle \theta, \tilde{\eta} \rangle^2$$

and as a result

$$\sigma_K\big(\mathcal{Q} \times_3 \eta^\top \times_4 \cdots \times_m \eta^\top\big) \gtrsim \widetilde{\gamma}_n \cdot \|\theta\|^2 \langle \theta, \tilde{\eta} \rangle^{m-2}.$$

We first bound the diagonal part. Following the same argument as the proof of Lemma 1,

$$\|\text{diag}(\mathcal{Q}) \times_3 \eta^\top \times_4 \cdots \times_m \eta^\top\| \leq m^2 \|\mathcal{D}_{\mathcal{Q}}^{(1,2)} \times_3 \eta^\top \times_4 \cdots \times_m \eta^\top\|$$
$$\leq m^2 \cdot \max_i \mathcal{Q}(i, i, :, \cdots, :) \times_3 \eta^\top \times_4 \cdots \times_m \eta^\top$$
$$\leq m^2 \mathcal{P}_{\max} \theta_{\max}^2 \langle \theta, \eta \rangle^{m-2}.$$

Now, we bound (conditioned on $\eta$) $\|(\mathcal{A} - \mathbb{E}\mathcal{A}) \times_3 \eta^\top \times_4 \cdots \times_m \eta^\top\|$. To cope with the complicated dependence of entries due to the symmetricity of $\mathcal{A}$, we write

$$\mathcal{A} = \sum_{\pi \in \text{ permutation of } [m]} \mathcal{A}_\pi$$

where $\mathcal{A}_\pi = \sum_{i_{\pi(1)} \geq i_{\pi(2)} \geq \cdots \geq i_{\pi(m)}} \mathcal{A}(i_1, \cdots, i_m)$. By definition, the entries of $\mathcal{A}_\pi$ are independent. For any permutation $\pi$, denote

$$\Delta_\pi = (\mathcal{A}_\pi - \mathbb{E}\mathcal{A}_\pi) \times_3 w^\top \times_4 \cdots \times_m w^\top$$
$$= \sum_{\substack{i_1, i_2 \geq 1 \\ i_{\pi(1)} \geq \cdots \geq i_{\pi(m)}}} e_{i_1} e_{i_2}^\top \cdot \Delta_{\pi, i_1, i_2}$$
$$= \sum_{\substack{i_1, i_2 \geq 1 \\ i_{\pi(1)} \geq \cdots \geq i_{\pi(m)}}} e_{i_1} e_{i_2}^\top \cdot \underbrace{\sum_{\substack{i_3, \cdots, i_m \\ i_{\pi(1)} \geq \cdots \geq i_{\pi(m)}}} (\mathcal{A}(i_1, \cdots, i_m) - \mathbb{E}\mathcal{A}(i_1, \cdots, i_m)) \eta_{i_3} \cdots \eta_{i_m}}_{\Delta_{\pi, i_1, i_2}}.$$

By Bernstein inequality, for all $t > 0$,

$$\mathbb{P}(|\Delta_{\pi, i_1, i_2}| \geq C_1 \sqrt{\mathcal{P}_{\max} \theta_{\max}^2 \langle \theta, \eta^2 \rangle^{m-2} t} + C_2 t) \leq e^{-t}$$

for some absolute constant $C_1, C_2 > 0$ and $\eta^2$ denotes the entry-wise square of $\eta$. Therefore, $\Delta_{\pi, i_1, i_2}$ is centered sub-exponential with $\psi_1$-norm bounded as

$$\|\Delta_{\pi, i_1, i_2}\|_{\psi_1} \leq C_1 \mathcal{P}_{\max} \theta_{\max}^2 \langle \theta, \eta \rangle^{m-2} + C_2.$$

Meanwhile, for any $i_1 \in [n]$,

$$\sum_{i_2} \mathbb{E}\Delta_{\pi, i_1, i_2}^2 \leq \sum_{i_2, \cdots, i_m} Q(i_1, \cdots, i_m) \eta_{i_3}^2 \cdots \eta_{i_m}^2$$
$$\leq \mathcal{Q}(i_1, :, \cdots, :) \times_2 \mathbf{1}_n^\top \times_3 \eta^{2\top} \times_4 \cdots \times_m \eta^{2\top} \leq \mathcal{P}_{\max} \theta_{\max} \|\theta\|_1 \langle \theta, \eta^2 \rangle^{m-2}.$$

Therefore,

$$\max \left\{ \max_{i_2} \sum_{i_1} \mathbb{E}\Delta_{\pi, i_1, i_2}^2, \max_{i_1} \sum_{i_2} \mathbb{E}\Delta_{\pi, i_1, i_2}^2 \right\} \leq C_1 \mathcal{P}_{\max} \theta_{\max} \|\theta\|_1 \langle \theta, \eta^2 \rangle^{m-2}.$$

By matrix Bernstein inequality (Tropp (2012); Koltchinskii et al. (2011); Koltchinskii and Xia (2015)) and the fact $\mathcal{A} = \sum_\pi \mathcal{A}_\pi$, with probability at least $1 - n^{-2}$,

$$\|(\mathcal{A} - \mathbb{E}\mathcal{A}) \times_3 \eta^\top \times_4 \cdots \times_m \eta^\top\| \le C_1(m!)\sqrt{\mathcal{P}_{\mathsf{max}}\theta_{\mathsf{max}}\|\theta\|_1 \langle \theta, \eta^2 \rangle^{m-2} \log n} + C_2(m!) \log n$$

Then, by Davis Kahan Theorem, for any $\eta$ and $\tilde{\eta}$, with probability at least $1 - n^{-2}$,

$$\|\widehat{\Xi}^{(0)}\widehat{\Xi}^{(0)\top} - \Xi\Xi^\top\| \le C_1(m!)\frac{\sqrt{\mathcal{P}_{\mathsf{max}}\theta_{\mathsf{max}}\|\theta\|_1 \langle \theta, \eta^2 \rangle^{m-2} \log n} + \mathcal{P}_{\mathsf{max}}\theta_{\mathsf{max}}^2 \langle \theta, \eta \rangle^{m-2} + \log n}{\widetilde{\gamma}_n\|\theta\|^2 \langle \theta, \tilde{\eta} \rangle^{m-2}}.$$

By the definition of $\eta$ and $\tilde{\eta}$, we obtain $(1-\varepsilon)\|\theta\|_1 \le \langle \theta, \eta \rangle \le (1+\varepsilon)\|\theta\|_1$ and $\langle \theta, \eta^2 \rangle \le (1+\varepsilon)^2\|\theta\|_1$. As a result, with probability at least $1 - n^{-2}$,

$$\|\widehat{\Xi}^{(0)}\widehat{\Xi}^{(0)\top} - \Xi\Xi^\top\| \le C_1\Big(\frac{1+\varepsilon}{1-\varepsilon}\Big)^m (m!)\frac{\sqrt{\theta_{\mathsf{max}}\|\theta\|_1^{m-1} \log n} + \log n + \theta_{\mathsf{max}}^2\|\theta\|_1^{m-2}}{\widetilde{\gamma}_n\|\theta\|^2\|\theta\|_1^{m-2}}$$

for some absolute constant $C_1 > 0$.

# B Proofs of tensor concentration and power iterations

## B.1 Proof of Theorem 5

Let $\mathcal{R} \in \{+1, -1\}^{n \times \cdots \times n}$ be a $m$-th order symmetric random tensor where each entry is a Rademacher random variable, i.e.,

$$\mathbb{P}(\mathcal{R}(i_1, \cdots, i_m) = +1) = \mathbb{P}(\mathcal{R}(i_1, \cdots, i_m) = -1) = 1/2.$$

Let $\odot$ be the Hadamard product of two tensors, i.e.,

$$(\mathcal{R} \odot \mathcal{A})(i_1, \cdots, i_m) = \mathcal{R}(i_1, \cdots, i_m)\mathcal{A}(i_1, \cdots, i_m), \quad \forall\, i_j \in [n], j \in [m].$$

Recall the definition of incoherent tensor operator norm:

$$\|\mathcal{A} - \mathbb{E}\mathcal{A}\|_\delta = \sup_{\mathcal{X} \in \mathcal{U}(\delta)} \langle \mathcal{X}, \mathcal{A} - \mathbb{E}\mathcal{A} \rangle.$$

By a standard symmetrization argument (see, e.g., Yuan and Zhang (2016) Giné and Zinn (1984)), we obtain

$$\mathbb{P}\Big\{\|\mathcal{A} - \mathbb{E}\mathcal{A}\|_\delta \ge 3t\Big\} \le \max_{u_1 \otimes \cdots \otimes u_m \in \mathcal{U}(\delta)} \mathbb{P}\Big\{\langle \mathcal{A} - \mathbb{E}\mathcal{A}, u_1 \otimes \cdots \otimes u_m \rangle \ge t\Big\} + 4\mathbb{P}\Big\{\|\mathcal{R} \odot \mathcal{A}\|_\delta \ge t\Big\}$$

for any $t > 0$. We begin with the upper bound for $\langle \mathcal{A} - \mathbb{E}\mathcal{A}, u_1 \otimes \cdots \otimes u_m \rangle$. For any $u_1 \otimes \cdots \otimes u_m \in \mathcal{U}(\delta)$, write

$$\langle \mathcal{A}, u_1 \otimes \cdots \otimes u_m \rangle = \sum_{(i_1, \cdots, i_m) \in \mathfrak{P}(n,m)} \mathcal{A}(i_1, \cdots, i_m) \sum_{(i_1', \cdots, i_m') \in \mathfrak{P}(i_1, \cdots, i_m)} u_1(i_1')u_2(i_2')\cdots u_m(i_m')$$

where the set $\mathfrak{P}(n,m)$ is the collection of all subsets of $[n]$ with cardinality $m$ so that $|\mathfrak{P}(n,m)| = \binom{n}{m}$ and $\mathfrak{P}(i_1, \cdot, i_m)$ denotes the set of all the permutations of $(i_1, \cdots, i_m)$, i.e.,

$$\mathfrak{P}(i_1, \cdots, i_m) = \{(i_{\pi(1)}, i_{\pi(2)}, \cdots, i_{\pi(m)}) : \pi \text{ is a permutation of } (1, \cdots, m)\}.$$

Since $u_1 \otimes \cdots \otimes u_m \in \mathcal{U}(\delta)$, we get

$$\Big| \sum_{(i'_1, \cdots, i'_m) \in \mathfrak{P}(i_1, \cdots, i_m)} u_1(i'_1) u_2(i'_2) \cdots u_m(i'_m) \Big| \le (m!) \cdot \delta^{m-2}.$$

For $u_1 \otimes \cdots \otimes u_m \in \mathcal{U}_{1,2}(\delta)$, if $m \ge 3$, then

$$\begin{aligned}
&\mathrm{Var}\big(\langle \mathcal{A}, u_1 \otimes \cdots \otimes u_m \rangle\big) \\
&\le \sum_{(i_1, \cdots, i_m) \in \mathfrak{P}(n,m)} \mathbb{E}\mathcal{A}(i_1, \cdots, i_m) \Big( \sum_{(i'_1, \cdots, i'_m) \in \mathfrak{P}(i_1, \cdots, i_m)} u_1(i'_1) u_2(i'_2) \cdots u_m(i'_m) \Big)^2 \\
&\le (m!) \cdot \sum_{(i_1, \cdots, i_m) \in \mathfrak{P}(n,m)} \mathbb{E}\mathcal{A}(i_1, \cdots, i_m) \sum_{(i'_1, \cdots, i'_m) \in \mathfrak{P}(i_1, \cdots, i_m)} u_1(i'_1)^2 u_2(i'_2)^2 \cdots u_m(i'_m)^2 \\
&\le (m!) \cdot \sum_{i_1 \ne \cdots \ne i_m \ge 1}^{n} \mathbb{E}\mathcal{A}(i_1, \cdots, i_m) u_1(i_1)^2 u_2(i_2)^2 \cdots u_m(i_m)^2 \\
&\le (m!)^2 \cdot \sum_{i_m=1}^{n} u_m^2(i_m) \sum_{i_1 \ge \cdots \ge i_{m-1} \ge 1}^{n} \mathbb{E}\mathcal{A}(i_1, \cdots, i_m) u_1(i_1)^2 \cdots u_{m-1}(i_{m-1})^2 \\
&\le (m!)^2 \Big[ \max_{i_1 \ge \cdots \ge i_{m-1} \in [n]} \sum_{i_m=1}^{n} u_m(i_m)^2 \mathbb{E}\mathcal{A}(i_1, \cdots, i_m) \Big] \cdot \sum_{i_1, \cdots, i_{m-1} \ge 1}^{n} u_1(i_1)^2 \cdots u_{m-1}(i_{m-1})^2 \\
&\le (m!)^2 \delta^2 \beta(\mathcal{Q})
\end{aligned}$$

where $\beta(\mathcal{Q})$ is defined by

$$\beta(\mathcal{Q}) := \max_{i_1 \ge \cdots \ge i_{m-1} \ge 1} \sum_{i_m=1}^{n} \mathcal{Q}(i_1, \cdots, i_m).$$

By Bernstein inequality, for any $t > 0$,

$$\max_{u_1 \otimes \cdots \otimes u_m \in \mathcal{U}(\delta)} \mathbb{P}\Big\{ \langle \mathcal{A} - \mathbb{E}\mathcal{A}, u_1 \otimes \cdots \otimes u_k \rangle \ge t \Big\} \le \exp\left( \frac{-t^2}{4(m!)^2 \delta^2 \beta(\mathcal{Q})} \right) + \exp\left( \frac{-3t}{(m!) \cdot \delta^{m-2}} \right).$$

(B.1)

We then prove the bound for $\mathbb{P}\{\|\mathcal{R} \odot \mathcal{A}\|_\delta \ge t\}$. Recall the definition of $\|\cdot\|_\delta$ and the symmetricity of $\mathcal{A}$, we write

$$\|\mathcal{R} \odot \mathcal{A}\|_\delta = \sup_{u_1 \otimes \cdots \otimes u_m \in \mathcal{U}_{1,2}(\delta)} \langle \mathcal{R} \odot \mathcal{A}, u_1 \otimes \cdots \otimes u_m \rangle.$$

Therefore,

$$\mathbb{P}\{\|\mathcal{R} \odot \mathcal{A}\|_\delta \ge t\} = \mathbb{P}\Big\{ \sup_{u_1 \otimes \cdots \otimes u_m \in \mathcal{U}_{1,2}(\delta)} \langle \mathcal{R} \odot \mathcal{A}, u_1 \otimes \cdots \otimes u_m \rangle \ge t \Big\}.$$

We bound

$$\mathbb{P}\Big\{ \sup_{u_1 \otimes \cdots \otimes u_m \in \mathcal{U}_{1,2}(\delta)} \langle \mathcal{R} \odot \mathcal{A}, u_1 \otimes \cdots \otimes u_m \rangle \geq t \Big\}.$$

We now discretize $\mathcal{U}_{1,2}(\delta)$ as

$$\overline{\mathcal{U}}_{1,2}(\delta) := \Big\{ u_1 \otimes \cdots \otimes u_m \in \mathcal{U}_{1,2}(\delta) : \|u_j\| \leq 1, u_j \in \big\{ \pm 2^{i_j/2}/2^{\lceil \log \sqrt{2n} \rceil}, i_j = 0, \cdots, p_j \big\}^n \Big\}$$

where $p_j = p^\star = \lceil \log(n) - 1 \rceil$ for $j = 1, 2$ and $p_j = p_\star = \lceil \log(\delta^2 n) - 1 \rceil$ for $j = 3, \cdots, m$. For any $u_1 \otimes \cdots \otimes u_m \in \overline{\mathcal{U}}_{1,2}(\delta)$, we have

$$u_1, u_2 \in \big\{ \pm 1, \pm 2^{-1/2}, \cdots, \pm 2^{-p^\star/2} \big\}^n$$

and also

$$u_3, u_4, \cdots, u_m \in \big\{ \pm 2^{-\lceil \log(\delta^{-2}) \rceil/2}, \pm 2^{-(1+\lceil \log(\delta^{-2}) \rceil)/2}, \cdots, \pm 2^{-p^\star/2} \big\}.$$

By (Yuan and Zhang, 2017, Lemma 1), we have

$$\mathbb{P}\Big\{ \sup_{u_1 \otimes \cdots \otimes u_m \in \mathcal{U}_{1,2}(\delta)} \langle \mathcal{R} \odot \mathcal{A}, u_1 \otimes \cdots \otimes u_m \rangle \geq t \Big\}$$

$$\leq \mathbb{P}\Big\{ \sup_{u_1 \otimes \cdots \otimes u_m \in \overline{\mathcal{U}}_{1,2}(\delta)} \langle \mathcal{R} \odot \mathcal{A}, u_1 \otimes \cdots \otimes u_m \rangle \geq 2^{-m-1} \cdot t \Big\}$$

where the entropy of $\overline{U}_{1,2}(\delta)$ plays the key role. The main idea of proof id entropy and variance trade-off. A simple fact (Xia et al. (2021)) is

$$\mathrm{Card}(\overline{\mathcal{U}}_{1,2}(\delta)) \leq \exp(C_2 mn)$$

for some absolute constant $C_2 > 0$. Finer characterization of $\overline{U}_{1,2}(\delta)$ is needed. The idea is similar to Yuan and Zhang (2017). For any $U = u_1 \otimes \cdots \otimes u_m \in \overline{U}_{1,2}(\delta)$, define

$$A_p(U) = \big\{ (i_1, i_2) : |u_1(i_1)u_2(i_2)| = 2^{-p/2} \big\}, \quad \forall \, p = 0, 1, \cdots, 2p^\star$$

$$B_{p,q}(U) = \big\{ (i_3, i_4, \cdots, i_m) : (i_1, i_2) \in A_p(U), (i_3, \cdots, i_m) \in \Omega, |u_3(i_3) \cdots u_m(i_m) = 2^{-q/2}| \big\}$$

where $\Omega = \big\{ (i_1, \cdots, i_m) : \mathcal{A}(i_1, \cdots, i_m) = 1 \big\}$, i.e., the position of all observed edges. The positive integer $q = (m-2)\lceil \log \delta^{-2} \rceil, \cdots, (m-2)p^\star$.

The main strategy is to exploit $\Omega$'s sparsity and investigate the effective entropy of $\overline{U}_{1,2}(\delta)$. For notational simplicity, we now write

$$U_{1,2} = u_1 \otimes u_2 \quad \text{and} \quad U_{3,\cdots,m} = u_3 \otimes \cdots \otimes u_m.$$

Let $p_{1,2} \geq 0$ be an integer to be determined later. Then, for $U \in \overline{U}_{1,2}(\delta)$, we write

$$\langle \mathcal{R} \odot \mathcal{A}, U \rangle = \langle \mathcal{R} \odot \mathcal{A}, \mathcal{P}_{S_{1,2}}(U_{1,2}) \otimes U_{3,\cdots,m} \rangle$$

$$+ \sum_{0 \leq p \leq p_{1,2}} \sum_{q=(m-2)\lceil \log \delta^{-2} \rceil}^{(m-2)p^\star} \langle \mathcal{R} \odot \mathcal{A}, (\mathcal{P}_{A_p} U_{1,2}) \otimes (\mathcal{P}_{B_{p,q}} U_{3,\cdots,m}) \rangle$$

where we omit the dependence of $A_p$ and $B_{p,q}$ on $U$. Here the set $S_{1,2}$ is defined by (for each $U \in \overline{U}_{1,2}(\delta)$)

$$S_{1,2} = \{(i_1, i_2) : |U_{1,2}(i_1, i_2)| \leq 2^{-p_{1,2}/2 - 1/2}\}.$$

Clearly, $p_{1,2} m p^\star \leq m^2 \log^2 n$, then

$$\mathbb{P}\bigg(\sum_{0 \leq p \leq p_{1,2}} \sum_{q=(m-2)\lceil \log \delta^{-2}\rceil}^{(m-2)p^\star} \max_{u_1 \otimes \cdots u_m \in \overline{U}_{1,2}(\delta)} \big\langle \mathcal{R} \odot \mathcal{A}, (\mathcal{P}_{A_p} U_{1,2}) \otimes (\mathcal{P}_{B_{p,q}} U_{3,\cdots,m}) \big\rangle \geq 2^{-(m+1)} t/2 \bigg)$$

$$\leq \sum_{0 \leq p \leq p_{1,2}} \sum_{q=(m-2)\lceil \log \delta^{-2}\rceil}^{(m-2)p^\star} \mathbb{P}\bigg( \max_{u_1 \otimes \cdots u_m \in \overline{U}_{1,2}(\delta)} \big\langle \mathcal{R} \odot \mathcal{A}, (\mathcal{P}_{A_p} U_{1,2}) \otimes (\mathcal{P}_{B_{p,q}} U_{3,\cdots,m}) \big\rangle \geq 2^{-(m+1)} t/(2m^2 \log^2 n) \bigg)$$

$$\leq (m-2) p^\star p_{1,2} \max_{0 \leq p \leq p_{1,2}, (m-2)\lceil \log \delta^{-2}\rceil \leq q \leq (m-2)p^\star}$$

$$\times \mathbb{P}\bigg( \max_{u_1 \otimes \cdots u_m \in \overline{U}_{1,2}(\delta)} \big\langle \mathcal{R} \odot \mathcal{A}, (\mathcal{P}_{A_p} U_{1,2}) \otimes (\mathcal{P}_{B_{p,q}} U_{3,\cdots,m}) \big\rangle \geq 2^{-(m+1)} t/(2m^2 \log^2 n) \bigg)$$

Define the aspect ratio of $\Omega$ with respect to $(1,2)$-dimension as

$$\nu_{1,2}(\Omega) = \max_{i_1 \in [n], i_2 \in [n]} \mathrm{Card}\big(\{(i_1, \cdots, i_m) \in \Omega : i_3, \cdots, i_m \in [n]\}\big).$$

By Chernoff bound, we get

$$\mathbb{P}\Big(\nu_{1,2}(\Omega) \geq 13\big(\mathcal{P}_{\mathsf{max}} \cdot \theta_{\mathsf{max}}^2 \|\theta\|_1^{m-2} + \log n\big)\Big) \leq n^{-2} \tag{B.2}$$

where we used the fact that each $\mathcal{A}(i_1, \cdots, i_m)$ is a Bernoulli random variable and

$$\mathbb{P}\big(\mathcal{A}(i_1, \cdots, i_m) = 1\big) \leq \mathcal{P}_{\mathsf{max}} \theta_{i_1} \theta_{i_2} \cdots \theta_{i_m}.$$

Denote

$$\nu_{1,2}^\star = 13\big(\mathcal{P}_{\mathsf{max}} \theta_{\mathsf{max}}^2 \|\theta\|_1^{m-2} + \log n\big).$$

Denote $\mathcal{E}_0$ the event of (B.2) under which $\nu_{1,2}(\Omega) \leq \nu_{1,2}^\star$ and $|B_{p,q}| \leq \nu_{1,2}^\star |A_p|$. By the definitions of $A_p$, we have

$$|A_p| \leq 2^p.$$

Meanwhile, $\|U_{3,\cdots,m}\|_{\mathsf{max}} \leq 2^{-q/2} \leq \delta^{m-2}$. Regarding to the sparsity levels of the hypergraph networks, we consider two cases.

*Case 1*: $\mathcal{P}_{\mathsf{max}} \theta_{\mathsf{max}}^2 \|\theta\|_1^{m-2} \geq \log n$ so that $\nu_{1,2}^\star = 26 \mathcal{P}_{\mathsf{max}} \theta_{\mathsf{max}}^2 \|\theta\|_1^{m-2}$, i.e., the hypergraph is not extremely sparse. This scenario is easy. We can simply decouple the dependence between $B_{p,q}$ and $A_p$ (ignore $\Omega$) and define the set

$$\mathfrak{B}_p(\ell) :=$$
$$\big\{V = (\mathcal{P}_{A_p} U_{1,2}) \otimes (\mathcal{P}_B U_{3,\cdots,m}) : |A_p| \leq 2^{p-\ell}, |B| \leq \nu_{1,2}^\star |A_p|, U_{1,2} \otimes U_{3,\cdots,m} \in \overline{U}_{1,2}(\delta)\big\}$$

which does not depend on $q$. Clearly, for any $U \in \overline{U}_{1,2}(\delta)$ and on event $\mathcal{E}_0$,

$$(\mathcal{P}_{A_p} U_{1,2}) \otimes (\mathcal{P}_{B_{p,q}} U_{3,\cdots,m}) \in \mathfrak{B}_p(\ell).$$

For any $p, q$, we have

$$\mathbb{P}\Big(\max_{u_1 \otimes \cdots u_m \in \overline{U}_{1,2}(\delta)} \langle \mathcal{R} \odot \mathcal{A}, (\mathcal{P}_{A_p}U_{1,2}) \otimes (\mathcal{P}_{B_{p,q}}U_{3,\cdots,m})\rangle \geq 2^{-(m+1)}t/(2m^2 \log^2 n)\Big)$$

$$\leq \max_{0 \leq \ell \leq p} \mathbb{P}\Big(\max_{V \in \mathfrak{B}_p(\ell)} \langle \mathcal{R} \odot \mathcal{A}, V\rangle \geq 2^{-(m+1)}t/(2m^2 \log^2 n)\Big).$$

As shown in [Yuan and Zhang (2017)](#), $\log\big|\mathfrak{B}_p(\ell)\big| \leq 2^{(p-\ell)/2}m\sqrt{4\nu^\star_{1,2}n\log n}$.

To prove the bound for $\mathbb{P}\{\max_{V \in \mathfrak{B}_p(\ell)} \langle \mathcal{R} \odot \mathcal{A}, V\rangle \geq 2^{-(m+1)/2}t/2\}$. Write

$$\langle \mathcal{R} \odot \mathcal{A}, V\rangle = \sum_{(i_1,\cdots,i_m)\in\mathfrak{P}(n,m)} \mathcal{R}(i_1,\cdots,i_m)\mathcal{A}(i_1,\cdots,i_m) \sum_{(i'_1,\cdots,i'_m)\in\mathfrak{P}(i_1,\cdots,i_m)} V(i'_1,\cdots,i'_m)$$

which is a sum of zero-mean random variables. For $V \in \mathfrak{B}_{p,q}(\ell)$,

$$\Big| \mathcal{R}(i_1,\cdots,i_m)\mathcal{A}(i_1,\cdots,i_m) \sum_{(i'_1,\cdots,i'_m)\in\mathfrak{P}(i_1,\cdots,i_m)} V(i'_1,\cdots,i'_m)\Big|$$

$$\leq (m!)\|V\|_{\max} \leq (m!)2^{-p/2}\delta^{m-2}.$$

Moreover, if $m \geq 3$ and $V = V_{1,\cdots,m-1} \otimes V_m$, then

$$\sum_{(i_1,\cdots,i_m)\in\mathfrak{P}(n,m)} \mathbb{E}\mathcal{A}(i_1,\cdots,i_m)\Big(\sum_{(i'_1,\cdots,i'_m)\in\mathfrak{P}(i_1,\cdots,i_m)} V(i'_1,\cdots,i'_m)\Big)^2$$

$$\leq (m!) \sum_{(i_1,\cdots,i_m)\in\mathfrak{P}(n,m)} \mathbb{E}\mathcal{A}(i_1,\cdots,i_m) \sum_{(i'_1,\cdots,i'_m)\in\mathfrak{P}(i_1,\cdots,i_m)} V(i'_1,\cdots,i'_m)^2$$

$$\leq (m!) \sum_{i_1\neq\cdots\neq i_m} \mathbb{E}\mathcal{A}(i_1,\cdots,i_m)V(i_1,\cdots,i_m)^2$$

$$\leq (m!)^2 \sum_{i_m=1}^n V_m(i_m)^2 \sum_{i_1\geq\cdots\geq i_{m-1}\geq 1} \mathbb{E}\mathcal{A}(i_1,\cdots,i_m)V_{1,\cdots,m-1}(i_1,\cdots,i_{m-1})^2$$

$$\leq (m!)^2\delta^2\beta(\mathcal{Q})\|V_{1,\cdots,m-1}\|_F^2 \leq 2^{-\ell}(m!)^2\delta^2\beta(\mathcal{Q}).$$

By Bernstein inequality, $\log\big|\mathfrak{B}_p(\ell)\big| \leq \min\{2^{(p-\ell)/2}m\sqrt{4\nu^\star_{1,2}n\log n}, 4mn\}$ and the union bound, we get

$$\mathbb{P}\Big(\max_{V \in \mathfrak{B}_p(\ell)} \langle \mathcal{R} \odot \mathcal{A}, V\rangle \geq 2^{-(m+1)}t/2\Big) \leq \exp\Big(4mn - \frac{2^{\ell-2m-2}t^2}{4(m!)^2\delta^2\beta(\mathcal{Q})m^4\log^4 n}\Big)$$

$$+ \exp\Big(2^{(p-\ell)/2}m\sqrt{4\nu^\star_{1,2}n\log n} - \frac{(3/4)2^{(p-2m-2)/2}t}{2(m!)\delta^{m-2}m^2\log^2 n}\Big).$$

If

$$t \geq 16m^3 2^m(m!)\sqrt{n\delta^2\beta(\mathcal{Q})}\log^2 n$$

, then $2^{\ell-2m-2}t^2/(4m!\delta^2\beta(Q)m^4\log^4 n) \geq 8mn$ and as a result

$$\exp\Big(4mn - \frac{2^{\ell-2m-2}t^2}{4(m!)\delta^2\beta(\mathcal{Q})m^4\log^4 n}\Big) \leq \exp\Big(-\frac{2^{\ell-2m-2}t^2}{8(m!)\delta^2\beta(\mathcal{Q})m^4\log^4 n}\Big) \leq \exp\Big(-\frac{2^{-2m-2}t^2}{8(m!)\delta^2\beta(\mathcal{Q})m^4\log^4 n}\Big).$$

Similarly, if

$$t \geq \frac{32m^3}{3}(m!)2^m \delta^{m-2}\sqrt{\nu_{1,2}^\star n \log n} \log^2 n$$

, then $3 \cdot 2^{-(m+1)}t/(8m!\delta^{m-2}m^2 \log^2 n) \geq 4m\sqrt{\nu_{1,2}^\star n \log n}$ and as a result,

$$\exp\left(2^{(p-\ell)/2}m\sqrt{4(\nu_{1,2}^\star \wedge n)n \log n} - \frac{(3/4)2^{(p-2m-2)/2}t}{2(m!)\delta^{m-2}m^2 \log^2 n}\right) \leq \exp\left(-\frac{(3/4)2^{(p-2m-2)/2}t}{4(m!)\delta^{m-2}m^2 \log^2 n}\right)$$
$$\leq \exp\left(-\frac{3}{16} \cdot \frac{2^{-(m+1)}t}{(m!)\delta^{m-2}m^2 \log^2 n}\right).$$

Therefore, we conclude that if (recall that $\mathcal{P}_{\mathsf{max}} \cdot \theta_{\mathsf{max}}^2 \|\theta\|_1^{m-2} \geq \log n$)

$$t \geq \max\left\{16m^3 2^m \sqrt{(m!)n\delta^2\beta(\mathcal{Q})} \log^2 n, \frac{128m^3}{3}(m!)2^m \delta^{m-2}\sqrt{\mathcal{P}_{\mathsf{max}}\theta_{\mathsf{max}}^2\|\theta\|_1^{m-2}n \log^5 n}\right\}$$

, then for any $\ell$,

$$\mathbb{P}\left(\max_{\mathbf{V}\in\mathfrak{B}_p(\ell)}\langle \mathcal{R}\odot\mathcal{A}, \mathbf{V}\rangle \geq 2^{-(m+1)}t/(2m^2 \log^2 n)\right) \leq \exp\left(-\frac{2^{-2m-2}t^2}{8(m!)\delta^2\beta(\mathcal{Q})m^4 \log^4 n}\right)$$
$$+ \exp\left(-\frac{3}{16} \cdot \frac{2^{-(m+1)}t}{(m!)\delta^{m-2}m^2 \log^2 n}\right)$$

which proves the desired bound in *Case 1*.

*Case 2*: $\mathcal{P}_{\mathsf{max}}\theta_{\mathsf{max}}^2\|\theta\|_1^{m-2} < \log n$ so that $\nu_{1,2}^\star = 26 \log n$, the hypergraph is extremely sparse. In this case, the above entropy control is not sharp enough to control

$$\mathbb{P}\left(\max_{u_1\otimes\cdots u_m\in\overline{U}_{1,2}(\delta)}\langle \mathcal{R}\odot\mathcal{A}, (\mathcal{P}_{A_p}U_{1,2})\otimes(\mathcal{P}_{B_{p,q}}U_{3,\cdots,m})\rangle \geq 2^{-(m+1)}t/(2m^2 \log^2 n)\right).$$

To this end, for $0 \leq \ell \leq p$, we define ($\Omega$ related)

$$\mathfrak{B}_{\Omega,p,q}(\ell) = \left\{V = \mathcal{P}_{A_p}(U_{1,2})\otimes(\mathcal{P}_{B_{p,q}}U_{3,\cdots,m}) : |A_p| \leq 2^{p-\ell}, |B_{p,q}| \leq \nu_{1,2}(\Omega)|A_p|\right\}.$$

To this end, we need to define the aspect ratio on each fiber (see, e.g., Xia et al. (2021)) of $\mathcal{P}_{A_p}(U_{1,2})$ and $\mathcal{P}_{B_{p,q}}U_{3,\cdots,m}$. More specifically, define

$$\nu_1(\Omega) := \max_{i_1\in[n]}\text{Card}(\{(i_1,\cdots,i_m)\} \in \Omega : i_2,\cdots,i_m \in [n]\}).$$

Similarly, by Chernoff bound, we have

$$\mathbb{P}\left(\nu_1(\Omega) \geq 13(\mathcal{P}_{\mathsf{max}}\theta_{\mathsf{max}}\|\theta\|_1^{m-1} + \log n)\right) \leq n^{-2}.$$

We then denote $\nu_1^\star = 26\mathcal{P}_{\mathsf{max}}\theta_{\mathsf{max}}\|\theta\|_1^{m-1}$ since we assume $\mathcal{P}_{\mathsf{max}}\theta_{\mathsf{max}}\|\theta\|_1^{m-1} \geq \log n$. We denote the above event by $\mathcal{E}_1$ so that $\mathbb{P}(\mathcal{E}_1) \geq 1 - n^{-2}$. Basically, under the event $\mathcal{E}_0 \cap \mathcal{E}_1$, then for any $U \in \mathfrak{B}_{\Omega,p,q}(\ell)$, we have $|A_p| \leq 2^{p-\ell}$ and $|B_{p,q}| \leq 26 \log n \cdot |A_p|$. In addition, it suffices to consider

those $A_p$ and $B_{p,q}$ satisfying the aspect ratio $\nu_1^\star$, i.e.,

$$\nu(\mathcal{P}_{A_p}U_{1,2}) := \max_{k=1,2}\max_{i_k\in[n]}\left\|(\mathcal{P}_{A_p}U_{1,2})_{i_k,:}\right\|_{\ell_0} \leq \nu_1^\star$$

and similarly,

$$\nu(\mathcal{P}_{B_{p,q}}U_{3,\cdots,m}) := \max_{k=3,\cdots,m}\max_{i_j\in[n],3\leq j\leq m,j\neq k}\left\|(\mathcal{P}_{B_{p,q}}U_{3,\cdots,m})_{i_j,j\neq k,:}\right\|_{\ell_0} \leq \nu_1^\star$$

which allows us to obtain sharper characterization of the entropy of $\mathfrak{B}_{\Omega,p,q}(\ell)$. Indeed, we define (which overrides the previous definition of $\mathfrak{B}_{\Omega,p,q}(\ell)$)

$$\mathfrak{B}_{\Omega,p,q}(\ell) = \Big\{V = \mathcal{P}_{A_p}(U_{1,2}) \otimes (\mathcal{P}_{B_{p,q}}U_{3,\cdots,m}) : |A_p| \leq 2^{p-\ell}, |B_{p,q}| \leq \nu_{1,2}(\Omega)|A_p|$$
$$, \nu(\mathcal{P}_{A_p}U_{1,2}) \leq \nu_1(\Omega), \nu(\mathcal{P}_{B_{p,q}}U_{3,\cdots,m}) \leq \nu_1(\Omega)\Big\}.$$

and

$$\mathfrak{B}_{\Omega,p}^{(1,2)}(\ell) := \{\mathcal{P}_{A_p}(U_{1,2}) : |A_p| \leq 2^{p-\ell}, U \in \overline{U}_{1,2}(\delta), \nu(\mathcal{P}_{A_p}U_{1,2}) \leq \nu_1(\Omega)\}$$

and

$$\mathfrak{B}_{\Omega,p,q}^{(3,\cdots,m)}(\ell) := \{\mathcal{P}_{B_{p,q}}(U_{3,\cdots,m}) : |B_{p,q}| \leq \nu_{1,2}(\Omega) \cdot 2^{p-\ell}, U \in \overline{U}_{1,2}(\delta), \nu(\mathcal{P}_{B_{p,q}}U_{3,\cdots,m}) \leq \nu_1(\Omega)\}.$$

Then, $\mathfrak{B}_{\Omega,p,q}(\ell) \subset \mathfrak{B}_{\Omega,p}^{(1,2)}(\ell) \otimes \mathfrak{B}_{\Omega,p,q}^{(3,\cdots,m)}(\ell)$ implying that

$$\mathfrak{B}_{p,q}^\star(\ell) := \bigcup_{\substack{\Omega:\nu_1(\Omega)\leq\nu_1^\star \\ \nu_{1,2}(\Omega)\leq\nu_{1,2}^\star}} \mathfrak{B}_{\Omega,p,q}(\ell) \subset \underbrace{\bigcup_{\substack{\Omega:\nu_1(\Omega)\leq\nu_1^\star \\ \nu_{1,2}(\Omega)\leq\nu_{1,2}^\star}} \mathfrak{B}_{\Omega,p}^{(1,2)}(\ell)}_{\mathfrak{B}_p^{(1,2)\star}} \bigotimes \underbrace{\bigcup_{\substack{\Omega:\nu_1(\Omega)\leq\nu_1^\star \\ \nu_{1,2}(\Omega)\leq\nu_{1,2}^\star}} \mathfrak{B}_{\Omega,p,q}^{(3,\cdots,m)}(\ell)}_{\mathfrak{B}_{p,q}^{(3,\cdots,m)\star}}.$$

As shown in Yuan and Zhang (2016) and Xia et al. (2021), for any $p,q$ and under event $\mathcal{E}_0 \cap \mathcal{E}_1$, it holds that

$$\max_{U\in\overline{U}_{1,2}(\delta)}\left\langle\mathcal{R}\odot\mathcal{A}, (\mathcal{P}_{A_p}U_{1,2})\otimes(\mathcal{P}_{B_{p,q}}U_{3,\cdots,m})\right\rangle \leq \max_{\ell\leq p}\max_{\mathbf{V}\in\mathfrak{B}_{p,q}^\star(\ell)}\left\langle\mathcal{R}\odot\mathcal{A},\mathbf{V}\right\rangle.$$

It suffices to investigate the entropy of $\mathfrak{B}_{p,q}^\star(\ell)$. Clearly,

$$\log\mathrm{Card}\big(\mathfrak{B}_{p,q}^\star(\ell)\big) \leq \log\mathrm{Card}\big(\mathfrak{B}_p^{(1,2)\star}\big) + \log\mathrm{Card}\big(\mathfrak{B}_{p,q}^{(3,\cdots,m)\star}\big).$$

Basically, $\mathfrak{B}_p^{(1,2)\star}$ is the set that all non-zero entries are $2^{-p/2}$ and there are $2^{p-\ell}$ non-zero entries and its aspect ratio is bounded by $\nu_1^\star$. By (Xia et al., 2021, Lemma 1),

$$\log\mathrm{Card}\big(\mathfrak{B}_p^{(1,2)\star}\big) \leq (p-\ell)p^m\log 4 + 2m^3 p^m\sqrt{\nu_1^\star 2^{p-\ell}}\log n$$

and similarly,

$$\log\mathrm{Card}\big(\mathfrak{B}_{p,q}^{(3,\cdots,m)\star}\big) \leq (p-\ell)q^m\log n + 48m^3 q^m\sqrt{\nu_1^\star 2^{p-\ell}\log n}\log n$$

where we used the fact that each element in $\mathfrak{B}_{p,q}^{(3,\cdots,m)\star}$ has at most $2^{p-\ell}\cdot 26\log n$ non-zero entries and its aspect ratio is bounded by $\nu_1^\star$. Since $p,q \leq m\log n$, we conclude with

$$\log\mathrm{Card}(\mathfrak{B}_{p,q}^\star(\ell)) \leq C_1(m\log n)^{m+1} + C_2\sqrt{\nu_1^\star 2^{p-\ell}(m\log n)^{m+3}}$$

for some absolute constants $C_1, C_2 > 0$. Now, we can prove the bound for $\mathbb{P}\big(\max_{V\in\mathfrak{B}_{p,q}^\star(\ell)}\langle\mathcal{R}\odot\mathcal{A},V\rangle \geq 2^{-(m+1)}t/2\big)$. By Bernstein inequality,

$$\mathbb{P}\Big(\max_{V\in\mathfrak{B}_{p,q}^\star(\ell)}\langle\mathcal{R}\odot\mathcal{A},V\rangle \geq 2^{-(m+1)}t/(2m^2\log^2 n)\Big) \leq \exp\Big(4mn - \frac{2^{\ell-2m-2}t^2}{2(m!)\delta^2\beta(\mathcal{Q})m^4\log^4 n}\Big)$$
$$+ \exp\Big(C_1(m\log n)^{m+1} + C_2\sqrt{\nu_1^\star 2^{p-\ell}(m\log n)^{m+3}} - \frac{(3/4)2^{(p+q)/2}2^{-m-1}t}{2(m!)m^2\log^2 n}\Big).$$

Since $q \geq (m-2)\lceil\log\delta^{-2}\rceil$, we have $2^{-q/2} \leq \delta^{m-2}$. By choosing (recall that $\nu_1^\star \geq \log n$)

$$t \geq \max\Big\{C_1 2^{m+3}\sqrt{m^5(m!)n\delta^2\beta(\mathcal{Q})}\log^2 n, \frac{C_2(m!)}{3}(4m\log n)^{\frac{m+7}{2}}\sqrt{\nu_1^\star}\delta^{m-2}\Big\}$$

for some absolute constants $C_1, C_2 > 0$, then

$$\mathbb{P}\Big(\max_{V\in\mathfrak{B}_{p,q}^\star(\ell)}\langle\mathcal{R}\odot\mathcal{A},V\rangle \geq 2^{-(m+1)}t/(2m^2\log^2 n)\Big)$$
$$\leq \exp\Big(-\frac{2^{-2m-2}t^2}{4(m!)\delta^2\beta(\mathcal{Q})m^4\log^4 n}\Big) + \exp\Big(-\frac{3}{16}\cdot\frac{2^{-m-1}t}{2(m!)\delta^{m-2}m^2\log^2 n}\Big).$$

By combining *Case 1* and *Case 2* and observing $n\theta_{\mathsf{max}} \geq \|\theta\|_1$, if we choose

$$t \geq (m!)2^m \cdot \max\Big\{C_1\sqrt{m^5 n\delta^2\beta(\mathcal{Q})}\log^2 n, \ C_2 m(m\log n)^{(m+7)/2}\delta^{m-2}\sqrt{\mathcal{P}_{\mathsf{max}}\theta_{\mathsf{max}}^2\|\theta\|_1^{m-2}n\log n}\Big\}$$

, then we get (union bound for all $0 \leq \ell \leq p \leq 3\log n$)

$$\mathbb{P}\Big(\max_{u_1\otimes\cdots u_m\in\overline{U}_{1,2}(\delta)}\langle\mathcal{R}\odot\mathcal{A},(\mathcal{P}_{A_p}U_{1,2})\otimes(\mathcal{P}_{B_{p,q}}U_{3,\cdots,m})\rangle \geq 2^{-(m+1)}t/(2m^2\log^2 n)\Big)$$
$$\leq (3\log n)\exp\Big(-\frac{2^{-2m-2}t^2}{8(m!)\delta^2\beta(\mathcal{Q})m^4\log^4 n}\Big) + (3\log n)\exp\Big(-\frac{3}{16}\cdot\frac{2^{-(m+1)}t}{2(m!)\delta^{m-2}m^2\log^2 n}\Big).$$

Now, by making the bound uniform over all pairs of $(p,q)$, we obtain (since $p_{1,2} \leq 3\log n$)

$$\mathbb{P}\Big(\sum_{0\leq p\leq p_{1,2}}\sum_{q=(m-2)\lceil\log\delta^{-2}\rceil}^{(m-2)p^\star}\max_{u_1\otimes\cdots u_m\in\overline{U}_{1,2}(\delta)}\langle\mathcal{R}\odot\mathcal{A},(\mathcal{P}_{A_p}u_{1,2})\otimes(\mathcal{P}_{B_{p,q}}u_{3,\cdots,m})\rangle \geq 2^{-(m+1)}t/2\Big)$$
$$\leq (9m\log^2 n)\exp\Big(-\frac{2^{-2m-2}t^2}{8(m!)\delta^2\beta(\mathcal{Q})m^4\log^4 n}\Big) + (9m\log^2 n)\exp\Big(-\frac{3}{16}\cdot\frac{2^{-(m+1)}t}{2(m!)\delta^{m-2}m^2\log^2 n}\Big).$$

**Bounding $\langle \mathcal{R} \odot \mathcal{A}, \mathcal{P}_{S_{1,2}}(U_{1,2}) \otimes U_{3,\cdots,m} \rangle$.** Clearly, for $V \in \mathcal{P}_{S_{1,2}}(U_{1,2}) \otimes U_{3,\cdots,m}$,

$$\left| \mathcal{R}(i_1, \cdots, i_m) \mathcal{A}(i_1, \cdots, i_m) \sum_{(i'_1, \cdots, i'_m) \in \mathfrak{P}(i_1, \cdots, i_m)} V(i'_1, \cdots, i'_m) \right|$$
$$\leq (m!) \|V\|_{\mathsf{max}} \leq (m!) 2^{-(1+p_{1,2})/2} \delta^{m-2}.$$

Similarly, if $m \geq 3$, then

$$\sum_{(i_1, \cdots, i_m) \in \mathfrak{P}(n,m)} \mathbb{E}\mathcal{A}(i_1, \cdots, i_m) \left( \sum_{(i'_1, \cdots, i'_m) \in \mathfrak{P}(i_1, \cdots, i_m)} V(i'_1, \cdots, i'_m) \right)^2 \leq (m!) \delta^2 \beta(\mathcal{Q}).$$

Then, by Bernstein inequality,

$$\mathbb{P}\Big( \max_{U \in \overline{U}_{1,2}(\delta)} |\langle \mathcal{R} \odot \mathcal{A}, \mathcal{P}_{S_{1,2}}(U_{1,2}) \otimes U_{3,\cdots,m} \rangle| \geq 2^{-(m+1)} t/2 \Big) \leq \exp\Big( 4mn - \frac{2^{-2m-2} t^2}{4(m!)\delta^2\beta(\mathcal{Q})} \Big)$$
$$+ \exp\Big( 4mn - \frac{(3/4)2^{-m-1} t}{2(m!)\delta^{m-2}2^{-(1+p_{1,2})/2}} \Big).$$

By choosing $p_{1,2} = \lceil 2 \log n \rceil$ so that $2^{-p_{1,2}/2} \leq n^{-1}$, then if

$$t \geq \max \Big\{ 6 \cdot 2^{m+1} \sqrt{m(m!)\delta^2 n \beta(\mathcal{Q})}, \frac{64m}{3} 2^{m+1} \delta^{m-2} \Big\}$$

, we get

$$\mathbb{P}\Big( \max_{U \in \overline{U}_{1,2}(\delta)} |\langle \mathcal{R} \odot \mathcal{A}, \mathcal{P}_{S_{1,2}}(U_{1,2}) \otimes U_{3,\cdots,m} \rangle| \geq 2^{-(m+1)} t/2 \Big) \leq \exp\Big( -\frac{2^{-2m-2} t^2}{8(m!)\delta^2\beta(\mathcal{Q})} \Big)$$
$$+ \exp\Big( -\frac{3}{16} \cdot \frac{2^{-m-1} nt}{(m!)\delta^{m-2}} \Big).$$

**Putting them together.** By choosing (recall that $\mathcal{P}_{\mathsf{max}} \theta_{\mathsf{max}} \|\theta\|_1 \geq \log n$)

$$t \geq (m!) 2^m \cdot \max \Big\{ C_1 \sqrt{m^5 n \delta^2 \beta(\mathcal{Q})} \log^2 n, \ C_2 m (m \log n)^{(m+7)/2} \delta^{m-2} \sqrt{\mathcal{P}_{\mathsf{max}} \theta_{\mathsf{max}}^2 \|\theta\|_1^{m-2} n \log n} \Big\} \tag{B.3}$$

for some absolute constants $C_1, C_2 > 0$, we get (recall the event $\mathcal{E}_0$ and $\mathcal{E}_1$)

$$\mathbb{P}\Big\{ \sup_{u_1 \otimes \cdots \otimes u_m \in \mathcal{U}_{1,2}(\delta)} \langle \mathcal{R} \odot \mathcal{A}, u_1 \otimes \cdots \otimes u_m \rangle \geq t \Big\}$$
$$\leq \mathbb{P}\Big\{ \sup_{u_1 \otimes \cdots \otimes u_m \in \overline{\mathcal{U}}_{1,2}(\delta)} \langle \mathcal{R} \odot \mathcal{A}, u_1 \otimes \cdots \otimes u_m \rangle \geq 2^{-m-1} \cdot t \Big\} \leq 2n^{-2}$$
$$+ (10m \log^2 n) \exp\Big( -\frac{2^{-2m-2} t^2}{8(m!)\delta^2\beta(\mathcal{Q}) m^4 \log^4 n} \Big) + (10m \log^2 n) \exp\Big( -\frac{3}{16} \cdot \frac{2^{-(m+1)} t}{2(m!)\delta^{m-2} m^2 \log^2 n} \Big).$$

Finally, we conclude that if $t$ is chosen as (B.3)

$$\mathbb{P}\{\|\mathcal{A} - \mathbb{E}\mathcal{A}\| \geq 3t\} \leq 2n^{-2}$$

$$+(10m^3\log^2 n)\exp\Big(-\frac{2^{-2m-2}t^2}{8(m!)\delta^2\beta(\mathcal{Q})m^4\log^4 n}\Big) + (10m^3\log^2 n)\exp\Big(-\frac{3}{16}\cdot\frac{2^{-(m+1)}t}{2(m!)\delta^{m-2}m^2\log^2 n}\Big)$$

which concludes the proof.

## B.2  Proof of Theorem 4

For every $t \geq 0$, we define

$$E_t := \|\widehat{\Xi}^{(t)}\big(\widehat{\Xi}^{(t)}\big)^\top - \Xi\Xi^\top\|.$$

As shown in the proof of Remark 6.2 of Keshavan et al. (2010), the regularization procedure produces an output $\widehat{\Xi}^{(t)}_\star$ so that if $E_t < 1/3$, then

$$\|\widehat{\Xi}^{(t)}_\star\widehat{\Xi}^{(t)\top}_\star - \Xi\Xi^\top\| \leq \sqrt{\frac{1+3E_t}{1-3E_t}}\cdot E_t$$

and $\|\widehat{\Xi}^{(t)}_\star\|_{2,\max} \leq \delta\sqrt{1+3E_t} \leq 2\delta$ where $\delta$ is the regularization parameter chosen in algorithm. In addition, the warm initialization guarantees that $E_0 \leq \varepsilon_0 < 1/4$.

By the definition of HOOI, we write

$$\widehat{\Xi}^{(t+1)} = \mathrm{SVD}_K\Big(\mathcal{M}_1\big(\mathcal{A}\underset{k=2}{\overset{m}{\times}}(\widehat{\Xi}^{(t)}_\star)^\top\big)\Big)$$

where $\mathcal{M}_1(\cdot)$ denotes the matricization such that $\mathcal{M}_1(\mathcal{A}) \in \mathbb{R}^{n \times n^{m-1}}$ and $\mathrm{SVD}_K$ returns the top-$K$ left singular vectors. Recall that

$$\mathcal{A} = \mathcal{Q} + \mathcal{A} - \mathcal{Q} \quad\text{and}\quad \mathcal{A} - \mathcal{Q} = (\mathcal{A} - \mathbb{E}\mathcal{A}) - \mathrm{diag}(\mathcal{Q})$$

We write

$$\mathcal{M}_1\big(\mathcal{A}\underset{k=2}{\overset{m}{\times}}(\widehat{\Xi}^{(t)}_\star)^\top\big) = \mathcal{M}_1(\mathcal{Q})\big(\widetilde{\bigotimes}_{k=2}^m\widehat{\Xi}^{(t)}_\star\big) + \mathcal{M}_1\big((\mathcal{A}-\mathcal{Q})\underset{k=2}{\overset{m}{\times}}(\widehat{\Xi}^{(t)}_\star)^\top\big)$$

where $\widetilde{\bigotimes}$ denotes the Kronecker product. Observe that top-$K$ left singular vectors of $\mathcal{M}_1(\mathcal{Q})\big(\widetilde{\bigotimes}_{k=2}^m\widehat{\Xi}^{(t)}_\star\big)$ span the same column space as $\Xi$. Clearly,

$$\mathcal{M}_1(\mathcal{Q})\big(\widetilde{\bigotimes}_{k=2}^m\widehat{\Xi}^{(t)}_\star\big) = \Xi\mathcal{M}_1(\mathcal{C})\big(\widetilde{\bigotimes}_{k=2}^m(\Xi^\top\widehat{\Xi}^{(t)}_\star)\big).$$

Therefore,

$$\sigma_K\Big(\mathcal{M}_1(\mathcal{Q})\big(\widetilde{\bigotimes}_{k=2}^m\widehat{\Xi}^{(t)}_\star\big)\Big) \geq \sigma_K(\mathcal{M}_1(\mathcal{C}))\sigma_{\min}^{m-1}(\Xi^\top\widehat{\Xi}^{(t)}_\star)$$

$$\geq \sigma_{\min}^{m-1}(\Xi^\top\widehat{\Xi}^{(t)}_\star)\cdot\kappa_K\|\theta\|^m$$

$$\geq \big(1-\sqrt{(1+3E_t)/(1-3E_t)}E_t\big)^{(m-1)/2}\cdot\kappa_K\|\theta\|^m$$

where we used the fact

$$\sigma_{\min}(\Xi^\top \widehat{\Xi}_\star^{(t)}) = \sqrt{\sigma_{\min}(\Xi^\top \widehat{\Xi}_\star^{(t)} (\widehat{\Xi}_\star^{(t)})^\top \Xi)}$$

$$= \sqrt{\sigma_{\min}\left(I + \Xi^\top \left(\widehat{\Xi}_\star^{(t)} (\widehat{\Xi}_\star^{(t)})^\top - \Xi\Xi^\top\right)\Xi\right)} \geq \sqrt{1 - \|\widehat{\Xi}_\star^{(t)} (\widehat{\Xi}_\star^{(t)})^\top - \Xi\Xi^\top\|}$$

$$\geq \sqrt{1 - \sqrt{(1 + 3E_t)/(1 - 3E_t)}E_t}.$$

If $E_t \leq 1/4$, then

$$\sigma_K\left(\mathcal{M}_1(\mathcal{Q})\left(\widetilde{\bigotimes}_{k=2}^m \widehat{\Xi}_\star^{(t)}\right)\right) \geq (1 - 3\varepsilon_0)^{(m-1)/2}\kappa_K\|\theta\|^m.$$

As the proof of Lemma 1 shows, the effect of diagonal tensor is bounded as

$$\|\mathcal{M}_1(\operatorname{diag}(\mathcal{Q}))\| \leq m^2 \kappa_1 d_{\min}^{-2} \cdot \theta_{\max}^2 \|\theta\|^{m-2}.$$

Recall that

$$\mathcal{M}_1\left((\mathcal{A} - \mathcal{Q})\bigtimes_{k=2}^m (\widehat{\Xi}_\star^{(t)})^\top\right) = \mathcal{M}_1\left((\mathcal{A} - \mathbb{E}\mathcal{A})\bigtimes_{k=2}^m (\widehat{\Xi}_\star^{(t)})^\top\right)$$

$$+ \mathcal{M}_1\left((\operatorname{diag}(\mathcal{Q}))\bigtimes_{k=2}^m (\widehat{\Xi}_\star^{(t)})^\top\right). \tag{B.4}$$

We begin with bounding the first term on the right hand side of (B.4). Observe that

$$\bigtimes_{k=2}^m (\widehat{\Xi}_\star^{(t)}) = \sum_{j=2}^m \left(\bigtimes_{k=2}^{j-1} \mathcal{P}_\Xi \widehat{\Xi}_\star^{(t)}\right) \times_j \mathcal{P}_\Xi^\perp \widehat{\Xi}_\star^{(t)} \left(\bigtimes_{k=j+1}^m \widehat{\Xi}_\star^{(t)}\right)$$

$$+ \bigtimes_{k=2}^m \mathcal{P}_\Xi \widehat{\Xi}_\star^{(t)}$$

where $\mathcal{P}_\Xi$ denotes the projection onto $\Xi$, i.e., $\mathcal{P}_\Xi = \Xi\Xi^\top$. Then, $\mathcal{P}_\Xi^\perp = \mathcal{I} - \mathcal{P}_\Xi$. We then write

$$\mathcal{M}_1\left((\mathcal{A} - \mathbb{E}\mathcal{A})\bigtimes_{k=2}^m (\widehat{\Xi}_\star^{(t)})^\top\right)$$

$$= \underbrace{\mathcal{M}_1\left((\mathcal{A} - \mathbb{E}\mathcal{A})\bigtimes_{k=2}^m (\mathcal{P}_\Xi \widehat{\Xi}_\star^{(t)})^\top\right)}_{\mathcal{R}_1}$$

$$+ \underbrace{\sum_{j=2}^m \mathcal{M}_1\left((\mathcal{A} - \mathbb{E}\mathcal{A})\left(\bigtimes_{k=2}^{j-1} \mathcal{P}_\Xi \widehat{\Xi}_\star^{(t)}\right)^\top \times_j (\mathcal{P}_\Xi^\perp \widehat{\Xi}_\star^{(t)})^\top \left(\bigtimes_{k=j+1}^m \widehat{\Xi}_\star^{(t)}\right)^\top\right)}_{\mathcal{R}_2}.$$

The following lemma establishes a simple connection between the tensor spectral norm and matrix spectral norm.

**Lemma 2.** (Xia et al., 2021, Lemma 6) For a $k$-th order tensor $\mathcal{A} \in \mathbb{R}^{n_1 \times \cdots \times n_k}$ with multilinear ranks $\leq (r, \cdots, r)$, the following bound holds for all $j = 1, \cdots, k$,

$$\|\mathcal{M}_j(\mathcal{A})\| \leq \|\mathcal{A}\| \cdot r^{(m-2)/2}.$$

We next bound $\|\mathcal{R}_1\|$ and $\|\mathcal{R}_2\|$. Clearly,

$$\text{rank}\big(\mathcal{P}_\Xi \widehat{\Xi}_\star^{(t)}\big) \leq K, \quad \text{and} \quad \text{rank}\big(\mathcal{P}_\Xi^\perp \widehat{\Xi}_\star^{(t)}\big) \leq K.$$

By Lemma 2,

$$\|\mathcal{R}_2\| \leq K^{(m-2)/2} \cdot \left\| (\mathcal{A} - \mathbb{E}\mathcal{A}) \Big( \bigtimes_{k=2}^{j-1} \mathcal{P}_\Xi \widehat{\Xi}_\star^{(t)} \Big)^\top \times_j (\mathcal{P}_\Xi^\perp \widehat{\Xi}_\star^{(t)})^\top \Big( \bigtimes_{k=j+1}^{m} \widehat{\Xi}_\star^{(t)} \Big)^\top \right\|.$$

Note the following fact

$$\|\mathcal{P}_\Xi^\perp \widehat{\Xi}_\star^{(t)}\| = \left\| \Big( \widehat{\Xi}_\star^{(t)} (\widehat{\Xi}_\star^{(t)})^\top - \Xi \Xi^\top \Big) \widehat{\Xi}_\star^{(t)} \right\| \leq \sqrt{\frac{1 + 3E_t}{1 - 3E_t}} E_t.$$

Recall that $\|\widehat{\Xi}_\star^{(t)}\|_{2,\max} \leq 2\delta$ and $\|\Xi\|_{2,\max} \leq \delta$. By the definition of incoherent norm, we have

$$\left\| (\mathcal{A} - \mathbb{E}\mathcal{A}) \Big( \bigtimes_{k=2}^{j-1} \mathcal{P}_\Xi \widehat{\Xi}_\star^{(t)} \Big)^\top \times_j (\mathcal{P}_\Xi^\perp \widehat{\Xi}_\star^{(t)})^\top \Big( \bigtimes_{k=j+1}^{m} \widehat{\Xi}_\star^{(t)} \Big)^\top \right\| \leq \|\mathcal{A} - \mathbb{E}\mathcal{A}\|_{2\delta} \cdot \sqrt{\frac{1 + 3E_t}{1 - 3E_t}} E_t.$$

By Theorem 5,

$$\frac{\|\mathcal{R}_2\|}{E_t \cdot K^{(m-2)/2}} \leq 3\|\mathcal{A} - \mathbb{E}\mathcal{A}\|_{2\delta}$$
$$\leq C_1(m!)2^m \sqrt{mn\delta^2 \beta(\mathcal{Q})} + C_2(m!)2^m (m \log n)^{(m+3)/2} \delta^{m-2} \sqrt{\mathcal{P}_{\max} \theta_{\max}^2 \|\theta\|_1^{m-2} n \log n} \qquad \text{(B.5)}$$

which holds with probability at least $1 - n^{-2}$.

Next, we bound $\|\mathcal{R}_1\| = \|\mathcal{M}_1((\mathcal{A} - \mathbb{E}\mathcal{A}) \times_{k=2}^m (\mathcal{P}_\Xi \widehat{\Xi}_\star^{(t)})^\top)\|$. The following fact is obvious.

$$\|\mathcal{M}_1((\mathcal{A} - \mathbb{E}\mathcal{A}) \times_{k=2}^m (\mathcal{P}_\Xi \widehat{\Xi}_\star^{(t)})^\top)\| \leq \|\mathcal{M}_1((\mathcal{A} - \mathbb{E}\mathcal{A}) \times_2 \Xi^\top \times_3 \cdots \times_m \Xi^\top)\|.$$

**Lemma 3.** The following bound holds with probability at least $1 - n^{-2}$,

$$\left\| \mathcal{M}_1\Big( (\mathcal{A} - \mathbb{E}\mathcal{A}) \bigtimes_{k=2}^m \Xi^\top \Big) \right\| \leq C_1 \sqrt{(m!)K^{m-2} \beta(\mathcal{Q}) \log n} + C_2(m!)\delta^{m-1} \log n.$$

By Davis-Kahan Theorem, then

$$\left\| \widehat{\Xi}^{(t+1)} \big( \widehat{\Xi}^{(t+1)} \big)^\top - \Xi \Xi^\top \right\| \leq \frac{2\|\mathcal{R}_1\| + 2\|\text{diag}(\mathcal{Q})\| + 2E_t K^{(m-2)/2} \|\mathcal{A} - \mathbb{E}\mathcal{A}\|_{2\delta}}{(1 - 3\varepsilon_0)^{(m-1)/2} \kappa_K \|\theta\|^m}$$
$$\leq \frac{2^m(\|\mathcal{R}_1\| + \|\text{diag}(\mathcal{Q})\|)}{\kappa_K \|\theta\|^m} + 2E_t K^{(m-2)/2} \frac{\|\mathcal{A} - \mathbb{E}\mathcal{A}\|_{2\delta}}{(1 - 3\varepsilon_0)^{(m-1)/2} \kappa_K \|\theta\|^m}$$

Therefore, we obtain

$$E_{t+1} \leq \frac{2^m \|\mathcal{R}_1\| + 2^m \|\text{diag}(\mathcal{Q})\|}{\kappa_K \|\theta\|^m} + 2E_t K^{(m-2)/2} \frac{\|\mathcal{A} - \mathbb{E}\mathcal{A}\|_{2\delta}}{(1 - 3\varepsilon_0)^{(m-1)/2} \kappa_K \|\theta\|^m}.$$

To ensure the contraction property, it requires

$$(1 - 3\varepsilon_0)^{(m-1)/2}\kappa_K\|\theta\|^m \geq 4K^{(m-2)/2}\|\mathcal{A} - \mathbb{E}\mathcal{A}\|_\delta. \tag{B.6}$$

By Theorem 5, it requires that

$$(1-3\varepsilon_0)^{(m-1)/2}\kappa_K\|\theta\|^m$$
$$\geq C_1(m!)(4K)^{m/2}\max\left\{\sqrt{mn\delta^2\beta(\mathcal{Q})}, (m\log n)^{(m+3)/2}\delta^{m-2}\sqrt{\mathcal{P}_{\max}\theta_{\max}^2\|\theta\|_1^{m-2}n\log n}\right\}$$

for some large enough absolute constant $C_1 > 0$. Then, by Lemma 3,

$$E_{t+1} \leq 2^m\frac{\|\mathcal{R}_1\| + \|\mathcal{M}_1(\mathrm{diag}(\mathcal{Q}))\|}{\kappa_K\|\theta\|^m} + \frac{1}{2}E_t$$
$$\leq \frac{E_t}{2} + C_1 2^m\frac{\sqrt{(m!)K^{m-2}\beta(\mathcal{Q})\log n} + (m!)\delta^{m-1}\log n + \kappa_1(m!)\|\theta\|^{m-2}\theta_{\max}^2/d_{\min}^2}{\kappa_K\|\theta\|^m}$$

which proves the first claim.

Then, after at most

$$T = O\left(1 \vee m\log\frac{n\|\theta\|}{\theta_{\max}}\right)$$

iterations, then with probability at least $1 - 2n^{-2}$,

$$E_T \leq C_4 2^m\frac{\sqrt{(m!)K^{m-2}\beta(\mathcal{Q})\log n} + (m!)\delta^{m-1}\log n + m^2\kappa_1\theta_{\max}^2\|\theta\|^{m-2}/d_{\min}^2}{\kappa_K\|\theta\|^m}$$

which proves the theorem.

## B.3   Proof of Lemma 3

We write

$$\mathcal{M}_1(\mathcal{A})\mathop{\times}\limits_{k=2}^{m}\Xi^\top = \sum_{(i_1,\cdots,i_m)\in\mathfrak{P}(n,m)}\mathcal{A}(i_1,\cdots,i_m)\sum_{(i_1',\cdots,i_m')\in\mathfrak{P}(i_1,\cdots,i_m)}(e_{i_1'})(\widetilde{\bigotimes}_{k=2}^{m}e_{i_k'})^\top(\tilde{\otimes}_{k=2}^{m}\Xi)^\top$$

where $e_i$ denotes the $i$-th canonical basis vector in $\mathbb{R}^n$. For $(i_1,\cdots,i_m) \in \mathfrak{P}(n,m)$, denote a matrix

$$Z_{i_1,\cdots,i_m} = (\mathcal{A}(i_1,\cdots,i_m) - \mathbb{E}\mathcal{A}(i_1,\cdots,i_m))\sum_{(i_1',\cdots,i_m')\in\mathfrak{P}(i_1,\cdots,i_m)}(e_{i_1'})(\widetilde{\bigotimes}_{k=2}^{m}e_{i_k'})^\top(\widetilde{\bigotimes}_{k=2}^{m}\Xi)^\top$$

so that

$$\mathcal{M}_1\big((\mathcal{A} - \mathbb{E}\mathcal{A})\mathop{\times}\limits_{k=2}^{m}\Xi^\top\big) = \sum_{(i_1,\cdots,i_m)\in\mathfrak{P}(n,m)}Z_{i_1,\cdots,i_m}$$

which is a sum of independent random matrices. Clearly, $\|Z_{i_1,\cdots,i_m}\| \leq (m!)\delta^{m-1}$ since $\max_i\|e_i^\top\Xi\| \leq \delta$. In addition, we need the upper bound of $\|W\|$ where the symmetric matrix $W$ is defined by

$$W = \sum_{(i_1,\cdots,i_m)\in\mathfrak{P}(n,m)}\mathbb{E}Z_{i_1,\cdots,i_m}Z_{i_1,\cdots,i_m}^\top.$$

For any $u \in \mathbb{R}^n$ with $\|u\| \le 1$, observe that

$$\langle \mathbb{E}Z_{i_1,\cdots,i_m}Z_{i_1,\cdots,i_m}^\top, u \otimes u \rangle \le \mathcal{Q}(i_1,\cdots,i_m)\Big\| \sum_{(i_1',\cdots,i_m')} u_{i_1'}^2 \left(\Xi(i_2',:)\tilde{\otimes}\cdots\tilde{\otimes}\Xi(i_m',:)\right)\Big\|^2$$

$$\le (m!)\mathcal{Q}(i_1,\cdots,i_m) \sum_{(i_1',\cdots,i_m')\in\mathfrak{P}(i_1,\cdots,i_m)} u_{i_1'}^2 \|\Xi(i_2',:)\|^2 \cdots \|\Xi(i_m',:)\|^2$$

$$\le (m!)\mathcal{Q}(i_i,\cdots,i_m)u_{i_1}^2 \frac{\theta_{i_2}^2}{\|\theta\|^2}\cdots\frac{\theta_{i_m}^2}{\|\theta\|^2}.$$

Then, we get

$$\langle W, u \otimes u \rangle \le (m!)K^{m-1}\mathcal{Q}_{\mathsf{max}}.$$

implying that $\|W\| \le (m!)K^{m-1}\mathcal{Q}_{\mathsf{max}}$. In a similar fashion, we can show that

$$\Big\| \sum_{(i_1,\cdots,i_m)\in\mathfrak{P}(n,m)} \mathbb{E}Z_{i_1,\cdots,i_m}^\top Z_{i_1,\cdots,i_m}\Big\| \le (m!)K^{m-2}\beta(\mathcal{Q}).$$

By matrix Bernstein inequality (Tropp (2012)), with probability at least $1 - n^{-2}$,

$$\Big\|\mathcal{M}_1\big((\mathcal{A} - \mathbb{E}\mathcal{A}) \overset{m}{\underset{k=2}{\times}} \Xi^\top\big)\Big\| \le C_1\sqrt{(m!)K^{m-2}\beta(\mathcal{Q})\log n} + C_2(m!)\delta^{m-1}\log n$$

which proves the lemma.

