# OpenReview forum: "Community Detection for Hypergraph Networks via Regularized Tensor Power Iteration"
_SLADS/Section_A — Under review for SLADS_Section_A_

### Review · Reviewer_7pJS · 2026-07-14

**Summary Of Contributions:**

The paper develops Tensor-SCORE, a spectral community-detection method that operates directly on hypergraph adjacency tensors rather than first projecting a hypergraph onto a graph. Its principal methodological component is a regularized HOOI, reg-HOOI, which truncates high-leverage rows during tensor power iteration to improve stability in sparse settings. The estimated tensor factor matrix is then normalized using a hypergraph extension of SCORE to reduce the influence of node-level degree heterogeneity.

The paper also introduces a hypergraph degree-corrected block model, hDCBM, and characterizes its population tensor eigenspace. Under this model, the authors derive consistency and misclassification-error guarantees for Tensor-SCORE, analyze two proposed initialization procedures, and establish a concentration inequality for an incoherent tensor operator norm.

Empirically, the method is evaluated on simulated networks and two real datasets. The experiments suggest that retaining higher-order interactions can materially improve community recovery when graph projection loses information.

**Audience:**

Yes

**Claims And Evidence:**

Yes

**Requested Changes:**

1.In Theorem 1, the value of T and \delta for the algorithm are different from the choice in Eq. (3.9). Would it be possible to analyze the proposed algorithm on p. 11?

2.The choice of K in Eq. (3.10) seems to be weird. Because consecutive ordered singular-value ratios are at most one, while log(log(n)) is greater than 1 for moderately large n, this condition would ordinarily select r. Is there a typo?

3.The condition \kappa_1 - \kappa_2 \geq \gamma_n in Eq. (4.2) looks confusing. Could the authors explain why it is needed?

4.I observe that in the numerical studies, the tensor-SCORE uses two different initialization methods than the ones discussed in Section 4.2. Would it not be better to be consistent?

**Strengths And Weaknesses:**

Strength: The paper addresses an important limitation of conventional high-order network analysis: graph projection can erase information contained in higher-order interactions. The population example in Section 3.4 gives a useful mathematical illustration of how a projected signal matrix can become rank-deficient even when the original tensor signal remains informative. The legislator visualization on page 3 also communicates this issue effectively.

The authors provide a solid theoretical foundation for their methods. The analysis permits degree heterogeneity, growing numbers of communities, sparse networks, and nontrivial initialization error. The contraction result for reg-HOOI and the incoherent-norm concentration bound are technically substantial.


Weakness: The methodology is based on tensor decomposition and the SCORE method for community detection for 2-d graph data, which I think are relatively standard. The only difference is that we have to perform reg-HOOI for tensor decomposition due to the special network data structure.

There is a mismatch between the proposed Tensor-SCORE and the theoretically analyzed algorithm.

---

### Review · Reviewer_XBD7 · 2026-07-21

**Summary Of Contributions:**

This manuscript introduces a new method for community detection in degree-corrected stochastic blockmodel hypergraphs using a combination of the SCORE approach of Jin (2015) together with a regularized version of the Higher-Order Orthogonal Iteration Algorithm. They show that this procedure consistently recovers communities under relatively mild conditions on the degrees and sparsity of the hypergraphs, and they propose two different initializations for the algorithm.  They apply their results to real and simulated data.

**Audience:**

Yes

**Broader Impact Concerns:**

N/a.

**Claims And Evidence:**

Yes

**Requested Changes:**

1.) The reference Jing et al. (2021), "Community Detection on Mixture Multi-layer Networks via Regularized Tensor Decomposition" uses a very similar approach, only with a different symmetry requirement in the underlying tensor.  This paper is not cited, but the ideas are very similar.  The authors should discuss this work relative to their own and clarify the novelty.

2.) The authors should discuss in which cases Assumption 4.7 is violated, or in what cases it is a stronger assumption than the other ones.

3.) The authors should at least discuss the dependence on the order $m$ in their results.

4.)  Tensor data analysis of low Tucker rank is known to exhibit statistical-to-computational gaps depending on the signal strength.  However, no discussion of this appears in this paper.  The authors should discuss how they are either able to a.) bypass the statistical-computational gap, or b.) how their results rely on a stronger signal strength.  I think this may be related to Assumption 4.7, but because the noise model is different it is hard to tease out how the two are related.

**Strengths And Weaknesses:**

I find the paper to be quite strong.  The theory, data, and simulations clearly show the benefit of the proposed methodology, and I enjoyed Section 3.4.  I have a few concerns:

1.) In the paper self-hyperedges are excluded, but in a citation network it can be very common to cite oneself.  Does the theory change significantly if self-hyperedges are permitted?

2.) One concern I have is that the theory is presented for uniform hypergraphs, though the authors present an extension to non-uniform hypergraphs.  However, for non-uniform hypergraphs, the sparsity level and degree heterogeneity may be of different levels.  For example, papers with 2-4 authors are very common, but papers with 5+ authors are far less common.  Is there a way to take this into account both algorithmically and theoretically when extending to non-uniform hypergraphs? Perhaps the results can be made to depend on some other generalized notion of connectedness/sparsity besides smallest eigenvalues?  For example, suppose the hypergraph is extremely dense for all hyperedges of order 4 or less, but sparse for hyperedges of order 5+. Could this setting be studied theoretically or empirically?

3.) The dependence on $m$ is can be quite weak in a few places.

4.) Assumption 4.7 is hard to understand.